# Prominent involvement of acetylcholine dynamics in stable olfactory representation across the *Drosophila* brain

Jiaqi Fan [1,2,3,4,5,10], Yuling Wang[1,3,4,10], Lingbo Li[1,3,4,10], Jing He[1,3,4], Zhifeng Zhao[1,3,4], Fei Deng [6,7], Guochuan Li[6,7], Xinyang Li [8], Yiliang Zhou[1,3,4], Jiayin Zhao[1,3,4], Ning Huang[4,9], Yixin Hu[1,3,4], Yulong Li [6,7], Jiamin Wu [1,3,4,8] ✉, Lu Fang [2,3] ✉ & Qionghai Dai [1,3,4,8] ✉

Despite the vital role of neuromodulators and neurotransmitters in the neural system, their spatiotemporal correlation with neuronal activities across multiple brain regions remain unclear. Here, we employed two-photon synthetic aperture microscopy (2pSAM) and neurochemical indicators to simultaneously record calcium and acetylcholine (ACh)/5-HT dynamics across multiple regions of the *Drosophila* brain over 2 h. Presenting 3 different odors across multiple trials, our analyses revealed signal-specific differences in responsiveness, functional connectivity, and odor classification accuracy across the brain. We further constructed low-dimensional manifolds to characterize the global odor-related dynamics. Incorporating both calcium and ACh signals enhanced odor classification accuracy in the global low-dimensional manifold and in specific brain regions where their functional connectivity network features exhibited complementary patterns. Moreover, ACh dynamics demonstrated relatively stable temporal characteristics compared to calcium and 5-HT. These results suggest the potential contribution of ACh to consistent odor representations and illustrate the utility of multi-signal imaging in studying neural computation.

High-throughput neuronal recording techniques have significantly advanced the understanding of spatiotemporal neuronal activities and functions[1–10]. However, these advancements stand in contrast to the gap in the accessibility of other crucial brain signals, such as neuromodulators and neurotransmitters, primarily due to existing technical limitations[11]. This gap impedes our ability to comprehensively understand the systemic functional roles of these neurochemicals[12–14] and the neural system. For instance, in sensory processing, while different neurochemicals have distinct cellular effects, the purported functions overlap substantially[15,16]. Thus the current hypotheses of the functions

and roles of particular neurochemicals may be limited[12]. Furthermore, the quantitative relationship between neuronal activity and neurochemical release in vivo is poorly characterized due to limited indicators and imaging systems for simultaneous recording[13,17–21]. Consequently, how these dynamics correlate, differ, and synergistically represent and process information remains largely unknown. Previous research reported dissociable dopamine dynamics, indicating possible firing-independent dopamine release and the insufficiency of solely recording dopamine cells for understanding dopamine signals[20]. Therefore, elucidating the spatiotemporal relationships and

[1]Department of Automation, Tsinghua University, Beijing, China. [2]Department of Electronic Engineering, Tsinghua University, Beijing, China. [3]Institute for Brain and Cognitive Science (THUIBCS), Tsinghua University, Beijing, China. [4]IDG/McGovern Institute for Brain Research, Tsinghua University, Beijing, China. [5]Tsinghua-Berkeley Shenzhen Institute (TBSI), Tsinghua Shenzhen International Graduate School, Tsinghua University, Shenzhen, China. [6]State Key Laboratory of Membrane Biology, School of Life Sciences, Peking University, Beijing, China. [7]PKU-IDG/McGovern Institute for Brain Research, Beijing, China. [8]College of AI, Tsinghua University, Beijing, China. [9]MOE Key Laboratory for Protein Science, School of Life Sciences, Tsinghua University, Beijing, China. [10]These authors contributed equally: Jiaqi Fan, Yuling Wang, Lingbo Li. ✉e-mail: wujiamin@tsinghua.edu.cn; fanglu@tsinghua.edu.cn; qhdai@tsinghua.edu.cn

information representation of neuronal activities and neurochemical dynamics in a systemic view is critical for advancing our understanding of the neural system.

A typical example is the representation of sensory information, specifically olfaction, which is considered a fundamental cognitive function of the brain with significant involvement of neuromodulators and neurotransmitters[22–25]. Despite the longstanding research interest in olfactory processing, comprehension of olfactory representation in terms of neuronal activity and neurochemical dynamics remains largely unexplored. The current understanding of olfactory processing is biased towards early brain layers, leaving olfactory information representation and arrangement in higher-order regions elusive[26–32]. Moreover, there is evidence of representational drift in neuronal activity related to olfaction and other sensory modalities occurring over hours and days[1,2], yet the extent of these changes requires further characterization. Neurochemicals, such as acetylcholine (ACh)[16,33–35] and serotonin (5-HT)[12,15,23], play crucial roles in olfactory processing and memory. Nonetheless, their spatiotemporal dynamics in relation to olfactory stimuli across the brain over extended periods remain an open question. Recent rapid advancements in fluorescent neurochemical indicators and high-throughput microscopy techniques offer the potential for large field-of-view (FOV) simultaneous recording of neuronal activities and neurochemical dynamics, which can enlighten us regarding these unresolved issues. A recent study utilizing these techniques revealed spatiotemporal heterogeneous coordination of cholinergic and neocortical activity across different waking states[13]. However, there is still a gap in understanding information representation by neuronal activities and neurochemical dynamics.

Here, we employed two-photon synthetic aperture microscopy (2pSAM) for 2-h (2 h) high-speed volumetric imaging[36] across multiple regions of the *Drosophila* brain, which was pan-neuronally labeled by the green-fluorescent calcium indicator jGCaMP7f (G7f)[37] and a red-fluorescent indicator (rGRAB_ACh-0.5 (rACh) for ACh, or rGRAB_HTR2C-0.5 (r5-HT) for 5-HT)[38,39], to elucidate the spatiotemporal odor representation of neuronal activities and neurochemical dynamics. Unsupervised deep-learning-based denoising algorithms were applied to increase the signal-to-noise ratio[40–42]. Our findings indicated diverse response properties, functional connectivity, and odor representation across the brain and among these dynamic signals. In accordance with the good performance of multiple-brain-region odor identity representation, we identified odor-specific representation ensembles in physical space. Furthermore, integrating ACh dynamics improved the performance of odor identity representation by calcium, suggesting that ACh dynamics may provide additional features beyond simple correlation with calcium signals. Consistently, voxel-level functional connectivity networks of ACh in specific brain regions[43] also exhibited connectivity strength complementation to the networks of calcium signals. Analyzing the low-dimensional manifolds of olfactory responses, we further characterized odor-specific and neurochemical-specific representations and the temporal changes. We found that ACh dynamics exhibited higher temporal stability compared to calcium or 5-HT dynamics in the low-dimensional manifolds and functional connectivity networks over 2 h. These findings suggest a potential avenue for future studies on systemic information representation and neural networks encompassing both calcium and neurochemical dynamics, and indicate potential contributions of neurochemicals in information representation.

## Results

### Multiple-brain-region recording of neuronal activities and neurochemical dynamics

To simultaneously record neuronal activities and neurochemical dynamics across the *Drosophila* brain, we employed a three-step crossbreeding method to generate flies with pan-neuronal expression of G7f together with either rACh or r5-HT. We surgically exposed the whole central brain by gently opening the posterior head cuticle of the fly and keeping the posterior brain surface flat. 2pSAM was used for dual-color volumetric imaging of the *Drosophila* brain (Fig. 1a), with the FOV measuring 458.7 μm × 458.7 μm × 100 μm. This FOV covered nearly entire lateral range and about half of the axial range of the fly central brain (about 80 to 180 μm under the surface), corresponding to approximately 43 brain regions (Fig. 1b, c; Supplementary Table 1 and Supplementary Movie 1)[3,44].

To study neuronal activities and neurochemical dynamics during odor stimulation, we administered three different single compounds as olfactory stimuli—3-octanol (OCT), 4-methylcyclohexanol (MCH), and ethyl acetate (EA)—to the flies in a pseudo-random order (Fig. 1a and Supplementary Movie 1) and recorded the olfactory responses (Supplementary Fig. 1a–d). At the concentration adopted in this study, OCT and MCH are two comparably representative aversive odors for flies, whereas EA is relatively more attractive[45–48], which enable the investigation of odor identity as well as latent preference. Utilizing the 3-dimensional (3D) imaging ability and low phototoxicity of 2pSAM[36], we conducted volumetric recording of neuronal activities and neurochemical dynamics at a sampling rate of 30 Hz for approximately two hours, with 180 trials (60 sessions) of odor stimulation in total, preceded by a 10-min resting-state period (Fig. 1a, d). This approach facilitated the analysis of odor representation and temporal changes across multiple brain regions. Throughout the imaging process, the flies maintained a favorable condition and exhibited consistent odor responses (Fig. 1d). We employed denoising algorithms[40–42] to extract the temporal traces with a high signal-to-noise ratio (SNR) for more than 200,000 voxels across multiple brain regions per fly (Fig. 1c). Heterogeneity and distinction of responses are observed for calcium, ACh, and 5-HT (Fig. 1b–d; and Supplementary Movie 1), which will be discussed in detail in the following section.

### Diverse olfactory responses and functional connectivity across the brain and among the dynamic signals

Calcium, ACh and 5-HT signals all response to odor stimuli (Fig. 1c, d). We analyzed the distribution, intensity, and dynamic characteristics of their responses and observed diverse response patterns across the brain.

To examine the distribution of responses, we calculated the correlation between odor stimuli and the dynamic signals as a measurement of responsiveness (Fig. 2a and Supplementary Fig. 1e, f). We found that three dynamics displayed distinction in responsiveness distribution across the brain (two-way ANOVA for factors dynamics and brain regions: $P < 0.0001$ for both factors, $F = 43.2392$, degrees of freedom = 2, and effect size (partial eta-squared) = 0.7004 for dynamics. The non-parametric Scheirer–Ray-Hare test yielded similar significance; see the Source data for details). For calcium, all olfactory regions exhibit higher responsiveness than the average level of non-olfactory regions (Supplementary Fig. 1f; see the Source data for detailed statistical results). For ACh, among olfactory regions, the LH and SLP show low responsiveness comparable to non-olfactory regions (Fig. 2a, Supplementary Fig. 1f). This result differs from previous findings that the lateral protocerebrum, encompassing these regions, is a crucial higher-order olfactory center implicated in ACh-mediated modulation[29]. In contrast, the responsiveness of ACh in the MB is relatively high (Fig. 2a, Supplementary Fig. 1f), implying a potentially critical role of ACh in the MB[49,50]. Regarding 5-HT, it exhibits low overall responsiveness across all regions (Supplementary Fig. 1f), relevant to its fast decrease in response in initial trials (Fig. 1d). Among olfactory regions, the LH exhibits even lower responsiveness for 5-HT (Fig. 2a, Supplementary Fig. 1f).

Response intensity was quantified using the standard deviation and the area under the curve (AUC) of $\Delta F/F$ during odor stimulation (Fig. 2a and Supplementary Fig. 2, "Methods"). It exhibits large heterogeneity across the brain, with only a small subset of voxels

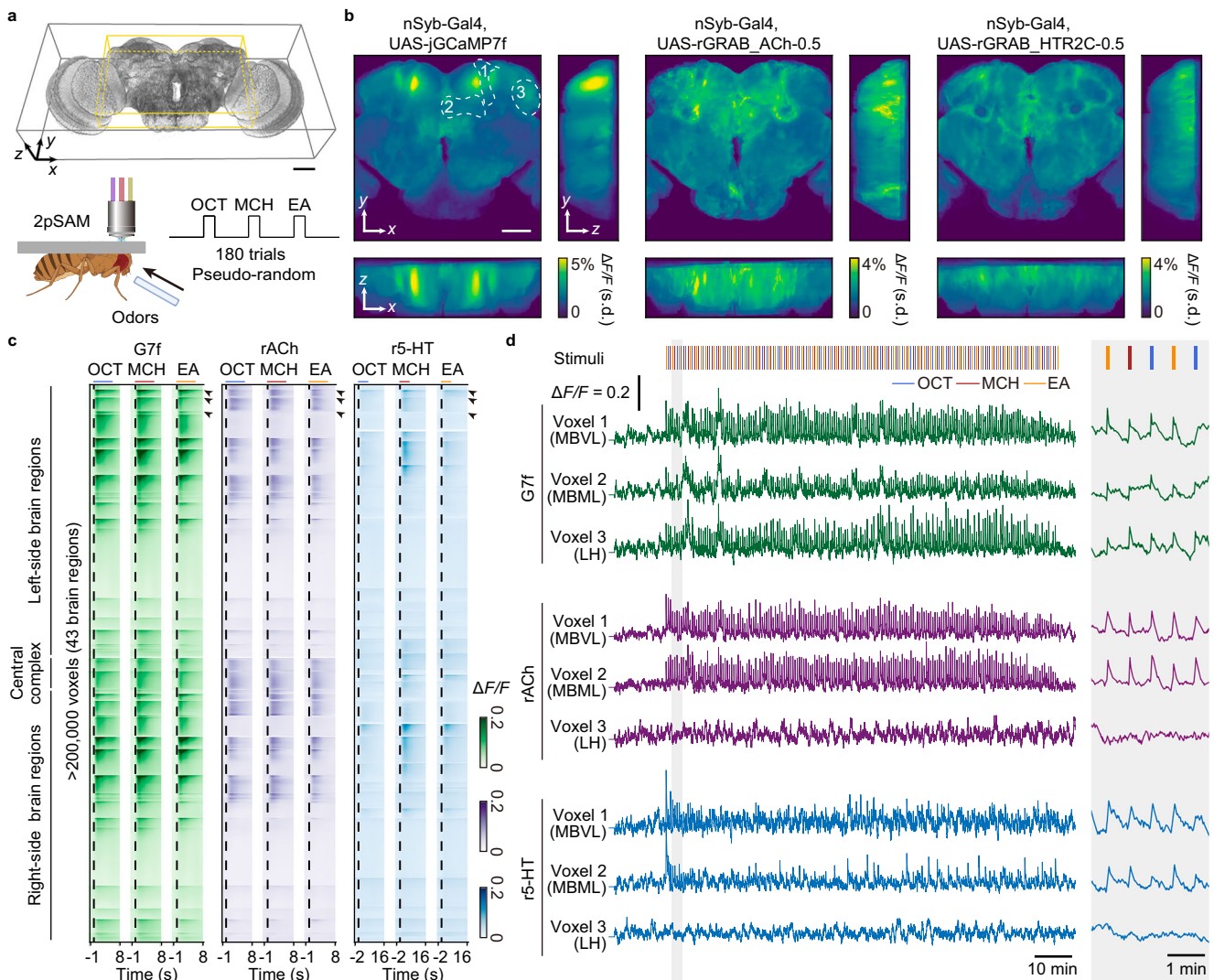

**Fig. 1 | Volumetric imaging of neuronal activities and neurochemical dynamics across multiple regions in the *Drosophila* central brain by 2pSAM across 2 h.** **a** Experimental setup and imaging FOV. Bottom: Imaging setup (left) and odor delivery paradigm (right). Created in BioRender. Fan, J. (2025) https://BioRender.com/cp0dmbd. Top: Schematic of the FOV, spanning most of the lateral and approximately half of the axial extent of the central brain, including 43 brain regions (yellow box). **b** Maps of the average standard deviation of $\Delta F/F$ during odor stimulation across trials for G7f, rACh, and r5-HT (averaged across flies, $n = 15$ for G7f, $n = 5$ for rACh, $n = 10$ for r5-HT). 10 flies co-labeled by G7f and rACh and 10 flies co-labeled by G7f and r5-HT are analyzed. Flies co-labeled by G7f and rACh with strong non-specific fluorescence on the upper edge of the brain are excluded for clarity. **c** Trial-averaged G7f, rACh, and r5-HT responses to 3 odor stimuli of every voxel in the FOV of a fly. Over 200,000 voxels across 43 brain regions are recorded at 30 Hz. The voxels are arranged in the order of brain regions and the voxels within a brain region are listed by the response intensity from high to low. Since r5-HT response is slower, its response in 16 s after odor delivery is shown, while 8 s is shown for G7f and rACh. Odor stimulus periods (5 s) are shown as lines above (blue: OCT; red: MCH; orange: EA). **d** Left: Example $\Delta F/F$ traces of the voxels in 3 brain regions in (**c**) (labeled by black arrowheads). Right: The zoomed-in traces of the labeled period (gray shade). The vertical short lines above mark odor stimulus periods (blue: OCT; red: MCH; orange: EA). Regions MBVL, MBML, and LH of the right semi-brain are labeled by dashed lines with numbers 1, 2, and 3 in (**b**). G7f and rACh responses are from a co-labeled fly, and the r5-HT response is from another fly co-labeled by G7f and r5-HT (the G7f response of this fly is not shown). Scale bars: 50 μm in (**a**) and (**b**).

demonstrating high values (Fig. 2a and Supplementary Fig. 2, "Methods"). Consequently, when assessing the average intensity of every brain region, many olfactory brain regions do not show evident higher values compared to other regions (Supplementary Figs. 1g, h and 2). At the voxel level, there is variation in response intensity even within single brain regions (Figs. 1c and 2b). Additionally, the response intensity of calcium and ACh exhibits some degree of complementation, with specific voxels displaying low calcium intensity but high ACh intensity (indicated by black arrows in Fig. 2b, c).

Additionally, we characterized odor response dynamics measuring phase delays and pulse widths (Fig. 2d, "Methods"). We refined the analyses by including only olfactory brain regions with high responsiveness (Supplementary Fig. 1e, f). For calcium signals, the MB,

especially the MBPED, exhibits faster response than other regions (Fig. 2d; phase delay for the MBPED (mean ± s.e.m): 0.79 ± 0.05 s, see the Source data for detailed statistical results). For ACh, the SIP displays shorter pulse width than the MB and CRE (Fig. 2d; pulse width for the SIP (mean ± s.e.m): 2.64 ± 0.41 s, see the Source data for detailed statistical results). As a reference, we measured the kinetics of rACh and r5-HT on HEK293T cells (Supplementary Fig. 1k, l). The on-kinetics of rACh and r5-HT are about 120 ms and 80 ms, and the off-kinetics are about 1.82 s and 1.1 s, respectively.

To further characterize distinctions among calcium, ACh, and 5-HT dynamics at the systems level, we constructed functional connectivity matrices and networks for each signal during odor stimulation (Fig. 2e) and the resting state (Supplementary Fig. 3a). The

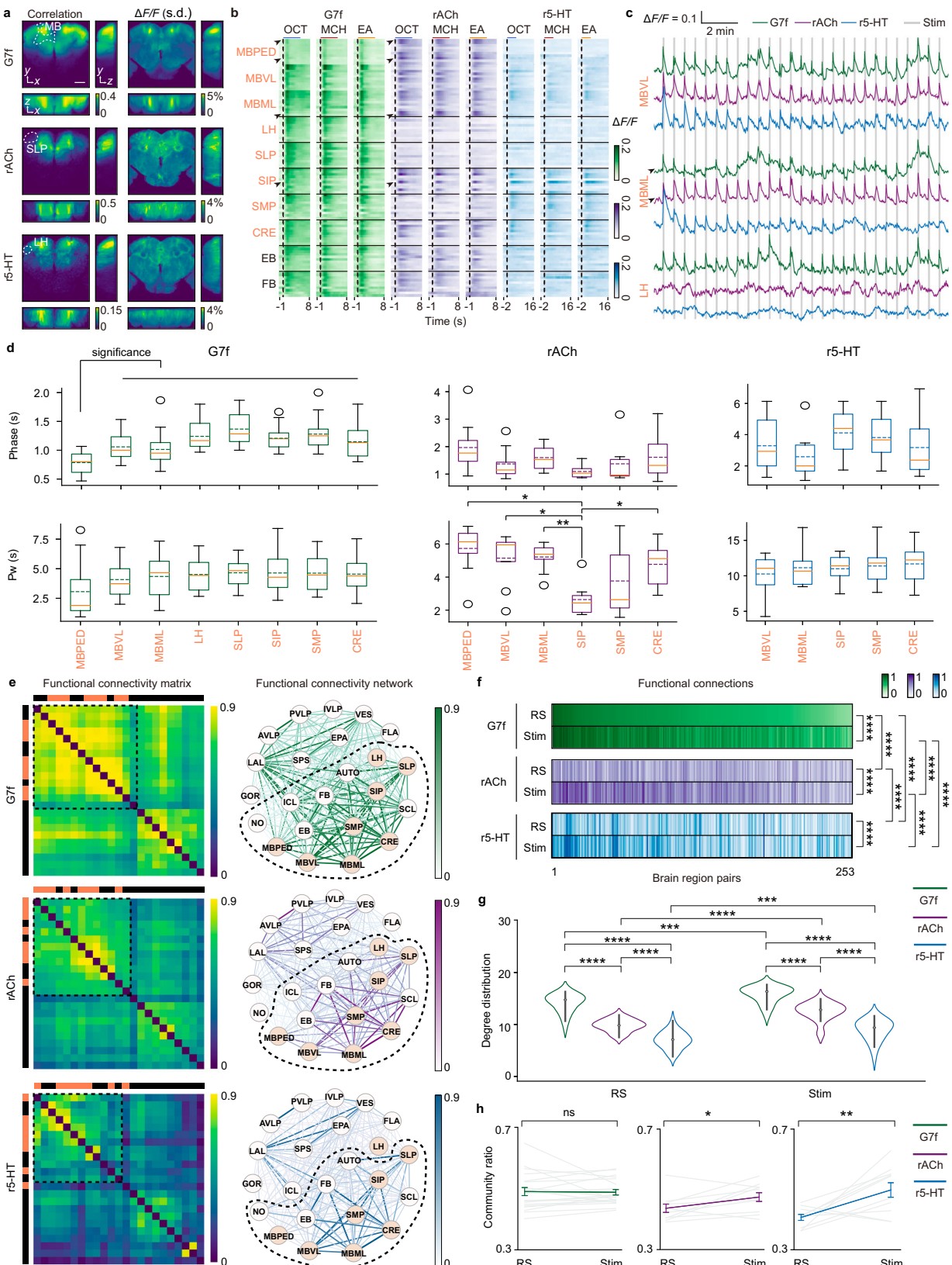

networks exhibit distinct functional connectivity patterns across dynamic signals and states (Fig. 2e, f; and Supplementary Fig. 3a). During odor stimulation, two primary communities emerge within the networks ("Methods"), with similar community divisions observed across these dynamics. Whereas, the LH is separated from other olfactory regions in the 5-HT communities (Fig. 2e). Calcium displays a

higher node degree compared to ACh and 5-HT in both states (Fig. 2g and Supplementary Fig. 3b). Additionally, the connection strength within the major olfactory-related community is higher for calcium than for ACh and 5-HT during odor stimulation (Supplementary Fig. 3c). Examining network changes between odor stimulation and the resting state, all signals exhibit altered degree distributions across

**Fig. 2 | Heterogeneous and distinct olfactory responses and functional connectivity across the brain for G7f, rACh, and r5-HT. a** Left: Maps of correlation between three dynamics and odor stimuli averaged across flies. Right: Maps of the average standard deviation of $\Delta F/F$ during odor stimulation. **b** Trial-averaged responses of sample voxels. Dashed lines mark odor onset, and short lines above mark stimulation periods. **c** $\Delta F/F$ traces. Black arrows indicate voxels with low response intensity for G7f but high for rACh in (**b**, **c**). **d** Phase delay and pulse width in olfactory brain regions with high responsiveness. Box plots: center lines, median; dashed lines, mean; box limits, upper and lower quartiles; whiskers, 1.5x inter-quartile range. Only the significance related with the MBPED is shown in the phase delay of G7f. In other panels, the absence of significance markers indicates no significance. **e** Left: Clustered functional connectivity matrices during odor stimulation averaged across flies. Color blocks mark olfactory (coral) and non-olfactory (black) regions. Right: Functional connectivity networks. Edge colors and

widths reflect connection strength, and node colors indicate region classes. Dashed boxes and circles sign the communities containing most olfactory regions. **f** Functional connection patterns. **g** Degree distribution in functional connectivity networks. **h** Community ratio, mean ± s.e.m. Each light-colored line represents a fly. 10 flies co-labeled by G7f and rACh and 10 co-labeled by G7f and r5-HT analyzed. $n = 20$ for G7f, $n = 10$ for rACh, $n = 10$ for r5-HT in (**a**, left), and (**d**–**h**). Flies co-labeled by G7f and rACh with high-intensity non-specific fluorescence are excluded in (**a**, right; $n = 15$ for G7f, $n = 5$ for rACh, $n = 10$ for r5-HT) and specific regions in (**d**). Stim: Odor stimulation. RS: The resting state. Two-sided Mann–Whitney U test in (**d**); Scheirer–Ray–Hare test in (**f**); Two-sided Kolmogorov–Smirnov test in (**g**); Two-sided Wilcoxon signed-rank test in (**h**); Benjamini/Hochberg multi-comparison correction applied in (**d**), (**f**), and (**g**). Source data are provided as a Source data file. ****$P < 0.0001$, ***$P < 0.001$, **$P < 0.01$, *$P < 0.05$, ns - not significant ($P > 0.05$). Scale bar: 50 μm.

states (Fig. 2g), with average node degrees increasing for calcium and 5-HT dynamics while remaining stable for ACh (Supplementary Fig. 3d). Additionally, the connection strength ratio of the major olfactory-related community (community ratio) for calcium dynamics remains consistent across states, indicating a stable olfactory-related community significance under both conditions (Fig. 2e, h; and Supplementary Fig. 3a). However, changes are observed in this ratio for ACh and 5-HT, suggesting connectivity alterations in response to odor stimulation (Fig. 2e, h; and Supplementary Fig. 3a).

## Odor identity representation of calcium, ACh, and 5-HT dynamics across multiple brain regions

We observed voxel-level response variations among different odors (Fig. 2b). To further investigate odor representation, we conducted odor identity classification based on calcium, ACh, and 5-HT responses. Initially, we calculated classification accuracy within blocks of $4 \times 4 \times 2$ voxels and generated accuracy maps across the FOV for each signal (Fig. 3a). These accuracy maps show different spatial distributions of odor identity representation (Fig. 3a). The average accuracy in most brain regions shows distinction among three signals (Fig. 3b). Calcium displays widespread odor coding across multiple brain regions, with high accuracy in the LH (Supplementary Fig. 1i, j). All olfactory regions show higher accuracy than the average level of non-olfactory regions for calcium (Supplementary Fig. 1i, j). However, the SLP for ACh and the MBPED, LH, and SLP for 5-HT display low decoding accuracy comparable to the average level of non-olfactory regions (Supplementary Fig. 1i, j), aligning with their low responsiveness (Fig. 2a–c; Supplementary Fig. 1e, f). In contrast, ACh shows high decoding accuracy predominantly in the MB, especially in the MBML, consistent with its strong response (Fig. 2a–c; Supplementary Fig. 1e, f), indicating the important role of ACh in the MB.

Given the observed heterogeneity in response properties within individual brain regions (Fig. 2a, b), we further integrated signals in specific brain regions for odor classification ("Methods"; and Supplementary Fig. 4a, b). We then compared the classification accuracy obtained from signal integration with the average accuracy of the blocks within each region. As a result, signal integration led to an increase in accuracy (Supplementary Fig. 5a–c), suggesting a coordinated odor identity representation within individual brain regions. Additionally, we compared the classification accuracy of two simultaneously recorded signals (calcium and ACh/5-HT) in these regions (Supplementary Fig. 5d). The accuracy of 5-HT is either lower or comparable to that of calcium, whereas ACh outperforms calcium in specific brain regions, such as the MBML and EB (Supplementary Fig. 5d).

Furthermore, we integrated signals and employed dimensionality reduction to investigate odor identity representation at the multiple-brain-region level (Fig. 3c; "Methods"; and Supplementary Fig. 4c–f). First, since functional connectivity networks displayed two primary communities (Fig. 2e), we applied a brain-region mask to select voxels

within the community containing the majority of olfactory brain regions. Next, we performed block-wise principal component analysis (PCA) to effectively reduce data dimensionality. We then consolidated principal components (PCs) from all blocks and implemented PCA again, followed by linear discriminant analysis (LDA), to map the high-dimensional activity into a stimulus-related low-dimensional space. LDA was performed on training sets, and the resultant transformation was applied to testing sets without any overlap in trials to avoid data leakage (Fig. 3d–f; Supplementary Movie 1). In the low-dimensional space, individual trial traces formed manifolds, where distinct trajectories separated trials associated with different odor stimuli, leading to clear segregation (Fig. 3d–f, top). Trial traces colored by delivery orders showed temporal properties of the manifolds. For example, in the manifolds of calcium and 5-HT, latter trials exhibited shorter extensions, indicating temporal changes in representation (Fig. 3d–f, middle; discussed in detail below). When averaging trials of the same odor identity, we observed that responses emerged from a random state around the center point, extended during odor presentation, and then gradually returned back (Fig. 3d–f, bottom). To further quantify odor identity representation, we applied a support vector machine (SVM) classifier in the LDA space. Instead of classifying trials solely based on the responses at a specific time point[24,51], we leveraged full trial trajectories, incorporating richer temporal information. Classification accuracy was evaluated using 5-fold cross-validation. Voxel-level multiple-brain-region data yields higher accuracy than region-level data ("Methods"; Supplementary Figs. 4g, h, and 5e–g) and single brain regions (Supplementary Fig. 5h–j), indicating a coordinated odor identity representation across the brain. The overall performances for calcium and ACh are comparable (calcium (mean ± s.e.m): 92.14% ± 1.33%, ACh (mean ± s.e.m): 94.83% ± 1.88%; two-sided Mann–Whitney U test: U-value = 61, Effect size (rank-biserial correlation) = −0.39, $P = 0.09$; Fig. 3g), while the accuracy for 5-HT is lower (5-HT (mean ± s.e.m): 63.11% ± 4.30%; two-sided Mann–Whitney U test: compared with G7f: U-value = 197, Effect size (rank-biserial correlation) = 0.97, $P < 0.0001$; compared with ACh: U-value = 100, Effect size (rank-biserial correlation) = 1, $P = 0.0003$; Fig. 3g), as depicted in the manifolds (Fig. 3d–f).

Given that extensive neuronal activities correlate with motion and behavior[3,52–54], it is important to assess whether movement-related signals could confound our interpretation of odor identity representation. To address this, we captured videos of fly abdomens and extracted motions (Supplementary Fig. 6a). The correlation between odor stimuli and motion energy varied a lot across individual flies, showing weak, positive, or negative associations (Supplementary Fig. 6b). To further investigate, we applied PCA to extract behavioral features (Supplementary Fig. 6c). While these features could predict stimulus periods and intervals, they were unable to classify odor identities (Supplementary Fig. 6d). We further predicted calcium, ACh, and 5-HT dynamics from behavior, stimulus, and both (Supplementary Fig. 6e, f). Both behavior and stimulus could explain partial variance of

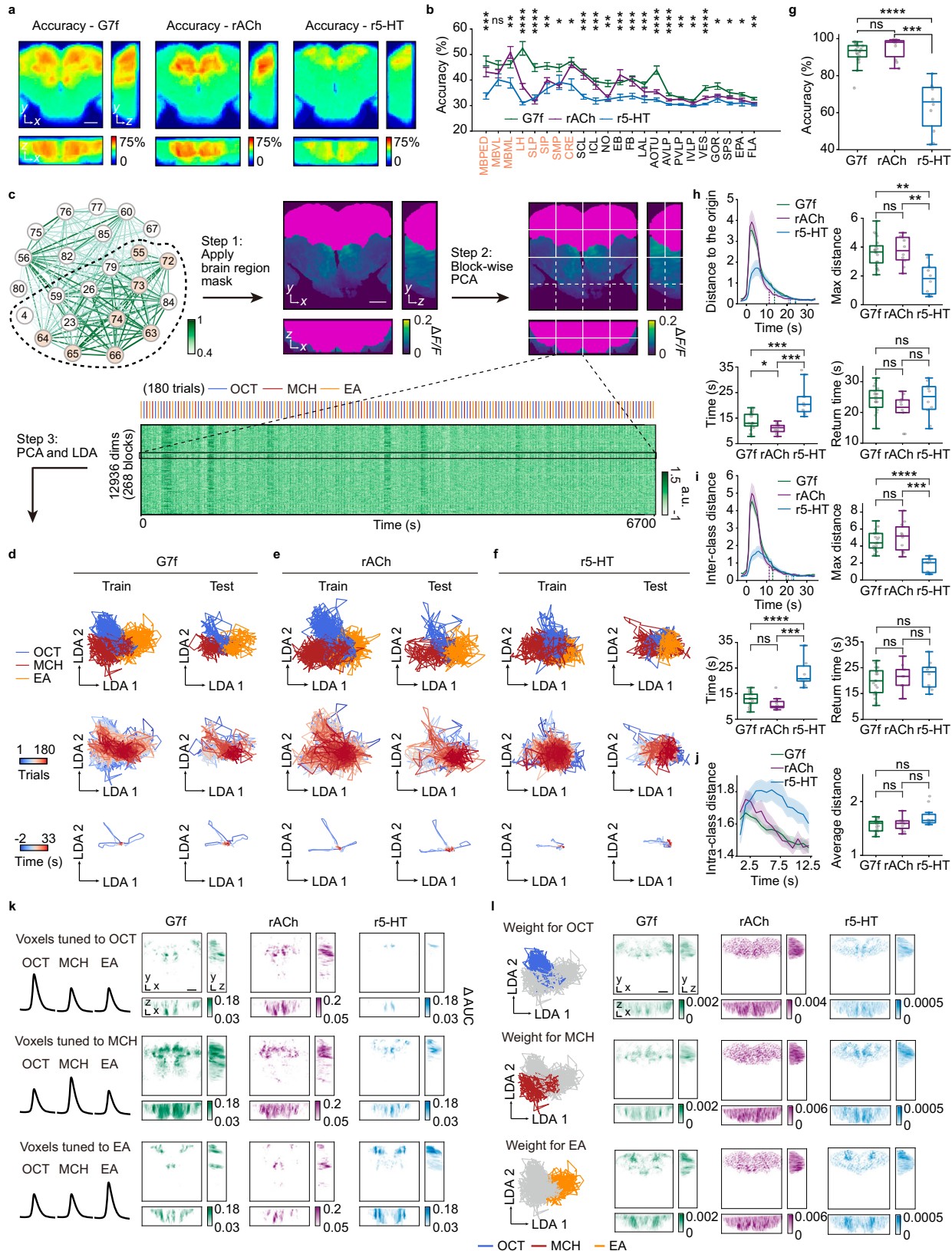

these dynamics, and behavior tended to account for more variance in ACh and 5-HT dynamics than in calcium (Supplementary Fig. 6f). However, the variance explained by behavior could not discriminate odor identities, and removing the components explained by behavior did not lower odor identity classification accuracy (Supplementary Fig. 6e–g). Therefore, we conclude that while motion modulates these

dynamics, it does not account for odor identity representation in our analysis.

We further analyzed the correlation between motion and these dynamics across multiple brain regions. Within individual flies, temporal traces show clear correlations (Supplementary Fig. 6h). To quantify these relationships, we calculated Pearson correlations at the

**Fig. 3 | Odor identity representation by G7f, rACh, and r5-HT across multiple brain regions. a** Accuracy maps averaged across flies. **b** Average accuracy in each brain region (mean ± s.e.m). **c** Schematic of the dimensionality reduction method. **d–f** Low-dimensional manifolds for G7f (**d**), rACh (**e**), and r5-HT (**f**) of a fly. Top and middle: Each line represents the response in one trial. Bottom: Each line is the average odor response of an odor identity. Top: Colors denote odor identities. Middle: Colors denote the trial order. Bottom: Colors denote the time relative to odor delivery. Manifolds in d and e are from the same fly. Arrows indicate consistent scale across dimensions (units arbitrary). **g** Voxel-level multiple-brain-region odor identity classification accuracy. $P = 0.09$ for G7f and rACh; $P = 0.0003$ for rACh and r5-HT. **h–j** Manifold metrics. **h** Top left: Distance to the origin (mean ± s.e.m). The opaque dashed lines sign the time for returning to the one-fifth of the maximum distance. The translucent dashed lines sign the time for returning to the

random state. Top right: Maximum distance. Bottom left: Time for returning to the one-fifth of the maximum distance. Bottom right: Time for returning to the random state. **i** Similar to (**h**), but for inter-class distance. **j** Left: Intra-class distance (mean ± s.e.m). Right: Statistics of intra-class distance. Box plot: center lines, median; box limits, upper and lower quartiles; whiskers, 1.5x interquartile range; each point represents a fly in (**g–j**). **k** Maps of odor tuning. **l** Similar to (**k**), but for odor identification weight. 10 flies co-labeled by G7f and rACh and 10 co-labeled by G7f and r5-HT are analyzed. $n = 20$ flies for G7f, $n = 10$ flies for rACh, $n = 10$ flies for r5-HT. Kruskal–Wallis test in (**b**); Two-sided Mann–Whitney U test and Benjamini/Hochberg multi-comparison correction in (**g–j**). Source data are provided as a Source data file. Detailed statistical results are also listed in the Source data file. $****P < 0.0001$, $***P < 0.001$, $**P < 0.01$, $*P < 0.05$, ns - not significant ($P > 0.05$). Scale bars: 50 μm.

voxel level across various time-delay conditions (Supplementary Fig. 6i), indicating distinct correlation patterns among these signals. Different brain regions exhibit varying time lags between these dynamics and motion (Supplementary Fig. 6j). Quantitative analyses show differences in both the average correlation (Supplementary Fig. 6k) and the percentage of correlation peaks (Supplementary Fig. 6l) across these dynamic signals at different time lags. Moreover, the correlation between calcium and motion is comparable to that of ACh but higher than that of 5-HT (Supplementary Fig. 6m, n). Average correlations of individual brain regions display variability across these dynamic signals (Supplementary Fig. 6o).

## Low-dimensional manifolds uncover the characteristics of odor representation

Low-dimensional manifolds offer additional insights into the properties of odor representation (Fig. 3d–f). We measured some characteristics for quantitative analyses. During odor presentation, the manifolds expand and subsequently contract, resulting in an increase and subsequent decrease in the distance to the origin (Fig. 3h, top left). Calcium and ACh dynamics show larger maximum distances compared to 5-HT, indicating greater dynamic changes in the low-dimensional space (Fig. 3h, top right). Comparing the time taken for the manifold to contract to one-fifth of its maximum distance after odor delivery, ACh returns the fastest, and calcium returns faster than 5-HT (Fig. 3h, bottom left). However, the time taken to return to the random state is similar for all signals (Fig. 3h, bottom right). We also measured inter-class (Fig. 3i) and intra-class distances (Fig. 3j) of the manifolds. As a result, 5-HT represents a lower maximum inter-class distance than calcium and ACh, suggesting a poorer ability to distinguish odor identities (Fig. 3i, top right). Calcium and ACh recover faster to one-fifth of the maximum inter-class distance than 5-HT, with no significant difference between the two (Fig. 3i, bottom left). Consistent with the previous result of the distance to the origin, the time taken to return to the random state is similar for all signals (Fig. 3i, bottom right). The average intra-class distances for three signals are comparable (Fig. 3j). Overall, calcium and ACh dynamics display similar properties in the low-dimensional manifolds and outperform 5-HT in odor identity representation.

The manifolds of three odors, OCT, MCH, and EA, also exhibit distinct characteristics. In the calcium manifolds (Supplementary Fig. 7a), EA has a shorter maximum distance to the origin and a faster return than OCT and MCH. However, such distinctions are absent in the manifolds of ACh and 5-HT. OCT and MCH (O-M, both negative) are harder to classify than other odor pairs for calcium and ACh, as the maximum inter-class distance of O-M is lower and the recovering time is shorter, especially for calcium (Supplementary Fig. 7b). No significant differences are observed in the intra-class distance across odors (Supplementary Fig. 7c). The disparate manifestation of EA and the relatively low discrimination of O-M might be relevant to odor preference (Supplementary Fig. 7d, e)[45–48], which is an aspect of odor perception better uncovered in higher-order brain regions[30,55,56] and is

likely to be reflected in our multiple-brain-region analysis. This effect is less evident for ACh and 5-HT than calcium dynamics. Two-odor classification results across multiple scales are presented in Supplementary Figs. 8, 9.

## Ensembles of odor representation in physical space

We further explored the ensembles of odor representation in physical space. First, we analyzed odor tuning across the brain, defined as the difference in the AUC of the responses to a specific odor compared to the other two odors. Voxels with significant tuning values were retained as ensembles (Fig. 3k and Supplementary Fig. 10a). Subsequently, we calculated the standardized ratio of voxels with specific tuning in different brain regions (Supplementary Fig. 10b, c), focusing exclusively on brain regions within the community that encompasses the majority of olfactory regions for each signal (Fig. 2e). The tuning exhibits a distributed pattern for all dynamic signals.

Additionally, based on our manifold and odor classification analysis, we calculated the identification weight for each odor across voxels and generated corresponding weight maps (Fig. 3l and Supplementary Fig. 10d–f). The distributions of tuning and weight exhibit clear similarities. For instance, both analyses indicate the strong role of the MBML for ACh and the LH for calcium in the identification of OCT (Supplementary Fig. 10a–f). Furthermore, all three signals suggest a strong effect of the MBVL in the identification of EA (Supplementary Fig. 10a–f).

Functional connectivity analysis also provides further insight into olfactory representation ensembles. We measured the functional connectivity of voxels with high tuning to specific odors and generated functional connectivity matrices (Supplementary Fig. 10g). Voxels responding to different odor stimuli displayed different activity traces (Supplementary Fig. 10g, right part). Correspondingly, voxels responding to the same stimulus exhibited strong functional connectivity, while the connectivity between voxel ensembles responding to different stimuli was weak, resulting in three clear functional modules in the functional connectivity matrices (Supplementary Fig. 10g, left part). This phenomenon is consistent across all three signals (Supplementary Fig. 10g), confirming that voxel ensembles for each signal are capable of odor identification.

We also discerned differences in the functional connectivity networks of voxel ensembles in physical space. We mapped the spatial distribution of the strongest functionally connected edges (those with top 10% weights) of a fly under three odor stimuli (Supplementary Fig. 10h). The functional connectivity networks for three odors exhibit differences in physical space, particularly in the location of core voxel ensembles and the spatial extent of the voxel network. This phenomenon is observed consistently across all three signals (Supplementary Fig. 10h). Statistical tests across multiple flies and signals show that for calcium and ACh, the odor MCH has a larger weighted coverage area in the functional network compared to OCT (Supplementary Fig. 10i), suggesting more long-range, large-weighted functional connectivity edges in the MCH network than in the OCT network. While for 5-HT

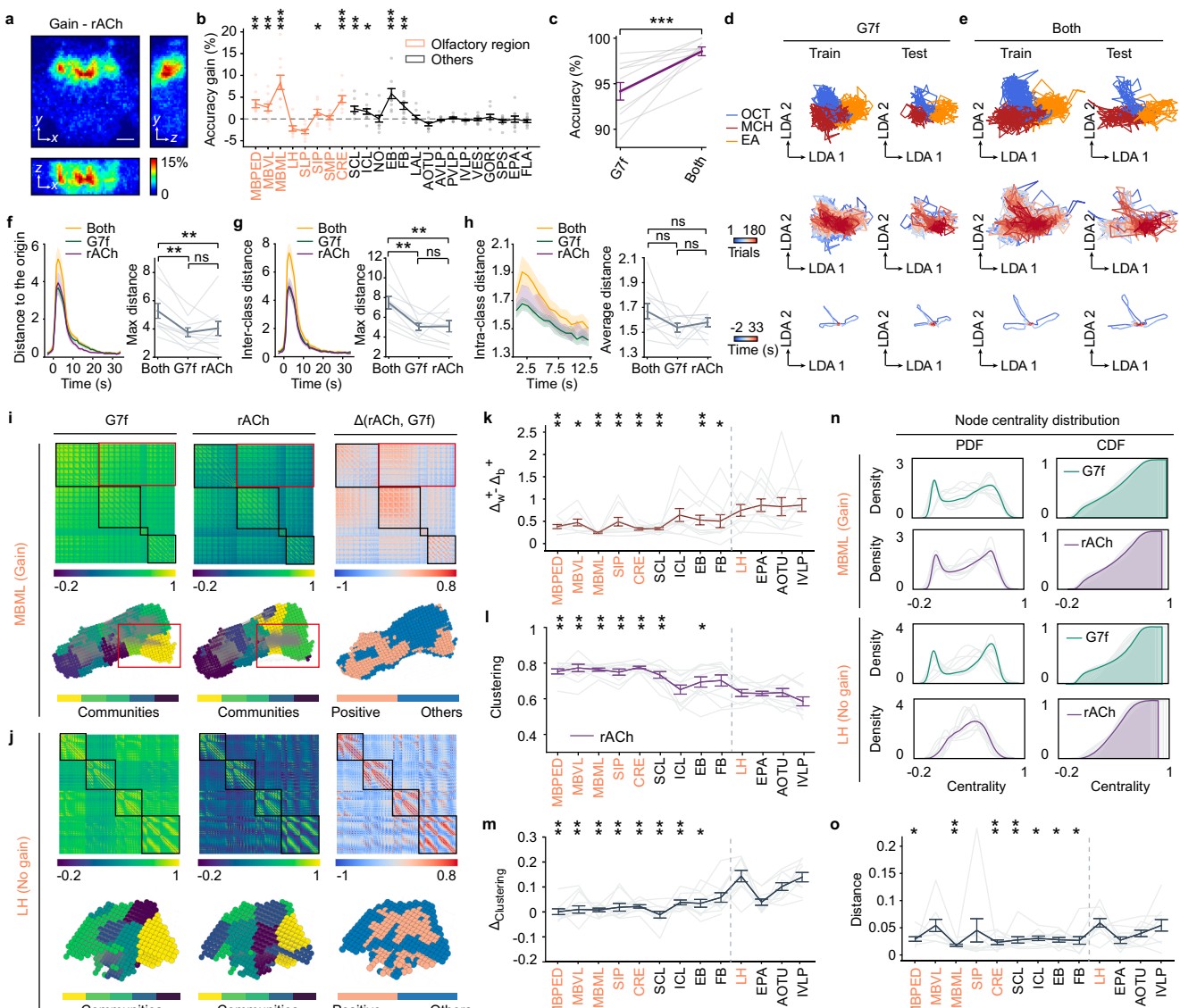

**Fig. 4 | Integration of ACh dynamics improves the odor identity representation by neuronal activity. a** Map of decoding-accuracy gain averaged across flies. **b** Average accuracy gain in each brain region. **c** Voxel-level multiple-brain-region odor identity classification accuracies. **d**, **e** Low-dimensional manifolds by G7f (**d**) and both channels (**e**) of the same fly as in Fig. 3d. Arrows are in arbitrary units but consistent across dimensions. **f**–**h** Metrics of the manifolds. **f** Distance to the origin. **g** Inter-class distance. **h** Intra-class distance. **i** Functional connectivity matrices (top) and mappings of communities and connectivity change in the physical space (bottom) of the voxels within the MBML. Three columns are the functional connectivity of G7f, rACh, and the difference in the deflation ratio between them. Connectivity complementation (top, red rectangles) and emphasis (top, black rectangles) to G7f by rACh are shown. Complementation is reflected in the different distribution of functional connections (bottom, red rectangles). Community divisions are similar for the two channels (bottom), and voxels with the greatest increased connectivity gather in space (bottom, coral). **j** Similar to (**i**), but for the LH. **k** Difference in increased deflation ratios within and between clusters of the functional connectivity of rACh compared to G7f. **l** Average clustering coefficient for rACh. **m** Average clustering coefficient difference between two channels. **n** Node centrality distribution. **o** Distance of the node centrality distributions between two channels. In (**k**–**m**), and (**o**), brain regions on the left side of the dashed line are regions with accuracy gain, and hypothesis testing is performed for them toward the average level of the LH, EPA, AOTU, and IVLP. $n = 10$ flies co-labeled by G7f and rACh, mean ± s.e.m. Each light-colored line represents a fly in (**c**), (**f**–**h**), and (**k**–**o**). One-sided Wilcoxon signed-rank test in (**b**), (**c**), (**k**–**m**), and (**o**); Two-sided Wilcoxon signed-rank test in (**f**–**h**); Benjamini/Hochberg multi-comparison correction in (**f**–**h**). Source data are provided as a Source data file. ****$P < 0.0001$, ***$P < 0.001$, **$P < 0.01$, *$P < 0.05$, ns - not significant ($P > 0.05$, not shown in **b**, **k**–**m**, **o**). Scale bar: 50 μm.

signals, EA displays a larger weighted coverage area than OCT and MCH.

## Integration of ACh and calcium dynamics improves odor identity representation

Building on our observations of distinct odor responses and representations across ACh, 5-HT, and calcium dynamics, we next asked whether integrating specific neurochemical signals with calcium signals could enhance odor identity representation. We explored these questions and came to the conclusion that integrating ACh rather than

5-HT dynamics improves odor identity representation by neuronal activity.

By integrating ACh or 5-HT signals with calcium dynamics, we could conduct odor identity classification utilizing the dual-channel data ("Methods"). Initially, we observed accuracy increase in many individual brain regions in the accuracy map when integrating ACh signals (Fig. 4a, b). This was further validated by the voxel-level classification within each region (Supplementary Fig. 5d). Subsequently, we integrated ACh signals from brain regions with accuracy gain (Fig. 4b) to the multiple-brain-region responses of calcium signals and performed voxel-level multiple-

brain-region odor identity classification. We found accuracy gain upon integrating ACh signals at the multiple-brain-region level (difference (mean ± s.e.m): 4.39% ± 0.94%; one-sided Wilcoxon signed-rank test: W-value = 55, Effect size (rank-biserial correlation) = 1, $P$ = 0.00098; Fig. 4c). This improvement is also evident in the manifolds (Fig. 4d, e; and Supplementary Movie 1), where odor responses extend farther than single signals and trials with different odor identities separate more distinctly (Fig. 4f–h). To further validate the conclusion, we analyzed odor representation and classification increasing odor number to 7 and also observed accuracy gain (Supplementary Fig. 11). For comparisons, integrating 5-HT did not yield similar accuracy gains (Supplementary Figs. 5d and 12, and Supplementary Movie 1).

Further, we examined voxel-level functional connectivity networks within specific brain regions to investigate the neural mechanisms underlying the accuracy gain. Our goal was to identify differences between regions with and without accuracy gain and determine the possible source of the observed gain. We calculated functional connectivity matrices for each signal and arranged the matrices according to the clustering in the calcium channel in several brain regions (Fig. 4i, j and Supplementary Fig. 13a). The difference matrix was also obtained (Fig. 4i, j and Supplementary Fig. 13a). In brain regions with accuracy gain, like the MBML, there are voxels with increased relative connectivity strength (deflation ratio, see Methods) both in and out of the clusters (Fig. 4i). The increased relative connectivity strength in the clusters underlies an emphasis of the connectivity structure in the ACh channel, while the increase out of the clusters underlies a connectivity complementation to the calcium channel. In contrast, regions without accuracy gain, like the LH, only display obvious emphasis without high complementation (Fig. 4j). Statistics of the difference between the mean of positive values within and between clusters in the difference matrix validate this observation (Fig. 4k). In regions with accuracy gain, the community divisions ("Methods") of the two channels exhibit similarities, with voxels demonstrating increased connectivity clustering together in space (Fig. 4i and Supplementary Fig. 13a). However, in regions without accuracy gain, although the community divisions are similar for the two channels, voxels with increased connectivity are not spatially clustered (Fig. 4j and Supplementary Fig. 13a). These phenomena are absent in the networks of the resting state (Supplementary Fig. 13b).

Brain regions with accuracy gain display stronger clustering (Fig. 4l) and more similar clustering level to that of the calcium channel (Fig. 4m). Moreover, these brain regions have higher node heterogeneity for ACh (Fig. 4n and Supplementary Fig. 13c), along with more similar node centrality distribution to calcium (Fig. 4o and Supplementary Fig. 13c). These results suggest that while the functional connectivity networks of the two signals in brain regions with accuracy gain show apparent complementation, their network properties are similar, suggesting a degree of functional segregation and potential rich information coding for both channels[57]. However, in regions without gain (e.g., the LH), ACh displays lower clustering and a more concentrated distribution of node centrality, indicating larger node homogeneity and lower functional segregation (Fig. 4l–o; and Supplementary Fig. 13c). The functional connectivity networks of 5-HT show similar characteristics (Supplementary Fig. 12). The connectivity complementation while maintaining the network characteristics shows a local mismatch of ACh release and neuronal activity and should relate to the accuracy gain.

We also analyzed odor representation and functional connectivity on raw data and compared the results with and without denoising, showing consistency of conclusions and the importance of denoising for better uncovering some characteristics (Supplementary Fig. 14).

### ACh exhibits higher temporal stability than calcium and 5-HT in odor representation

We further investigated the temporal stability of odor representation encoded by these distinct dynamic signals. We recorded consistent odor responses in 180 trials (60 sessions) of each fly. We evenly partitioned the 60 sessions into four stages (S1, S2, S3, S4) and obtained the low-dimensional manifolds for every stage (Fig. 5a). Analyzing the properties of the low-dimensional manifolds, we observe that calcium dynamics exhibit a gradual decrease in the maximum distance to the origin and inter-class distance across stages, indicating a continuous reduction in representational amplitude and odor discrimination (Fig. 5b, c). In contrast, ACh representation remains stable, at least during the first three stages, while the reduction in 5-HT representation occurs primarily between S1 and S2. Calcium exhibits a shift in the returning location of sessions from S1 to S2, suggesting a potential internal state change (Fig. 5a, d). However, this change is not observed for ACh and 5-HT manifolds. Additionally, the average intra-class distances of all signals exhibit a decrease from S1 to S2, potentially indicating a process of learning or habituation (Fig. 5e).

Overall, ACh dynamics exhibit a stable odor representation across stages, while calcium displays a continuous and gradual change. 5-HT shows reductions from S1 to S2, consistent with the fast decrease in response happening only in several trials at the very beginning (Figs. 1d and 2c). We further characterized the functional connectivity networks of the three signals, assessing functional connection patterns, node degrees, and community ratio across the four stages (Fig. 5f–j). The results corroborate our findings from the analysis of the low-dimensional manifolds. All these results indicate the stable odor representation of ACh.

### Odor representation in mushroom body neurons

To further validate our findings, we recorded odor responses of MB-specifically labeled flies (Supplementary Fig. 15a, b) and analyzed odor representation in these MB cholinergic neurons. First, we calculated accuracy maps for calcium, ACh, and their integration, assessing the accuracy gain from the integration compared to calcium alone (Supplementary Fig. 15c, d). The MBPED and MBML exhibited accuracy gains. Furthermore, we generated manifolds of olfactory representation within the mushroom body (Supplementary Fig. 15e, f). As expected, the separation of odors was less pronounced compared to the multiple-brain-region results. Nonetheless, integrating the dynamics of calcium and ACh yielded higher odor classification accuracy (Supplementary Fig. 15g) and better separation of odors in the manifolds (Supplementary Fig. 15h, i). We also analyzed the temporal changes of olfactory representation across the 2-h period but observed no obvious alteration (Supplementary Fig. 15j, k).

As a result, the compensatory role of ACh dynamics in olfactory representation to calcium dynamics remains evident in the mushroom body. However, the representation of both channels within the mushroom body appears stable over the 2-h period, suggesting that the temporal changes observed in calcium manifolds may originate from other brain regions.

## Discussion

High-throughput neuronal recording techniques have brought many discoveries either in the spatial or the temporal domain[1–10], inaccessible for localized studies. Understanding the spatiotemporal dynamics and information representation of neuronal activity serves as a fundamental step for a deeper and more comprehensive understanding of functions. In contrast, large-scale neuromodulator and neurotransmitter dynamics still lack studying despite their vital role in the neural system, due to technical limitations[13]. Therefore, abundant mysteries might be hidden in these dynamics as their quantitative relationship with neuronal activity in vivo remains elusive[13,17,18,20]. Our current understanding of their functions, primarily derived from targeted investigations, might be limited[12]. Moreover, information is traditionally considered coded in neuronal activity and can be decoded with high precision[58]. Neuromodulators and neurotransmitters have been primarily understood as neuronal activity regulators rather

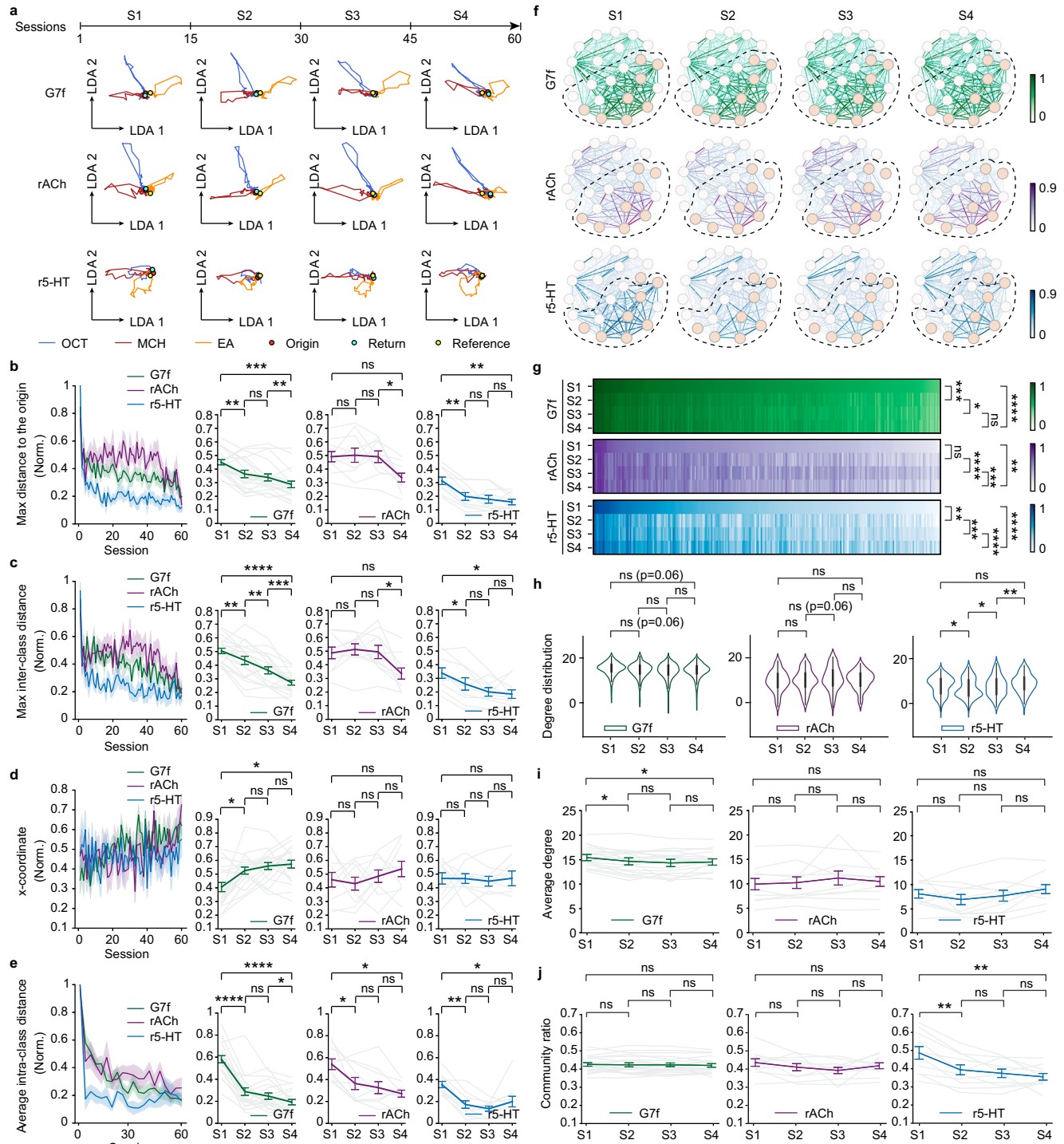

**Fig. 5 | ACh signals exhibit greater temporal stability in odor representation and the functional connectivity network than calcium and 5-HT. a** Average manifolds of a fly across 4 stages. Each session includes 3 trials. Arrow lengths are in arbitrary units but consistent across dimensions. Colors denote odor identities (blue: OCT; red: MCH; orange: EA). The red, cyan, and yellow dots denote the origins, return locations, and reference center points of the manifolds, respectively. **b**–**e** Metrics of the manifold change across four stages. **b** Maximum distance to the origin. **c** Maximum inter-class distance. **d** The x-coordinate of the return locations. **e** Average intra-class distance. Left: Changes of the metrics across sessions. Right: Statistics of the metrics across four stages for each indicator. **f** Functional connectivity networks of the brain regions for each indicator during odor stimulation in four stages. Consistent with Fig. 2e, the colors and widths of the edges indicate connection strengths, and the colors of the nodes indicate olfactory (coral) and non-olfactory (blank) regions. The dashed circles mark communities in Fig. 2e.

**g** The functional connection patterns of three indicators among four stages. **h** Statistics of the degree distribution in functional connectivity networks of three indicators among four stages. **i** Statistics of the average degree of the functional connectivity networks among four stages. **j** Statistics of the community ratio in the functional connectivity networks of three indicators among four stages. 10 flies co-labeled by G7f and rACh and 10 flies co-labeled by G7f and r5-HT are analyzed. $n = 20$ flies for G7f, $n = 10$ flies for rACh, $n = 10$ flies for r5-HT, mean ± s.e.m. Each light-colored line represents a fly in (**b**–**e**), (**i**), (**j**). Two-sided Wilcoxon signed-rank test in (**b**–**e**), (**i**), (**j**); Scheirer–Ray–Hare test (non-parametric two-way ANOVA test) in (**g**); Two-sided Kolmogorov–Smirnov test in (**h**); Benjamini/Hochberg multi-comparison correction applied in (**b**–**e**), (**g**–**j**). Source data are provided as a Source data file. Detailed statistical results are also listed in the Source data file. ****$P < 0.0001$, ***$P < 0.001$, **$P < 0.01$, *$P < 0.05$, ns - not significant ($P > 0.05$).

than direct information carriers. However, the recent evidence of dissociable dopamine firing and release and sensory-related fast cholinergic activities prompts a reevaluation of neuronal and neurochemical information representation[20,24,33,59].

The latest advancement in high-performance fluorescent neurochemical indicators[38,39,60] and microscopic techniques[36,40–42,61] has enabled us to investigate the spatiotemporal neuronal activities and neurochemical dynamics and explore their involvement in information representation. To address these objectives, we focused on olfactory processing as a context. Our imaging and analyses identified the complementation between ACh and calcium signals in information representation and functional connectivity networks, as well as the temporal stability of ACh representation. Our findings extend previous studies by providing an extensive description of olfactory representation across such a large temporal and spatial scale encompassing both neuronal activities and neurochemical dynamics, which may complement existing frameworks of neural information representation. The view and findings can inspire further studies of the underlying mechanisms.

This observational study has some limitations. First, approximately 43 brain regions are recorded by imaging, and there are some other important brain regions, like the antennal lobe, are not covered due to the FOV restraint. Also, since we used pan-neuronally labeled flies to perform a dynamic signal screen in this study, we could not resolve single neurons or distinguish neuron types. Further investigations of genetically targeted circuits and neuron types combined with targeted manipulations are necessary to further elucidate the mechanisms underlying our findings and the causality between different dynamics. Large-scale sparse labeling can also greatly assist in this endeavor[52,62]. Additionally, the extensive-surveying methodology employed in our study may hold promise for uncovering new findings in other areas, such as learning and memory, which have widespread effects across the brain. Moreover, the simultaneous volumetric recording of multiple indicators on a large scale also presents valuable opportunities to investigate the interactions between different neuromodulators[63,64] and the coupling of other important processes[65].

# Methods
## Fly stocks
Flies with pan-neuronal expression of jGCaMP7f and rGRAB_ACh-0.5 were of the genotype: w; UAS-rGRAB_ACh-0.5/+; nSyb-Gal4/UAS-jGCaMP7f. Flies with expression of jGCaMP7f and rGRAB_ACh-0.5 in the mushroom body were of the genotype: w; UAS-rGRAB_ACh-0.5/+; OK107-Gal4/UAS-jGCaMP7f. Flies with pan-neuronal expression of jGCaMP7f and rGRAB_HTR2C-0.5 were of the genotype: w; UAS-rGRAB_HTR2C-0.5/+; nSyb-Gal4/UAS-jGCaMP7f. Wild type flies used in Supplementary Fig. 7 were w1118 (isoCJ1). UAS-jGCaMP7f/+ was crossed by UAS-jGCaMP7f and w1118 (isoCJ1). UAS-rGRAB_ACh-0.5 and UAS-rGRAB_HTR2C-0.5 were from Yulong Li's Lab at Peking University. nSyb-Gal4 (BDSC: 51941), OK107-Gal4, UAS-jGCaMP7f[37], and w1118 (isoCJ1) were from Yi Zhong's Lab at Tsinghua University. Flies were raised on standard cornmeal medium with a 12-h light/12-h dark cycle at 23 °C and 60% humidity and housed in mixed male/female vials.

## Odor delivery
3-octanol (OCT; CAS# 589-98-0, Sigma-Aldrich), 4-methylcyclohexanol (MCH; CAS# 589-91-3, Sigma-Aldrich), ethyl acetate (EA; CAS# 141-78-6, ThermoFisher Scientific), 1-Octen-3-ol (1-OCT; CAS# 3391-86-4, Sigma-Aldrich), Benzaldehyde (BEN; CAS# 100-52-7, Sigma-Aldrich), Isopentyl acetate (IA; CAS# 123-92-2, Alfa Aesar), and Methyl salicylate (MS; CAS# 119-36-8, Sigma-Aldrich) diluted 1.5:1000, 1:1000, 1:1000, 1:500, 1:500, 1:1000, and 1:1000 in mineral oil were used as odors[66]. Odors were delivered for 5 s with 30 s inter-stimuli intervals across 180 trials (60 sessions) in a pseudo-random order avoiding consecutive presentations of the same odor. We used a custom-made odor delivery apparatus (supplementary Fig. 16)[31,55,66,67]. Air with the same airflow was delivered to flies during intervals.

## Control experiments for assessing neural responses to brief air-puffs
Control experiments were conducted to assess the effect of the brief air-puffs at the start of odor stimuli. Air through mineral oil was delivered as control. 9 trials of OCT, MCH, EA, and the control were delivered to flies in a pseudo-random order. 3 flies co-labeled with G7f and rACh were tested and their responses to the four stimuli were compared (Supplementary Fig. 1a–d).

## Odor avoidance test
Two- to five-day-old flies were used for the odor avoidance tests (Supplementary Fig. 7d, e). Flies were placed at the choice point of a T maze, where they were given 1 min to choose between the odor (OCT/MCH/EA) and air. We then counted the flies in each arm of the maze and calculated a performance index (PI). An index of 0 corresponded to a 50:50 distribution between the odor and air, while an index of 100% corresponded to complete avoidance of the odor. Additionally, a negative PI indicated a preference for the odor over air. The diluted odor was used in the same way as in the previous experiments. Two groups of the same stock were tested successively, and the side of the test tube with odor was alternated.

## Kinetics of rACh and r5-HT
Kinetics of the two indicators were measured on HEK293T cells. The confocal high-speed line scanning mode (1024 Hz) was used to record the fluorescence signal when the cells were locally puffed with drugs via a glass pipette positioned in close proximity to cells. To measure $\tau_{on}$, 10 μM ACh and 10 μM 5-HT was puffed on the cells expressing rACh and r5-HT, respectively; to measure $\tau_{off}$, 10 μM scopolamine was puffed on cells bathed in 100 μM ACh, and 100 μM SB242084 was puffed on cells bathed in 10 μM 5-HT.

## Fly preparation for functional imaging
Three- to eight-day-old female flies were selected for brain imaging. To prepare for imaging, flies were anesthetized on ice and mounted in a 3D-printed plastic disk that allowed free movement of the legs[66,68]. The posterior head cuticle was opened using sharp forceps (5SF, Dumont) at room temperature in fresh saline (103 mM NaCl, 3 mM KCl, 5 mM TES, 1.5 mM CaCl₂, 4 mM MgCl₂, 26 mM NaHCO₃, 1 mM NaH₂PO₄, 8 mM trehalose, and 10 mM glucose (pH 7.2), bubbled with 95% O₂ and 5% CO₂)[66,68]. After that, the fat body and air sac were also removed carefully. The position and angle of the flies were adjusted to keep the posterior of the head horizontal, and the window was made big and clean, for the convenience of multiple-brain-region observation. Brain movement was minimized by adding UV glue around the proboscis[54,69]. After preparation, flies were placed under the objective for two-photon imaging.

## Multiple-brain-region two-photon volumetric imaging by 2pSAM
We used a 25×/1.05 NA water immersion objective (Olympus) in this experiment. Min-NA 2pSAM was adopted to achieve an effective depth of focus of 100 μm[36], covering about half of the axial range of the fly brain (-80– 180 μm under the surface). The field of view was 458.7 μm × 458.7 μm (512 pixels × 512 pixels), covering the whole lateral range of the central brain and some part of the optic lobe. An excitation wavelength of 1035 nm was used for two-color imaging. The power of the excitation light was set at 25–35 mW through the ~2 h recording. For detection, a 525-nm filter (MF525-39, Thorlabs) was used for the green channel, and a 610-nm filter (ET610/75 m, Chroma) was used for the red channel. The acquiring rate was 30 Hz, and a 13-angle scanning was adopted.

## Image processing pipeline

**Preprocessing—registration and denoising**. We conducted pre-processing steps for the images of each angle. First, we performed motion correction using a piecewise rigid registration algorithm[36,70]. Then, we applied denoising algorithms to enhance the signal-to-noise ratio, which was essential for our voxel-based analyses. We utilized DeepCAD-RT in most of our analyses due to its superior capability in detecting subtle signal changes[40,41]. To improve the recovery of temporal characteristics in manifold analyses, we implemented a new algorithm SRDTrans[42]. We trained customized denoising models for each channel of each fly.

**Volume reconstruction**. We utilized VCD-Net for reconstruction to accelerate the process[71]. First, we sampled about 2% of the frames for reconstruction[36]. The reconstructed volumes together with the images of each angle served as training sets. Subsequently, all the volumes of an experiment were efficiently reconstructed using VCD-Net within a few hours. The reconstructed volumes were 256 pixels × 256 pixels × 25 pixels, with a lateral resolution of 1.79 μm and an axial resolution of 4 μm. This resolution was decided to strike a balance between achieving high resolution and managing the total data size for subsequent analyses. The temporal rate of volumes was 30 Hz and 2.3 Hz with and without the sliding-window reconstruction method, respectively.

**Alignment to atlas**. The *Drosophila* brain atlas was sourced from Virtual Fly Brain, the version used in a previous work[3]. The alignment was performed using landmarks with ImageJ (https://imagej.net/plugins/name-landmarks-and-register)[54]. We used the red channel for registration, as the structure of this channel was more distinct. An eroded atlas, in which the mask of each region was eroded, was used to extract brain regions, preventing incorrect assignment of edge voxels to regions. While completing volume registration in the previous steps, we only aligned the first volume to the atlas, thus aligning the entire stack.

**Temporal trace extraction**. We extracted the temporal traces of each voxel. For the computation of $\Delta F/F$, a sliding window was used. For each frame, the mean of the lower 30% intensity of the previous 200 frames (at 2.3 Hz) was taken as $F_0$, and $\Delta F/F$ was computed as $(F-F_0)/F_0$, in which $F$ was the current intensity of the voxel. The $\Delta F/F$ of each brain region was taken as the average of the $\Delta F/F$ of the voxels within the region. Since the odor responses exhibited persistence beyond the duration of odor delivery, we extended the time window for extracting these responses. Specifically, the selected time windows encompassed the period from the onset of the response to its decay to less than half of the peak magnitude. In most of our analyses, as the dynamics of r5-HT were slower, the time windows were approximately 8 s, 8 s, and 16 s (20 frames at 2.3 Hz, 2.3 Hz, and 1.15 Hz) for G7f, rACh, and r5-HT, respectively. In the case of odor identity classification and manifold analyses, the time windows were set to approximately 12 s (15 frames at 1.15 Hz) and 33 s (40 frames at 1.15 Hz) after odor delivery for all signals.

## The map of responsiveness

For each fly, the Pearson correlation between $\Delta F/F$ concatenated across 180 trials and stimulus (binary sequence) was calculated for each voxel in each channel to obtain the maps of responsiveness (Fig. 2a). Only significant correlations ($P < 0.05$, using the Python package scipy.stats.pearsonr) were recorded. The maps of individual flies were aligned and averaged to generate a summary map. The average of the top 20% correlation values for each region was calculated to make comparisons (Supplementary Fig. 1f).

## The map of response intensity

For each trial, the standard deviation of $\Delta F/F$ in 8 s (G7f and rACh) and 16 s (r5-HT) since odor delivery was calculated as the response intensity. To create a map, we calculated the average response intensity for each voxel across trials. The maps of individual flies were aligned and averaged to generate a summary map, as shown in Fig. 2a. The response intensity of each region was the average of the voxels within the region (Supplementary Fig. 1h). Other metrics usually used for response intensity, like the AUC, yielded similar results (Supplementary Fig. 2).

## Phase delay and pulse width

For each fly, phase delays and pulse widths were calculated for brain regions in each channel using the trial-averaged $\Delta F/F$ (Fig. 2d). We refined the analyses by including only the olfactory brain regions demonstrating high responsiveness (Supplementary Fig. 1e, f). The phase delay was the time-lapse from the start to the peak of the response. The pulse width was the full width at half maximum.

## Brain-region-level functional connectivity network analysis

Brain-region-level functional connectivity networks of the left-side brain regions and the central complex recorded (23 brain regions) were analyzed.

**Construction of functional connectivity networks**. To construct brain-region-level functional connectivity networks during odor stimulation, we first spliced the response records for each voxel over the time window of 180 trials. We then averaged all voxels within brain regions to obtain the average response record for each brain region and calculated the Pearson correlation of the averaged response records between each pair of brain regions. The above process was performed for each signal, such that the correlation matrices shaped in 23 × 23 dimensions were generated for G7f, rACh, and r5-HT, respectively. Each correlation matrix was used to construct a weighted undirected network (Fig. 2e), where the nodes represented brain regions, and the edges represented the correlation values between brain regions. We used the Louvain and greedy algorithms to detect the community structure in the functional connectivity networks and obtained consistent results. The detected communities formed by tightly connected brain regions were labeled with dashed circles (Fig. 2e). The functional connectivity networks during the resting state were constructed using the dynamic traces with a temporal length around 10 min. before the start of odor stimulation (Supplementary Fig. 3a).

**Comparison of functional connectivity patterns in physical space**. We examined changes in the distribution of functional connectivity in physical space using the non-parametric multifactor ANOVA test called the Scheirer–Ray–Hare test. Specifically, we arranged the functional connections of each fly during odor stimulation and the resting state in the same order and grouped data from multiple flies, thus testing whether odor stimulation was a factor that significantly affected the brain-region-level functional connectivity (Fig. 2f). In the same way, sorting and grouping data from G7f, rACh, and r5-HT allowed testing whether different signals exhibited significantly distinct functional connectivity patterns. Figure 2f displayed the functional connectivity between each pair of brain regions averaged across multiple flies, sorted by the connection strength of G7f in the resting state.

**Comparison of functional connectivity in network topology**. We analyzed distinctions in the topological characteristics of the brain-region-level functional connectivity networks across different signals and states. The degree of a node was calculated as the total connection strength of the edges connected, and the average degree of a network represented the average degree across all nodes. To assess the differences in the distribution of node degrees (Fig. 2g) and the average degree values (Supplementary Fig. 3b, d) across different signals and states, we used the Kolmogorov–Smirnov test and the Wilcoxon signed-rank test, respectively. Additionally, we calculated the

community ratio, which was defined as the connection strength ratio of the community containing most olfactory regions, to reflect the significance of the community (Fig. 2h and Supplementary Fig. 3c). A larger community ratio indicated a more influential role of the given community in the network.

## Accuracy map

The volume was divided into small blocks of 4 × 4 × 2 voxels. The data form within each block was trials × [voxels × frames × channels]. We first performed PCA (sklearn.decomposition.PCA) and kept the dimensions explaining 90% variance. Then, SVM (sklearn.svm) was applied to classify the trials according to odor identities, and the accuracy was obtained by 5-fold cross-validation. The maps of individual flies were aligned and averaged to generate a summary map (Fig. 3a and Supplementary Fig. 1i). The accuracy of each region was the average of the voxels within the region (Fig. 3b and Supplementary Fig. 1j). The map revealing the accuracy gain was generated by subtracting the accuracy map of G7f from the accuracy map of the dual-channel data (Fig. 4a and Supplementary Fig. 12a). We employed PCA and SVM on frames 1-14 (for odor response was high during this period) from the start of odor delivery.

## Voxel-level odor identity classification

**Odor identity classification in each brain region.** The data within each region was organized in a [trials × frames] × [voxels × channels] format. To reduce the dimensionality, we initially used PCA. Next, we employed LDA (sklearn.discriminant_analysis.LinearDiscriminantAnalysis.LDA) to identify a stable low-dimensional space where odor identity was well-represented. The data was then reshaped as trials × [frames × LDA dimensions]. SVM was utilized for trial classification, and the accuracy was determined by 5-fold cross-validation. It was worth noting that the classification accuracy varied based on the variance threshold used in the PCA step (Supplementary Fig. 4a, b). A threshold of 0.8 exhibited optimal performances for almost all regions and signals and was selected. The threshold was not sensitive and could be adjusted. We employed PCA and SVM on frames 1-14, and LDA on frame 3 from the start of odor delivery (for odor response was high on frames 1-14 and was almost the highest on frame 3). The parameters were not sensitive within a range. For odor identity representation analysis of the mushroom body cholinergic neurons, the MB was analyzed as a single brain region using the method above.

**Odor identity classification across multiple brain regions.** First, for single-channel classification, a brain-region mask was applied to selectively choose the voxels within the community containing most of the olfactory regions. For dual-channel classification, the brain-region mask for the rACh or r5-HT channel was the brain regions with accuracy gain, either in the average accuracy or the accuracy of the voxel-level classification (Fig. 4b; and Supplementary Figs. 5d and 12b). Then, to handle the higher dimensionality of multiple-brain-region data compared to single-region data, we implemented an additional block-wise PCA step prior to the steps of odor identity classification in each brain region. The volume was divided into 10 × 10 × 10 blocks. Within each block, PCA was carried out while retaining dimensions accounting for 90% of the variance. When dealing with dual-channel data, this approach was independently applied to each channel, and the resulting outputs were consolidated for the subsequent steps. We observed that the classification accuracy varied with the number of dimensions retained after the second PCA step (Supplementary Fig. 4c–f). The accuracy became stable after surpassing a certain threshold of dimensions. Hence, we set the threshold at 25 for optimal results. We employed PCA and SVM on frames 1-14, and LDA on frame 3 from the start of odor delivery. We utilized the two denoising algorithms, SRDTrans (Supplementary Fig. 4c, d) and DeepCAD-RT

(Supplementary Fig. 4e, f), and obtained similar results and robust accuracy gain. The results of SRDTrans were analyzed.

## Brain-region-level odor identity classification

The average Δ*F*/*F* of single brain regions were consolidated for classification. The steps were similar to odor identity classification in each brain region. The variance threshold of the PCA step was set to 0.998 for optimal performance (Supplementary Fig. 4g, h).

## Manifolds

For odor identity classification across multiple brain regions, we applied the PCA and LDA transformations to all frames to capture the low-dimensional dynamics in the LDA space. This approach allowed us to observe that the responses of individual trials within this low-dimensional space manifested as curved traces, which collectively formed a manifold. We obtained the manifold from 2 s before the onset of odor stimuli to 33 s thereafter to characterize the whole process of odor responses, starting from and returning to a random state. Each trace started from the common state and extended in different directions, symbolizing the essence of various odor identities. To visualize the process of extending and returning clearly, we averaged the traces with the same odor identity and colored the average trace with the time-lapse relative to odor delivery (Figs. 3d–f, 4d, e; and Supplementary Fig. 12d, e).

## Manifold analysis

We aligned and combined the manifolds of the testing sets in each fold and examined the characteristics of the combined manifold. The average traces of each odor identity were used to measure the distance to the origin and the inter-class distance. The distance to the origin referred to the distance between the locations of a specific time point and the average location of the start time point (2 s before the onset of odor stimuli) in the LDA space. The inter-class distance referred to the distance of odor pairs (O-M: OCT and MCH, O-E: OCT and EA, M-E: MCH and EA) at each time point. The intra-class distance referred to the average distance between the traces with the same odor identity at each time point. The return time was evaluated as the time-lapse for distance to the origin or intra-class distance to recover to the random level (the average value of 29–30 s) from the start of odor stimuli. The average intra-class distance was calculated within 12 s after odor delivery. The values of each odor were compared to analyze the representational distinctions among odor identities (Supplementary Fig. 7a–c). The average value of all odors was evaluated to analyze the representational distinctions among indicators (Figs. 3 and 4; and Supplementary Fig. 12).

## Functional connectivity network analysis within brain regions
**Construction of functional connectivity networks within brain regions.** To construct a functional connectivity network within each brain region, we first spliced the response records for each voxel of a given brain region over the time window of 180 trials. We then calculated the Pearson correlation of the response records between each pair of voxels and generated an N × N correlation matrix, N referring to the number of voxels in a given brain region. Thus, a G7f matrix and a rACh matrix were obtained for each brain region of each fly co-labeled by G7f and rACh (Fig. 4i, j); a G7f matrix and a r5-HT matrix were obtained for each brain region of each fly co-labeled by G7f and r5-HT (Supplementary Fig. 12i, j). We used all the N voxels of a given brain region as nodes and retained the top 30% correlations in the matrix as edges, thus obtaining a weighted undirected network for each region. We displayed the constructed networks of each brain region according to the relative position of each voxel in physical space, where the edges were shown in gray, and the nodes were colored by the community division (using the Louvain algorithm) of the network (Fig. 4i, j; Supplementary Figs. 12i, j, and 13a, b). Seven brain regions were selected from different neuropils of the fly brain (Supplementary Table 1) to

examine their functional correlation matrices and networks during odor stimulation (Supplementary Fig. 13a) and the resting state (Supplementary Fig. 13b) for rACh and G7f.

**Comparison of functional connectivity in physical space.** In order to compare the neurochemical (rACh or r5-HT) and neuronal (G7f) functional connectivity, we calculated the deflation ratio as $(M - \overline{M})/\overline{M}$, where $M$ and $\overline{M}$ denote the correlation matrix of G7f (or rACh, r5-HT) and its average value. The difference matrix $\Delta$ was defined as the difference between the deflation ratio of G7f and rACh (or r5-HT), which was calculated as:

$$\Delta = \frac{M_N - \overline{M_N}}{\overline{M_N}} - \frac{M_C - \overline{M_C}}{\overline{M_C}} \qquad (1)$$

Where $M_C$ and $\overline{M_C}$ denote the correlation matrix of G7f and its average value, $M_N$ and $\overline{M_N}$ denote the correlation matrix of rACh or r5-HT and its average value (Fig. 4i, j; Supplementary Figs. 12i, j, and 13a, b). The neurochemical and difference matrices were sorted according to the clustering of the G7f matrix, so that the black boxes marked the most prominent functional connections in the G7f matrix, while the outside of the black boxes represented the correlations between different clusters (Fig. 4i, j). The red boxes in Fig. 4i signed the functional connections highlighted by the rACh matrix that were not emphasized in the G7f matrix, suggesting the connectivity complementation. We showed the nodes involved in around top 1% correlations of the difference matrix $\Delta$ in coral color and the remaining nodes in blue in the networks (Fig. 4i, j; Supplementary Figs. 12i, j, and 13a, b). This reflected whether the connections emphasized by ACh/5-HT gathered together in physical space. We designed a metric $\Delta_w^+ - \Delta_b^+$ to measure connectivity complementation. The metric $\Delta_w^+ - \Delta_b^+$ was defined as the difference between the mean of positive values within and between clusters in the difference matrix $\Delta$. The results showed that this metric was always greater than 0. The smaller the metric $\Delta_w^+ - \Delta_b^+$, the more relative connectivity complementation there was (Fig. 4k and Supplementary Fig. 12k). All nine brain regions with accuracy gain and four without accuracy gain were compared for rACh. The Wilcoxon signed-rank test was used to detect whether the metric $\Delta_w^+ - \Delta_b^+$ of each brain region with accuracy gain was significantly less than the average level of the selected brain regions without accuracy gain, i.e., LH, EPA, AOTU, and IVLP. Although for r5-HT, only the MBVL exhibited accuracy gain (Supplementary Fig. 12b), we performed the same test as rACh.

**Comparison of functional connectivity in network topology.** In order to compare the topological characteristics between neuronal and neurochemical functional connectivity networks, we measured the average clustering coefficient and the node centrality (degree centrality) distribution. The clustering coefficient $C$ was a measure of the degree to which nodes clustered together. In a network with node $v_i$, the local clustering coefficient $C_{v_i}$ was calculated as $C_{v_i} = 2 * E(v_i)/k(v_i) * [k(v_i) - 1]$, where $E(v_i)$ was the number of edges between the neighbors of node $v_i$, and $k(v_i)$ was the degree of node $v_i$, i.e., the number of edges connected to it. The global clustering coefficient $\langle C \rangle$ represented the average of the local clustering coefficient of all the nodes. We calculated the global clustering coefficient as $\langle C \rangle = \frac{1}{N}\sum_{i=1}^{N} C_{v_i}$, $N$ referring to the number of nodes.

We calculated the average clustering coefficients of the functional connectivity networks for different signals in brain regions with and without accuracy gain (Fig. 4l and Supplementary Fig. 12l). The larger the average clustering coefficient was, the stronger the nodes in the functional connectivity network clustered together locally. We then measured the difference $\Delta_{Clustering}$ between the average clustering coefficients of neuronal and neurochemical functional connectivity networks (Fig. 4m and Supplementary Fig. 12m). We also calculated the

probability density function (PDF) and cumulative distribution function (CDF) of the degree centrality of nodes in neuronal and neurochemical functional connectivity networks in brain regions with and without accuracy gain (Fig. 4n; Supplementary Figs. 12n and 13c). A single-peaked PDF indicated that most nodes in the network had similar degree centrality, in contrast a bimodal distribution indicated that the nodes in the network had more heterogeneity in their attributes. The difference between the degree centrality distributions of different signals was measured by the Wasserstein distance[72] (Fig. 4o and Supplementary Fig. 12o). We also used another distance function called the Energy distance and obtained similar results[73]. The Wilcoxon signed-rank test was used to detect whether the above topology properties of each brain region with accuracy gain for rACh were significantly higher or lower than the average level of the selected brain regions without accuracy gain, i.e., LH, EPA, AOTU, and IVLP. Although for r5-HT, only MBVL exhibited accuracy gain (Supplementary Fig. 12b), we performed the same test as rACh.

## Ensemble analysis
**Odor tuning.** Odor tuning was defined as the difference in the AUC of responses to a specific odor compared to the other two odors. Voxels with significant tuning values were recorded. The maps of individual flies were aligned and averaged to generate a summary map (Fig. 3k). The standardized ratio of voxels with specific tuning in different brain regions within the community encompassing the majority of olfactory regions for each signal was calculated for comparison (Supplementary Fig. 10b, c).

**Odor identification weight.** The transformation matrices of the two-step PCA and LDA (see Odor identity classification across multiple brain regions) were multiplied to obtain the maps of odor identification weight. The maps of individual flies were aligned and averaged to generate a summary map (Fig. 3l). The standardized ratio of brain regions were calculated the same as *Odor tuning* (Supplementary Fig. 10e, f).

**Functional connectivity.** Voxels with high tuning to each odor were collected, and the Pearson correlations between the dynamics of these voxels were calculated to generate the functional connectivity matrices for G7f, rACh, and r5-HT (Supplementary Fig. 10g). Each functional connectivity matrix was organized based on hierarchical clustering, with the average traces of each voxel corresponding to the three odors displaying on the right side according to this ordering (Supplementary Fig. 10g). We mapped the spatial distribution of the strongest functionally connected edges (those with top 10% weights) of each functional connectivity matrix under three odor stimuli, i.e., OCT, MCH, and EA (Supplementary Fig. 10h). To compare the weighted coverage area of the functional connectivity networks for each odor stimulus, the weighted distance was calculated as the spatial distance between voxels multiplied by the functional correlation between them (Supplementary Fig. 10i).

## Temporal changes of manifolds
To analyze the temporal changes, the 60 sessions throughout the experiment were divided evenly into four stages, i.e., S1: Sessions 1–15, S2: Sessions 16–30, S3: Sessions 31–45 and S4: Sessions 46–60. We aligned and combined the manifolds of each fly to facilitate the following analyses. The low-dimensional traces of the corresponding sessions formed the manifolds of each stage. The characteristics of the manifolds at each stage were assessed. Distance to the origin, inter-class distance, and intra-class distance were measured as described above. Additionally, we examined the return locations (the average location of 29–30 s from the start of odor stimuli) of the sessions. We measured the x- and y-coordinates of the return locations and the distances between the return locations and the average location of the start time point (2 s before the onset of odor stimuli). Intra-class distance was measured for every three consecutive sessions, and other

metrics were measured for every session. The average values of each stage were compared (Fig. 5a–e).

## Temporal changes of functional connectivity networks

Brain-region-level functional connectivity networks for each stage were generated using the abovementioned method. We examined changes in the distribution of functional connectivity in physical space over time using the Scheirer–Ray–Hare test (Fig. 5g), and evaluated changes in the topological features of the functional connectivity networks. The changes in degree distribution (Fig. 5h) and average degree values (Fig. 5i) of the functional connectivity networks were examined using the Kolmogorov–Smirnov test and the Wilcoxon signed-rank test, respectively. We further examined whether the community structure of the functional connectivity networks for each signal changed significantly over time by calculating the community ratio (Fig. 5j).

## Motion analysis

We captured videos of fly abdomens and extracted the motions by calculating the absolute differences between consecutive frames (Supplementary Fig. 6a). The resulting motion energy, averaged across the FOV, was then correlated with the stimulus presented to the flies (binary sequence) using the Pearson correlation (Supplementary Fig. 6b). To match the temporal resolution and periods of the motion and neural dynamics, we downsampled the videos and extracted 12 s time windows from the 180 trials. PCA was applied to reduce the dimensionality and extract the behavioral features (Supplementary Fig. 6c). We retained the first 30 PCs. SVM was utilized to classify stimulus periods and intervals as well as odor identities based on these behavioral features (Supplementary Fig. 6d). Behavior during odor stimulation (5 s) was used in the classification. We conducted a label shuffling as a negative control and evaluated accuracies using 5-fold cross-validation. Next, we used the Ridge regression to predict the first 30 PCs of the multiple-brain-region dynamics from behavior, stimulus, or both (Supplementary Fig. 6e–g). Stimulus refers to the trial-averaged responses to each odor identity. $R^2$ was assessed by 5-fold cross-validation. We used the partial PCs explained by behavior and the residual after subtracting this explained part to conduct odor identity classifications (Supplementary Fig. 6g). The first 30 PCs were employed, following the same classification method as using the behavioral features.

The traces of motion and neural dynamics in the same fly during the same time period indicated some correlation (Supplementary Fig. 6h). To quantify the correlation, we calculated the Pearson correlations between motion and neural signals (G7f, rACh, and r5-HT) at the voxel level for different time-delay conditions (Supplementary Fig. 6i). We made neural dynamic traces differentially leading or delaying motor traces and then calculated the correlation between them. The temporal shift between neural signals and motion ranged within ±1 s at intervals of 0.1 s. Therefore, positive values of the horizontal coordinate in Supplementary Fig. 6i represent neural dynamics delayed from movement, while negative values represent neural dynamics advanced to movement. Correlations at the voxel level were averaged across each brain region and sorted by the temporal order of calcium signals (Supplementary Fig. 6j). We then calculated the average cross-correlation for each time-delay condition (Supplementary Fig. 6k), and measured the proportion of correlations above the mean for each delay condition (Supplementary Fig. 6l). To compare the three dynamics, the probability distributions of correlations between movement and neural dynamics (G7f, rACh, and r5-HT) for 23 brain regions were displayed (Supplementary Fig. 6m). The average correlations for three dynamics were compared and tested using the Wilcoxon signed-rank test (Supplementary Fig. 6n). We further compared the correlations across brain regions (Supplementary Fig. 6o).

## Statistics

Results of all statistical tests were listed in the Source data, providing the mean and standard error (SEM), statistic, effect size, and p value. Most of the statistical tests were performed using Python packages. Considering the small sample sizes, we selected non-parametric methods in the main text: Mann–Whitney U test for unpaired comparisons of two groups using pingouin.mwu and scipy.stats.mannwhitneyu; Wilcoxon signed-rank test for paired comparisons using pingouin.wilcoxon and scipy.stats.wilcoxon; Kolmogorov–Smirnov test for comparisons of the distribution of two groups using scipy.stats.ks_2samp; Kruskal–Wallis test (the non-parametric version of ANOVA) for comparisons of multiple groups using pingouin.kruskal; and Scheirer–Ray–Hare test used as the non-parametric two-way ANOVA test (https://github.com/jpinzonc/Scheirer-Ray-Hare-Test.git). For data passing the distribution normality test (pingouin.normality), the corresponding parametric tests were also performed with results included in the Source data. The parametric tests yielded similar results, which further solidified our conclusions. We used two-sided tests to show the difference of groups and used one-sided tests to examine whether the specific group of data showed a significantly higher/lower level compared to the other group or a certain threshold. We included multi-comparison corrections to mitigate the risk of false positives using statsmodels.stats.multitest.multipletests, selecting the Benjamini/Hochberg method (fdr_bh). Statistical tests of the behavioral experiment results in Supplementary Fig. 7d, e were performed in GraphPad Prism.

## Reporting summary

Further information on research design is available in the Nature Portfolio Reporting Summary linked to this article.

## Data availability

The data of an example fly generated in this study have been deposited on OneDrive (https://mailstsinghuaeducn-my.sharepoint.com/:f:/g/personal/fjq19_mails_tsinghua_edu_cn/EtZeYbE6qfFDpNpT_uv4Mi8BiAGpYAnsJEAz9RsjXmvZdw?e=asuQ30). The entire dataset with a total size of 5 TB, which includes the extracted neuronal and neurochemical traces within the 3D volumes of 10 flies co-labeled by G7f and rACh and 10 flies co-labeled by G7f and r5-HT, will be open-sourced as an important resource for the neurobiology and computational neuroscience communities. The link for data access will be provided on our GitHub page. Source data are provided with this paper.

## Code availability

The codes for data analysis are available on https://github.com/jqfan77/Dual_color_fly_brain_imaging_2pSAM_analysis.

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

## Acknowledgements

We thank Jun Zhou, Han Mo, Yunchuan Zhang, Wantong Hu, and Qi Yang at Yi Zhong's lab, and Xuelin Li at Yulong Li's lab for their help with *Drosophila* experiments, and Yeyi Cai, Yuanlong Zhang, and Tao Sun at our lab for their help with data processing. The figures of *Drosophila* are created with BioRender.com. This work is supported by the National Key Research and Development Project of China (2022YFF1202900), the National Natural Science Foundation of China (62088102, 62125106, 62222508, 62231018), Ministry of Science and Technology of China (2020AAA0105500), the China Postdoctoral Science Foundation (2023M741962) and Tsinghua Shuimu Scholar Program.

## Author contributions

Conceptualization: J.W., L.F., Q.D., and J.F.; Methodology: J.F., J.W., Y.W., L.L., J.H., Z.Z., F.D., G.L., X.L., Y.Z., J.Z., and N.H.; Investigation: J.F., Y.W., L.L., J.H., and Z.Z.; Visualization: J.F., Y.W., and L.L.; Funding acquisition: Q.D., L.F., and J.W.; Project administration: Q.D., L.F., and J.W.; Supervision: Q.D., L.F., and J.W.; Writing—original draft: J.F., J.W., Y.W., and L.L.; Writing—review & editing: Q.D., L.F., Y.L., and Y.H.

## Competing interests

The authors declare no competing interests.
