## [Transparent Peer Review file · Nature Communications]

Prominent involvement of acetylcholine dynamics in stable olfactory representation across the *Drosophila* brain

Corresponding Author: Professor Qionghai Dai

Version 0:

Reviewer comments:

Reviewer #1

(Remarks to the Author)

Review of "Prominent involvement of acetylcholine in shaping stable olfactory representation across the *Drosophila* brain" by Fan et al.

In this comprehensive manuscript the authors use two-photon synthetic aperture microscopy to record long-term neuronal and neuromodulatory olfactory responses to three odorants across the brains of 20 flies, 10 co-labeled by G7f and rACh and 10 co-labeled by G7f and r5-HT. The main take-home message is that neuromodulatory maps of acetylcholine maintain stable representations of odor identity, and therefore this neuromodulatory pattern likely takes a meaningful part in odor representation.

Major concerns

1. Sadly (as there is a lot of work here), I find the main claim unconvincing, and this for the following reason: As can be seen in Figure 2B, non-olfactory regions also have odor-specific responses. In fact, as stated on page 5: "While olfactory regions did not exhibit significantly higher responsiveness or response intensity compared to other regions for all three indicators, most olfactory regions surpassed non-olfactory regions in accuracy"

To state "most olfactory regions surpassed non-olfactory regions in accuracy" is not a scientific statement. What does most mean? How many and which olfactory regions surpassed the non-olfactory, and by how much? And critically, was this difference statistically significant?

This issue is compounded by the following: The authors used three odorants: 3-octanol (OCT), 4-methylcyclohexanol (MCH), and ethyl acetate (EA). OCT and MCH are aversive and EA is appetitive. In other words, if a brain region (or for that matter, the entire animal) is not classifying the odors at all, but only discriminating an aversive from appetitive condition, chance classification in this case is 49.5%. If we look at Figure 3B, this value is in fact almost never crossed! Thus, the manuscript contains no evidence for odor classification beyond separation of aversive and appetitive stimuli.... and moreover, this occurs in the non-olfactory regions as well. Frankly, testing the notion of "odor classification" using only 3 odorants is not really possible...

Given the amount of clearly valuable data in this manuscript, I would be very happy if the authors could tell me why I am wrong in this.

2. Even if the authors could convince that patterns of ACH neuromodulation are odor-specific, does this mean this is necessary for "shaping stable olfactory representation"? Wouldn't the authors need to somehow hamper/limit/change ACH neuromodulation and show that this influences behavioral odor discrimination? Again, I will be happy if the authors tell me why I am wrong....

Other concerns

1. Reporting statistics: The authors fail to report detailed statistics. They claim significance in the text, show significance

stars in the figures, but fail to report means/medians/variance/F stats/P stats/ d' / etc.

Minor concerns

1. The odor delivery apparatus is not described in depth ("using a custom-made air pump"). This is important as we see responses in non-olfactory regions. So was there an air-puff associated with the stimulus? If yes, was it always the same? If the authors had used many different odorants this would be less of an issue, but for three....
2. How was the sample size determined? Did the authors conduct a power analysis? Is this number typical in this field?
3. Statistical thresholds: The authors are very forthcoming on their threshold values, which should be commended. However, the decision process is not always clear. For example, the authors threshold PCA at 0.8, because this value provided optimal performance. Does this reasoning not make for a form of double-dipping? Also, they use PCA and SVM on frames 1-14, and LDA on frame 3 from odor delivery. Why these choices? Is there precedent? Is this just what worked? I am not very concerned about all this, but the authors should clarify
4. This question stems from my utter ignorance, but the authors should clarify for readers: Do we know for sure that olfaction is normal in such genetically modified flies?
5. The document is single-spaced, which rendered critical reading a challenge.

(Remarks on code availability)

Reviewer #2

(Remarks to the Author)

In this study, Fan and coauthors use a novel two-photon synthetic aperture microscopy (2pSAM), recently introduced by the same group (Zhao et al., 2023), to examine the relationship of brain-wide calcium, acetylcholine (ACh), and serotonin (5-HT) dynamics. In response to repeated odor exposures, these three types of signals exhibited distinct properties, including response profiles, information propagation, functional connectivity and odor identity representations. The main conclusion is that the ACh signal compensates for the calcium signal in decoding odor identity. Furthermore, ACh dynamics shows a more stable odor representation than the other two signals.

From a technical perspective, this is an impressive application of advanced two-photon synthetic aperture microscopy (2pSAM) combined with Deep-learning-powered denoising algorithms in imaging behaving *Drosophila*. The authors achieved significant advancements in fly whole-brain imaging by accomplishing a large field of view, covering nearly the entire fly brain. They also conducted imaging at a high speed of 30 Hz, which surpasses the slower rates (<5 Hz) in previous methods. Furthermore, they extended the imaging period to over 2 hours, compared to the previous limit of less than 30 minutes. I especially appreciate the team's commitment to making all the data and analysis code openly accessible. This suite of methods and the data analysis pipeline will benefit the neuroscience community, using fruit flies or other animals as model systems.

However, my main critique of the current manuscript is the limited biological insights gained from the sophisticated data analysis employed. While the study successfully observes distinct calcium, ACh, and 5-HT dynamics, it does not directly address the fundamental question the authors aimed to answer at the beginning of the paper, namely, how neuronal and neuromodulatory activities interact to synergistically represent and process information.

To improve the biological aspect of this study, I recommend that the authors add additional experiments and validations, as suggested in the following.

Major issue

The authors aim to study the interaction between neuromodulation and neural activity by simultaneously imaging brain-wide calcium and neurotransmitter/neuromodulator (ACh or 5-HT) dynamics, but the logic here is a bit circular. Global calcium activity itself serves as a proxy for various neuronal signals, including the release of neurotransmitters or neuromodulators. Hence, it is expected to observe differences among different signals. To address this concern and improve the study's mechanistic interpretation, it is crucial to incorporate neuron-type specificity and have testable hypotheses.

Specifically, the authors' claim that ACh dynamics compensate for neuronal activity in odor identity representation and exhibit better temporal stability, but it may merely reflect the fact that most primary olfactory inputs from olfactory sensory neurons, projection neurons, and Mushroom Body Kenyon cells are cholinergic. On the other hand, the calcium signals originate from various modulatory neuron types, such as GABAergic and dopaminergic neurons that innervate the same brain region. Therefore, the simplest interpretation of the results is that the different characteristics are inherited from different neuron types.

It is crucial to do additional experiments utilizing cell-type-specific labeling, such as MB-Gal4 or Cha-Gal4. If the ACh dynamics demonstrate superior identity coding compared to calcium dynamics when both signals are derived from MB Kenyon Cells, it may suggest an intrinsic difference between them. On the other hand, if both ACh and calcium dynamics exhibit similar identity coding patterns, it would imply that the differences observed using pan-neuronal labeling techniques are predominantly driven by variations between neuron types.

Another technical concern that arises when comparing calcium, ACh, and 5-HT dynamics is the differences in kinetic and

photostability among the indicators used. It is essential to characterize or discuss the extent to which the observed differences in phase delay and pulse width, as in Fig 2, are influenced by the indicators' characteristics. It would be beneficial for the authors to conduct basic validations by co-expressing calcium indicators and ACh or 5-HT indicators, specifically in a small cluster of cholinergic or serotonin neurons (such as DPM neurons). This approach would enable a direct comparison within the same neuronal population and help the readers to evaluate the other brain-wide observations presented in the study.

The denoising algorithms were previously published by the same group. However, this particular application lacks sufficient validation. In Extended Data Fig.1a, the traces have several obvious abnormalities. For example, the first and last cycles of odor responses in r5-HT MBVL traces look very similar in raw signals but differ significantly after denoising. This raises concern that the denoising procedures cause distortions in both amplitudes and dynamics. It would be helpful to perform some data analysis directly on raw data and then compare the results with and without denoising. For instance, conducting functional connectivity analysis and odor identity representation analysis on raw and denoised data could offer valuable insights into the potential impacts of the denoising procedures.

Minor points:

The authors claim the recording is 'long-term' in the abstract and several places in the text. Although the 2-hour duration is significantly improved than other whole-brain methods, the 'long-term' usually means over 24 hours to several days in the field. It would be more precise to state it as a 2-hour recording.

In the introduction, the authors state their aim is to explore the relationship between neuromodulation and neuron activity. However, it should be clarified that in the context of olfaction, ACh primarily functions as a neurotransmitter for the rapid relay of odor information rather than acting as a neuromodulator. The authors should consider rephrasing the question more precisely.

In Extended Data Figure 6, it would be beneficial to include the brain-wide correlation between movement and the three dynamics (calcium, ACh, and 5-HT), specifically during no-odor periods. Given that a previous whole-brain imaging study has examined the relationship between locomotion and neural activity (Brezovec et al., 2024), it would be valuable for the readers to compare the current technique with other relevant studies.

Most of the statistical analyses in this study are rigorous and appropriately used. However, when multiple comparisons are involved (e.g., Fig. 2e,f,k,l; Fig. 3h-j; Fig. 4f-h; Fig. 5b-j), the authors should use multi-comparison corrections, which helps mitigate the risk of false positives.

(Remarks on code availability)

Reviewer #3

(Remarks to the Author)

To understand the spatiotemporal dynamics of neuronal and neuromodulatory activities (e.g. serotonin (5-HT) and acetylcholine (ACh)), Fan et al employed 2pSAM to image neuronal intracellular calcium, ACh and 5-HT using pan-neuronally expressed genetically encoded indicators in the *Drosophila* brain. The authors performed volumetrically recording at 30 Hz for about 2 hours during presentation of three odors. To analyze odor representation, the authors used denoising algorithms to extract signals at voxel level across various brain regions. They had a few key findings: 1) odor-triggered responses of calcium, ACh and 5-HT are wide-spread throughout various brain regions, 2) the intensity of responses displays brain region specificity, 3) dynamic profile and global connectivity of each indicator are distinct. They further analyzed odor representation encoded by each signal and claimed that odor identity encoding is more accurately represented by calcium and ACh signals, but not by 5-HT. Specifically, ACh outperforms calcium activity in odor coding in certain brain regions, such as MBML and EB. Integrating ACh signals enhances the decoding of odor representation. Finally, the manifold analysis showed that calcium signals can distinguish between aversive and attractive/appetitive odors, while ACh and serotonin do not.

This study successfully demonstrated the utility of brain-wide, simultaneous two-color imaging of intracellular calcium and neuromodulator signals. The authors successfully performed characterization of responses detected by indicators to conclude that each signal can independently encode odor identity, albeit different levels of accuracy. They also successfully analyzed that integrating ACh signals to calcium signals enhances encoding of odor identity. To these two ends, analyses at the systems level were solid and well-presented.

However, any other conclusions beyond these two points require further evidence. The main weakness appears to be that the analyses do not elucidate/support whether calcium as the signal for neuronal activity, was modulated by either ACh or 5-HT signals, which was set and stated as the goals of this study (lines 49-51, 363, 369). Without cell-type specificity and anatomic level analysis, imaging information on its own is not sufficient to provide further evidence to enable this type of study. Although the manifold analysis showed that the three odors can be segregated, the authors did not demonstrate odor identification or segregation of odor identity coding in physical space at the voxel level (ensembles of odor representations). Conducting this analysis would further strengthen the data by showing odor encoding and potentially its segregation.

In conclusion, while the study is interesting, its significance, novelty and knowledge gained are very limited. Additionally, the methods used need further clarification. Therefore at current format, this study is not suitable for publication in this journal.

Minor concerns:

Fig. 1 - Did you try air puff stimulation to ensure these signals are odor-specific and not encoding mechanical stimulation to the antennae or a general aversive response (control experiment)?

Fig. 1b - What is the identity of the region showing the strongest G7f signal (Fig. 1b, region next to MB)? It would be helpful to indicate this. The 5-HT signal seems to correlate negatively with G7f activity in this region (compare G7f and r5-HT signals in Fig. 1b, 2a). Please comment.

Fig. 2a, Ext Fig. 1b - It would be beneficial to see the significance values of these apparently high correlations. Ext. Data Fig. 1b shows that most regions reported have generally high correlation with odor stimulus. Are there regions showing low correlation out of the 43 regions imaged to demonstrate the regional specificity of the response? It would be helpful to show these in the same plot to identify regions specifically involved in responding to odors and odor identification.

Fig. 2j - It would be better to have the nodes labeled as in Ext Data Fig. 3, with the names of the brain regions.

Fig. 1b and Fig. 2a - Why is the standard deviation of dF/F_0 , and not dF/F_0 , used to represent the response intensity (Methods, p.11, lines 498-499, and Fig. 2a, right panel)? Do the authors mean a z-score, which is the dF/F_0 (trace) divided by the standard deviation?

Fig. 2 - Is the onset of G7f before or after that of rACh or r5-HT signals? It would be helpful if a quantification is provided. Since the response of r5-HT is significantly slower and the authors show it 16 seconds after odor delivery compared to 8 seconds for G7f, it seems that r5-HT does not have a role in triggering the G7f response. This analysis will help elucidate whether the neuromodulators play a role in modulating the G7f signal.

Global information propagation - It is unclear why the phase delay (defined as the time between the onset and the peak of the response in Methods), rather than the time between the onset of odor delivery and the onset of the signals of individual sensors, is used to construct the schematic of global olfactory information propagation. Phase delay indicates the on kinetics of the sensor, so it is not a good measure for when the information is propagated to a specific region or voxel.

Fig. 3 - Are there specific voxels, especially in high-accuracy regions (e.g., LH for G7f and MBML for rACh), that are tuned to a specific odor, thus constituting tuning towards an odor? Would these "maps" be distinct for each of the three odors for any of the three sensors? Identification of voxels encoding each specific odor will enable their functional perturbation (e.g., by optogenetics) to test their functional involvement in odor coding in future studies, in addition to a theoretical description of their existence.

Fig. 3d, e, f, middle panels - It is unclear why trials cluster by number according to LDA. The authors mentioned in lines 234-235 that the trial order did not exhibit discernible separation. Please clarify.

Fig. 4 legends - Is the "deflation ratio" the same as the difference matrix? Its calculation is not explicitly mentioned in Methods. Please clarify.

Fig. 4i - Please state in Methods how the clustering coefficient C is calculated.

(Remarks on code availability)

Version 1:

Reviewer comments:

Reviewer #1

(Remarks to the Author)

I appreciate the efforts the authors made in their revision, and I remain impressed by the effort underlying this entire manuscript, but the product itself leaves me uneasy. A big part of this may reflect writing style, which impacts the paper overall. For example, this sentence in the abstract:

"We discovered the compensation between neuronal activity and cholinergic dynamics, both in the odor identity representation across the brain and the functional connectivity network structures of specific brain regions."

I read this again and again, but am not sure what is said here. I am not trying to be hard, or insistent, I just genuinely don't understand what is stated here. This is kind of word salad... This in part may merely be an English language usage issue, and that, with the help of Nature Editorial team, can be corrected with ease. For example, the above word-salad sentence may make (albeit only partial) sense if the third word, namely "the", is deleted. Moreover, I think the authors should run a find-and-replace on "we discovered" and replace it with "we found" (many instances), as well as delete the many uses of "interestingly" throughout the manuscript (for example: "Interestingly, the accuracy became stable after surpassing a certain threshold of dimensions." - why is this interesting...?). English alone is correctable, and doing so may take this manuscript way further.

But my issue goes beyond English alone. I remain particularly bothered by the countless statements of fact without statistical verification. For example: Lines 156-157 "For G7f, MB, especially MBVL exhibited faster response than other regions". In my discipline (indeed not the type of imaging used by the authors), this statement should appear as follows:

"For G7f, MB, especially MBVL exhibited faster response than other regions (MBVL time = $X \pm X$, Other regions mean = $X \pm X$, Statistic (df) = X , Effect size = X , p value = X).

However, the authors just state fact, and I am expected to accept it, or "see" it in the figure (here the authors direct to Figure 2e, but no statistics in 2e legend).

Now if this happened once or twice then fine, but its constant... e.g.:

Line 148: "Interestingly, the response intensities of G7f and rACh exhibited some degree of complementarity, with specific voxels displaying low intensity for G7f but high intensity for rACh (Fig. 2b, c)." Here I would expect a statistical test of "G7f vs. rACh".

Line 132: "We discovered that, three indicators displayed evident distinction in responsiveness." Here I would expect an ANOVA.

I could go on and on, results stated as facts without statistical tests. Notably, some stats were added after my previous comments on this front, but what was added was only p values, without the associated statistics, let alone measures of effect size. One should not report p values alone, and here the authors added entire tables of p values alone. Similarly, the authors report using Kolmogorov-Smirnov tests and Wilcoxon signed-rank tests, but don't justify why they use which test and when. For example, a test of distribution normality should be used to justify either parametric or non-parametric tests. And why once Kolmogorov-Smirnov and other times Wilcoxon signed-rank? Unclear... Moreover, the tests are occasionally one-sided, again without justification that I could find.

All and all, this remains a very impressive, and I think important data set, presented within a manuscript that I cannot say is good. I don't really know what to recommend here, as I don't think this effort should be rejected, nor would I publish it as is. Perhaps my review should receive less weight than the other Referees, and if they are positive, then go ahead and publish.

(Remarks on code availability)

Reviewer #2

(Remarks to the Author)

I thank the authors for providing a substantial revision which I believe have greatly improved the clarity of their text. These revisions along with additional experiments have improved the manuscript and addressed most of our previous concerns. After minor changes, this manuscript would be ready for publication.

Remaining issue:

In the Extended Data Fig. 10, it would be better to include example traces of GCaMP and rACh from the Mushroom Body regions (MBPED, MBVL, MBHL). Some plots similar to Fig. 2b,c will provide a clearer demonstration of this cell-type specific comparison. I'm particularly curious to see if the MB-specific GCaMP and rACh signals attenuate less across the 180 odor trials compared to the traces in Fig. 2c. This could be the reason why the manifolds are stable across all four stages now.

(Remarks on code availability)

Version 2:

Reviewer comments:

Reviewer #1

(Remarks to the Author)

I am sorry and feel badly about saying this, as I appreciate the authors have invested a lot of work, and yet, I cannot say that I think this is a good manuscript. For one, I remain irked by the style of reporting. This starts already at the abstract:

"Collectively, our unbiased and comprehensive investigation highlights the prominent involvement of ACh dynamics in stable olfactory representation across the brain, underscoring the inadequacy of solely considering neuronal activities when examining information representation of the brain."

What do you mean by "unbiased"? It is in fact super-biased... you looked at ACh/5-HT dynamics alone. Now, that is absolutely fine that you did that, it's your measure, and it's informative, but it is not unbiased... all measures, by definition, are biased. Moreover, I can't understand why you insist on all this "underscoring the inadequacy of solely considering neuronal activities", rather than concentrate on what it is you add....

But more substantively regarding reporting style, for example, page 12:

"Remarkably, we observed substantial accuracy gain upon integrating ACh signals at the multiple-brain-region level (Fig. 4c)."

Here (and CONSTANTLY throughout the manuscript), no values or statistics reported in the text. What is the "remarkable gain"? from what value to what value? If I try to pull it out of the figure (that doesn't start at 0, i.e., its stretched), it looks like a shift from about 94% to about 97%. Is that a remarkable gain? It does look consistent in the figure, but remarkable...? The manuscript is just full of this....

Beyond all this, and MOST importantly, what is it that here shows us that acetylcholine dynamics have a role in “stable olfactory representation”? Is there anything causal here? The fact that the authors improve THEIR decoding accuracy by integrating ACh signals doesn't imply that the fly brain is doing the same! The way things stand, the manuscript is a very impressive data collection on the dynamics of ACh/5-HT in the drosophila brain during odor exposure. It could be that this data will be useful. I don't know.

(Remarks on code availability)

Reviewer #2

(Remarks to the Author)

I thank the authors for addressing the final points raised in my previous review. I have reviewed the latest revisions and the accompanying response. The changes made have successfully resolved my remaining concerns. I am satisfied with the current version and have no further suggestions for improvement. I now recommend the manuscript for publication.

(Remarks on code availability)

Version 3:

Reviewer comments:

Reviewer #1

(Remarks to the Author)

The authors addressed the comments, and the manuscript provides valuable data

(Remarks on code availability)

We sincerely thank the reviewers for their thoughtful comments and constructive suggestions that we believe have helped us improve the manuscript substantially. We have provided a point-by-point response letter here with various new experiments and have revised the manuscript accordingly. The original reviewers' comments are shown in black color, whereas for ease of communication, our specific answers starting on the third page are in blue. Before we go into the details of each reviewer's comment and our point-by-point responses, here is a brief top-level summary of our revisions:

1. Improvement in biological insights: In response to the reviewer's comments, we have added substantial new experiments and analyses to enhance the biological insights of this paper. Firstly, we have added 2-class classifications at various scales and conducted new experiments with 7 odors to validate odor identification beyond mere preference separation. Secondly, we have added new experiments regarding cell-type specificity by specifically labelling the mushroom body cholinergic neurons, which complements our pan-neuronal labeling approach. We find clear evidence to show the distinct patterns of calcium and ACh. Thirdly, we compared odor representation and functional connectivity analysis results with and without the denoising process, which not only validates our conclusions but also shows the importance of denoising in our voxel-level, single-trial-based analyses. Finally, we investigated ensembles of olfactory representation for calcium, ACh, and r5-HT at the voxel level across the brain, uncovering odor- and indicator-specific ensemble patterns. These findings based on the constructive suggestions of reviewers significantly strengthen our data and provide deeper insights into olfactory representation.

2. Clarification of method details: To address the reviewer's concerns and enhance the overall clarity of our manuscript, we have provided additional clarifications on several methodological details. We have included more statistical information, such as exact *P* values, in the figure legends wherever possible. For multiple-comparison cases, we applied the Benjamini/Hochberg correction to reduce the risk of false positives. Moreover, we have revised the manuscript to clarify all methodological aspects, including the details of the odor delivery apparatus, the threshold determination process, and the parameters used in the functional connectivity analysis. Finally, we have added new experiments to validate the reliability of olfactory responses, the olfactory acuity of the genetically modified flies, and the kinetics of the rACh and r5-HT indicators. All these experiments help to make the conclusions of our manuscript solid.

3. Better presentation: Our study aimed to provide a novel perspective on the systemic study of information representation and neural networks, encompassing both neuronal activity and the dynamics of neuromodulators or neurotransmitters. Our main conclusion emphasized the significant role of ACh dynamics in olfactory representation across the brain, underscoring that considering only neuronal activities is insufficient for examining information representation. However, the original phrasing in the title and introduction sections may have inadvertently shifted attention towards the neuromodulatory properties of ACh and 5-HT, which extends beyond the intended scope of this paper. Additionally, we agree with Reviewer 2 that in the context of olfaction, acetylcholine (ACh) primarily functions as a neurotransmitter facilitating the rapid relay of odor information, rather than as a neuromodulator. As a result, we have revised the manuscript and the title to provide a clearer illustration of our scientific question. We removed misleading phrases to emphasize the core objectives of our study on the dynamic involvement of ACh during odor representation.

REVIEWER COMMENTS

Reviewer #1 (Remarks to the Author):

Review of “Prominent involvement of acetylcholine in shaping stable olfactory representation across the *Drosophila* brain” by Fan et al.

In this comprehensive manuscript the authors use two-photon synthetic aperture microscopy to record long-term neuronal and neuromodulatory olfactory responses to three odorants across the brains of 20 flies, 10 co-labeled by G7f and rACh and 10 co-labeled by G7f and r5-HT. The main take-home message is that neuromodulatory maps of acetylcholine maintain stable representations of odor identity, and therefore this neuromodulatory pattern likely takes a meaningful part in odor representation.

Response: We sincerely thank the reviewer for the invaluable comments. In response, we have enhanced our analysis of olfactory representation by providing more detailed characterizations to strengthen our findings. Additionally, we have explained the rationale behind the selection of odors and conducted supplemental analyses. We also performed further experiments, increasing the number of odors to 7, to substantiate our discoveries regarding multiple-brain-region olfactory representation. Furthermore, we have addressed other issues related to question illustration, experimental and statistical concerns, and presentations in detail listed below.

Major concerns

1. Sadly (as there is a lot of work here), I find the main claim unconvincing, and this for the following reason: As can be seen in Figure 2B, non-olfactory regions also have odor-specific responses. In fact, as stated on page 5: “While olfactory regions did not exhibit significantly higher responsiveness or response intensity compared to other regions for all three indicators,

most olfactory regions surpassed non-olfactory regions in accuracy”

To state “most olfactory regions surpassed non-olfactory regions in accuracy” is not a scientific statement. What does most mean? How many and which olfactory regions surpassed the non-olfactory, and by how much? And critically, was this difference statistically significant?

This issue is compounded by the following: The authors used three odorants: 3-octanol (OCT), 4-methylcyclohexanol (MCH), and ethyl acetate (EA). OCT and MCH are aversive and EA is appetitive. In other words, if a brain region (or for that matter, the entire animal) is not classifying the odors at all, but only discriminating an aversive from appetitive condition, chance classification in this case is 49.5%. If we look at Figure 3B, this value is in fact almost never crossed! Thus, the manuscript contains no evidence for odor classification beyond separation of aversive and appetitive stimuli..... and moreover, this occurs in the non-olfactory regions as well. Frankly, testing the notion of “odor classification” using only 3 odorants is not really possible...

Given the amount of clearly valuable data in this manuscript, I would be very happy if the authors could tell me why I am wrong in this.

Response: We sincerely thank the reviewer for the insightful comment. We are sorry for not making these points clear. We hope the following supplemental analyses and experiments can help address these problems.

(1) Detailed analysis of odor identity classification accuracy between olfactory and non-olfactory regions:

To address the reviewer’s first question, we calculated and mapped the responsiveness, response intensity, and classification accuracy across the field-of-view (FOV), to analyze the region heterogeneity during odor stimulus (**Response Fig. 1a-f**). We adjusted our calculation method to better capture regional variance and specificity in response to Reviewer 3's concerns, resulting in differences from the original responsiveness results. Consequently, we observe higher olfactory response specificity in both responsiveness and classification accuracy, while response

intensity (the standard deviation of $\Delta F/F$ during odor stimuli) shows lower regional variance.

For accuracy statistics, we compared the average classification accuracy of each olfactory region with the mean accuracy of non-olfactory regions and assessed the statistical significance. The results show that in the case of G7f, all olfactory regions significantly outperform the non-olfactory regions. For rACh, all regions except LH and SLP exceed the non-olfactory regions, and for r5-HT, all regions except MBPED, LH, and SLP surpass the non-olfactory regions. The accuracy for rACh and r5-HT demonstrates greater variability among the olfactory regions.

As the statistical charts depict the average accuracy of small blocks within each region, the overall accuracy level appears modest, primarily highlighting inter-regional differences. However, when integrating voxels within and across regions, the accuracy improves, suggesting a distributed coding pattern (**Response Fig.1g-l** and **Extended Data Fig. 4**).

Response Fig. 1 (Extended Data Figs. 1e-j, 4a-c, and 4h-j) | Responsiveness, response intensity, and odor identity classification accuracy across brain regions. a, Maps of correlation between G7f, rACh, and r5-HT dynamics and odor stimuli (averaged across flies, $n = 20$ for G7f, $n = 10$ for rACh, $n = 10$ for r5-HT). **b,** The average of the top 20% correlation values for each region (mean \pm s.e.m, $n = 20$ for G7f, $n = 10$ for rACh, $n = 10$ for r5-HT). **c,** Maps of the average standard deviation of $\Delta F/F$ during odor stimulation across trials for G7f, rACh, and r5-HT (averaged across flies, $n = 15$ for G7f, $n = 5$ for rACh, $n = 10$ for r5-HT). **d,** The average standard deviation of each region (mean \pm s.e.m, $n = 15$ for G7f, $n = 5$ for rACh, $n = 10$ for r5-HT). **e,** Maps of odor identity classification accuracy for G7f, rACh, and r5-HT (averaged across flies, $n = 20$ for G7f, $n = 10$ for rACh, $n = 10$ for r5-HT). **f,** The average odor identity

classification accuracy of each region (mean \pm s.e.m, $n = 20$ for G7f, $n = 10$ for rACh, $n = 10$ for r5-HT). The gray dashed lines indicate the average accuracy of the non-olfactory regions. One-sided Wilcoxon signed-rank test for each olfactory brain region against the average value. G7f: $P = 0.00031$ for SLP, $P < 0.0001$ for others. rACh: $P = 0.00081$ for MBPED, 0.00081 for MBVL, 0.00037 for MBML, 0.038 for LH, 0.996 for SLP, 0.00081 for SIP, 0.0093 for SMP, 0.00037 for CRE. r5-HT: $P = 0.18$ for MBPED, 0.0034 for MBVL, 0.0034 for MBML, 0.78 for LH, 0.16 for SLP, 0.0073 for SIP, 0.0036 for SMP, 0.0036 for CRE. **g-i**, Statistics of the average odor classification accuracy of the blocks and the accuracy integrating all voxels in each olfactory region of the left semi-brain for G7f (**g**), rACh (**h**), and r5-HT (**i**). Each light-colored line represents the average accuracy of a region across all flies. $n = 8$ regions, mean \pm s.e.m. One-sided Wilcoxon signed-rank test. $P = 0.0039$ in **g-i**. **j-l**, Statistics of the voxel-level odor identity classification accuracy across multiple brain regions and in single brain regions for G7f (**j**), rACh (**k**), and r5-HT (**l**). Each light-colored line represents the accuracy of a fly. $n = 9$ for G7f (G7f channel of flies co-labeled by G7f and rACh), $n = 9$ for rACh, $n = 9$ for r5-HT, mean \pm s.e.m. One-sided Wilcoxon signed-rank test is performed between “All” and each region and the Benjamini/Hochberg multi-comparison correction is applied. $P = 0.0020$ for all regions in **j, k**. $P = 0.0070$ for MBPED, 0.0075 for MBVL, 0.037 for MBML, 0.0044 for LH, 0.0044 for SLP, 0.0044 for SIP, 0.0075 for SMP, 0.011 for CRE, 0.0044 for EB in **l**. 10 flies co-labeled by G7f and rACh and 10 flies co-labeled by G7f and r5-HT are analyzed. Flies co-labeled by G7f and rACh with high-intensity non-specific fluorescence on the upper edge of the brain are excluded for clarity in **c, d**. Flies without a general coverage of region LH are excluded in **j-k** (one fly co-labeled by G7f and rACh and one fly co-labeled by G7f and r5-HT). Results of the left-side brain regions and the central complex are shown in the statistical charts. Scale bars: $50 \mu\text{m}$ in **a, c, e**. **** $P < 0.0001$, *** $P < 0.001$, ** $P < 0.01$, * $P < 0.05$, ns - not significant ($P > 0.05$).

(2) Odor classification beyond separation of aversive and appetitive stimuli:

For the second question, we agree that presenting the 3-class classification results without 2-class classification may lead to misunderstanding, particularly regarding the aversive and appetitive properties of the odors. To address this and demonstrate the ability of odor classification beyond separation of aversive and appetitive stimuli, we further conducted 2-odor classifications of each odor pair. The average area under the receiver operating characteristic (ROC) curve (AUC) (a metric of classification performance) of small blocks within each region is presented in **Response Fig. 2**, while the AUC of specific brain regions, obtained by integrating all

voxels within each region, is shown in **Response Fig. 3**. We evaluated whether the AUC was significantly higher than the chance level of 50% using a one-sided Wilcoxon signed-rank test. We used AUC rather than accuracy here because the number of each class is not entirely the same in the cross-validation process and AUC is a better metric in this case.

We observe that certain brain regions are able to significantly classify any two odors for each indicator, although the classification between OCT and MCH shows lower AUC compared to OCT/MCH and EA. Notably, r5-HT exhibits lower AUC than G7f and rACh. When voxels are integrated within each region, the AUC improves compared to the average AUC of small blocks within those regions as shown in **Response Figs. 2, 3**. These findings provide evidence for the ability of odor classification using the data of a single brain region.

Additionally, at the multiple-brain-region scale, the 3-odor classification AUC is notably high (**Response Fig. 4a, b**), especially for G7f, rACh, and integrating the two channels. The manifolds show clear distinctions among all three odors (**Response Fig. 4c-e**). To further clarify the classification results for each odor, we have also displayed the confusion matrices (**Response Fig. 4f, g**). These results highlight the capability of odor classification beyond the mere separation of aversive and appetitive stimuli when considering multiple-brain-region representations. These results are added in **Supplementary Figs. 2, 3** in the revised manuscript.

Response Fig. 2 (Supplementary Fig. 2) | Average 2-class classification AUC of the small blocks within each brain region for each odor pair. a-c, Average 2-class classification AUC between OCT and MCH (a), OCT and EA (b), MCH and EA (c) of each region for each indicator. $n = 20$ for G7f, $n = 10$ for rACh, $n = 10$ for r5-HT, mean \pm s.e.m. 10 flies co-labeled by G7f and rACh and 10 flies co-labeled by G7f and r5-HT are analyzed. Gray lines represent the chance level of 2-class classification AUC (50%). One-sided Wilcoxon signed-rank test for each brain region against the chance level. ** $P < 0.0001$, *** $P < 0.001$, ** $P < 0.01$, * $P < 0.05$, ns - not significant ($P > 0.05$, not labeled). O-M: OCT and MCH, O-E: OCT and EA, M-E: MCH and EA. Exact P values are shown in Table 1.**

Response Fig. 3 (Supplementary Fig. 3a-c) | 2-class classification AUC of each brain region for each odor pair. a-c, 2-class classification AUC between OCT and MCH (a), OCT and EA (b), MCH and EA (c) of each region for each indicator. $n = 20$ for G7f, $n = 10$ for rACh, $n = 10$ for r5-HT. Box plots: center line, median; box limits, upper and lower quartiles; whiskers, 1.5x interquartile range. Each point represents the result of a fly. 10 flies co-labeled by G7f and rACh and 10 flies co-labeled by G7f and r5-HT are analyzed. Gray lines represent the chance level of 2-class classification AUC (50%). One-sided Wilcoxon signed-rank test for each brain region against the chance level. ** $P < 0.0001$, *** $P < 0.001$, ** $P < 0.01$, * $P < 0.05$, ns - not significant ($P > 0.05$). O-M: OCT and MCH, O-E: OCT and EA, M-E: MCH and EA. Exact P values are shown in Table 1.**

1

Table 1.

2

Exact *P* values.

	MBPED	MBVL	MBML	LH	SLP	SIP	SMP	CRE	SCL	ICL	NO	EB	FB	LAL	AOTU	AVLP	PVLP	IVLP	VES	GOR	SPS	EPA	FLA
Response Fig. 2a - G7f	< 0.0001	< 0.0001	< 0.0001	0.011	< 0.0001	< 0.0001	< 0.0001	< 0.0001	0.0042	0.00013	0.013	0.0006	< 0.0001	< 0.0001	0.00029	< 0.0001	0.00043	0.14	0.00072	0.68	0.91	0.34	0.24
Response Fig. 2a - rACh	0.00098	0.00098	0.00098	0.42	0.0049	0.002	0.00098	0.00098	0.065	0.0029	0.12	0.00098	0.00098	0.0098	0.042	0.032	0.28	0.58	0.12	0.92	0.65	0.75	0.9
Response Fig. 2a - r5-HT	0.053	0.0098	0.0068	0.84	0.019	0.032	0.019	0.053	0.69	0.019	0.31	0.14	0.16	0.25	0.22	0.0049	0.65	0.08	0.69	0.93	1	0.5	0.72
Response Fig. 2b - G7f	< 0.0001	< 0.0001	< 0.0001	0.006	< 0.0001	< 0.0001	< 0.0001	< 0.0001	0.00085	< 0.0001	< 0.0001	< 0.0001	< 0.0001	< 0.0001	< 0.0001	< 0.0001	0.00085	0.87	< 0.0001	0.49	0.99	0.16	0.52
Response Fig. 2b - rACh	0.00098	0.00098	0.00098	0.12	0.002	0.00098	0.00098	0.00098	0.053	0.014	0.12	0.00098	0.00098	0.0049	0.002	0.002	0.0068	0.81	0.0068	0.62	0.92	0.097	0.72
Response Fig. 2b - r5-HT	0.08	0.00098	0.0098	0.81	0.042	0.00098	0.014	0.0068	0.28	0.65	0.58	0.35	0.12	0.35	0.024	0.042	0.38	0.46	0.38	0.9	0.98	0.065	0.08
Response Fig. 2c - G7f	< 0.0001	< 0.0001	< 0.0001	0.006	< 0.0001	< 0.0001	< 0.0001	< 0.0001	0.00035	< 0.0001	< 0.0001	< 0.0001	< 0.0001	< 0.0001	< 0.0001	< 0.0001	0.0006	0.018	< 0.0001	0.48	0.89	0.0028	0.012
Response Fig. 2c - rACh	0.00098	0.00098	0.00098	0.08	0.0049	0.002	0.00098	0.00098	0.042	0.002	0.032	0.00098	0.00098	0.0029	0.00098	0.00098	0.00098	0.12	0.0029	0.81	0.5	0.35	0.81
Response Fig. 2c - r5-HT	0.0049	0.00098	0.002	0.35	0.014	0.002	0.002	0.00098	0.0098	0.54	0.16	0.31	0.00098	0.54	0.097	0.0068	0.35	0.75	0.28	0.81	0.99	0.25	0.58
Response Fig. 3a - G7f	< 0.0001	< 0.0001	< 0.0001	0.006	< 0.0001	< 0.0001	< 0.0001	< 0.0001	--	--	--	< 0.0001	--	--	--	--	--	--	--	--	--	--	--
Response Fig. 3a - rACh	0.00098	0.00098	0.00098	0.08	0.31	0.00098	0.002	0.00098	--	--	--	0.00098	--	--	--	--	--	--	--	--	--	--	--
Response Fig. 3a - r5-HT	0.16	0.002	0.00098	0.98	0.14	0.0049	0.002	0.0049	--	--	--	0.032	--	--	--	--	--	--	--	--	--	--	--
Response Fig. 3b - G7f	< 0.0001	< 0.0001	< 0.0001	0.0053	< 0.0001	< 0.0001	< 0.0001	< 0.0001	--	--	--	< 0.0001	--	--	--	--	--	--	--	--	--	--	--
Response Fig. 3b - rACh	0.00098	0.00098	0.00098	0.053	0.014	0.00098	0.002	0.00098	--	--	--	0.00098	--	--	--	--	--	--	--	--	--	--	--
Response Fig. 3b - r5-HT	0.00098	0.00098	0.00098	1.0	0.12	0.00098	0.0049	0.0049	--	--	--	0.46	--	--	--	--	--	--	--	--	--	--	--
Response Fig. 3c - G7f	< 0.0001	< 0.0001	< 0.0001	0.0047	< 0.0001	< 0.0001	< 0.0001	< 0.0001	--	--	--	< 0.0001	--	--	--	--	--	--	--	--	--	--	--
Response Fig. 3c - rACh	0.00098	0.00098	0.00098	0.053	0.0029	0.00098	0.00098	0.00098	--	--	--	0.00098	--	--	--	--	--	--	--	--	--	--	--
Response Fig. 3c - r5-HT	0.024	0.002	0.00098	0.35	0.08	0.002	0.0049	0.042	--	--	--	0.053	--	--	--	--	--	--	--	--	--	--	--

3

Response Fig. 4 (Figs. 3d-g, 4c, and Supplementary Fig. 3d, e) | Odor identity classification at the multiple-brain-region scale. a, Statistics of the voxel-level multiple-brain-region odor identity classification accuracy by G7f, rACh, and r5-HT. $n = 20$ flies for G7f, $n = 10$ flies for rACh, $n = 10$ flies for r5-HT. Box plot: center line, median; box limits, upper and lower quartiles; whiskers, 1.5x interquartile range. Each point represents the result of a fly. Two-sided Mann-Whitney U test and Benjamini/Hochberg multi-comparison correction are applied. $P = 0.090$ for G7f vs rACh, $P < 0.0001$ for G7f vs r5-HT, $P = 0.00027$ for rACh vs r5-HT. **b**, Comparison of the voxel-level multiple-brain-region odor identity classification accuracy between using only the G7f channel and integrating both channels ($n = 10$, mean \pm s.e.m). Each light-colored line represents the result of a fly. One-sided Wilcoxon signed-rank test. $P = 0.00098$. **** $P < 0.0001$, *** $P < 0.001$, ** $P < 0.01$, * $P < 0.05$, ns - not significant ($P > 0.05$) in **a**, **b**. **c-e**, Low-dimensional manifolds of the G7f (**c**), rACh (**d**), and r5-HT (**e**)

responses to odor stimuli of a fly (left column: training set; right column: testing set). Top and middle: Each line is the odor response of a trial. Bottom: Each line is the average odor response of an odor identity. Manifolds from 2s before odor delivery to 33s after are plotted. Top: Colors denote odor identities (blue: OCT; red: MCH; orange: EA). Middle: Colors denote the trial numbers (blue: early; red: late). Bottom: Colors denote the time relative to odor delivery (blue: early; red: late). The manifolds in **c** and **d** are from the same fly. The arrow lines are arbitrary units but indicate an equivalent length in each dimension. **f**, Confusion matrices of odor identity classification for integrating G7f and rACh, G7f, and rACh ($n = 10$, averaged across flies). Color and values denote the ratio of the predicted and true class. **g**, Confusion matrices of odor identity classification for integrating G7f and r5-HT, G7f, and r5-HT ($n = 10$, averaged across flies), similar to **f**.

(3) Illustration of the number of odor used for classification during experiments:

The aim of this paper is to elucidate odor representation through the dynamics of calcium, acetylcholine, and serotonin across multiple brain regions using two-photon synthetic aperture microscopy (2pSAM), which offers 3D imaging capabilities and low phototoxicity. The simultaneous 3D imaging of calcium and neurotransmitter (ACh/5-HT) dynamics facilitates the comparison of different dynamics and their information representation characteristics, while also enabling the integration of information conveyed by different indicators. The low phototoxicity allows for relatively long recording periods, enabling a sufficient number of odor stimulus repetitions (trials) to analyze odor representation and its manifold, as well as the temporal changes in representation.

Using a small odor number facilitates a multifaceted analysis as it reduces the complexity of brain-wide analysis and allows for sufficient trials of each odor during the recording period (approximately 2 hours). Even for 3 odors, we have reached a huge TB-level dataset for 3D imaging. More odor number will significantly increase the data scale, requiring advanced algorithms to process. However, 3 odors have already enabled us to compare different odor representations within individual brain regions (through accuracy maps and voxel integration) and at the multiple-brain-region scale, exploring the differences among various odors and indicators, as well as the temporal changes in representation.

In the meantime, we do agree with the reviewer that supplemental experiments with a larger number of odors is important for further validation of certain aspects of our findings on odor representation. In this case, we have conducted new experiments with 7 odors (4 aversive and 3 appetitive), and analyzed the 7-class classification across the brain, as shown in **Response Fig. 5**. Although some odors exhibit higher levels of confusion, we find there is a clear increase of the classification accuracy by integrating G7f and rACh channels together, which is consistent with our main claim of the manuscript demonstrated by 3 odors. These results are added in **Supplementary Fig. 4** in the revised manuscript.

Response Fig. 5 (Supplementary Fig. 4) | Odor identity classification of 7 odors at the multiple-brain-region scale. a, Confusion matrices of odor identity classification for integrating G7f and rACh, G7f, and rACh of a fly. Color and values denote the ratio of the predicted and true class. **b**, The accuracy of the voxel-level multiple-brain-region odor identity classification changing with the dimensions of PCs retained (Step 3 in Fig. 3c), for G7f, rACh, and integrating both channels ($n = 3$, mean \pm s.e.m). Integrating both channels yields a higher accuracy for almost all dimension values. The dashed line (dimension = 25) marks the threshold taken, as the average accuracy reaches a high and stable level for all indicators. OCT: 3-octanol, MCH: 4-methylcyclohexanol, EA: ethyl acetate, 1-OCT: 1-Octen-3-ol, BEN: Benzaldehyde, IA: Isopentyl acetate, MS: Methyl salicylate.

2. Even if the authors could convince that patterns of ACH neuromodulation are odor-specific, does this mean this is necessary for “shaping stable olfactory representation”? Wouldn’t the authors need to somehow hamper/limit/change ACH neuromodulation and show that this influences behavioral odor discrimination? Again, I will be happy if the authors tell me why I am wrong....

Response: We sincerely thank the reviewer for pointing out this issue. We are sorry for the misleading due to inappropriate phrasing at the beginning of the main text. In fact, our study aimed to analyze information representation through the dynamics of calcium, ACh, and serotonin (5-HT), discussing their relationships and differences through comprehensive recording and analysis. Our main conclusion emphasized the significant role of ACh dynamics in olfactory representation across the brain, underscoring that considering only neuronal activities is insufficient for examining information representation. We provided a novel perspective for the systemic study of information representation and neural networks, as well as for understanding the dynamics and functions of neuromodulators and neurotransmitters, rather than focusing on the neuromodulation properties of ACh and 5-HT.

Discussions regarding neuromodulation and neuromodulators were intended to emphasize the importance of understanding the dynamics and information representation of these molecules. However, these expressions may inadvertently suggest a focus on "modulation," which extends beyond the scope of this paper. The wording "shaping" also causes this problem. Additionally, based on Reviewer 2's suggestions, it is crucial to clarify that, in the context of olfaction, acetylcholine (ACh) primarily functions as a neurotransmitter for the rapid relay of odor information, rather than as a neuromodulator. To illustrate the question more precisely and avoid potential confusion, we have revised and rephrased the manuscript. We have removed "shaping" from the title, and changed the title to "Prominent involvement of acetylcholine dynamics in stable olfactory representation across the *Drosophila* brain" to avoid confusion and better emphasize the question. In the abstract and introduction parts, we removed the content about neuromodulation and emphasized the importance and novelty of simultaneously recording and analyzing calcium and neurochemical (including neuromodulators and neurotransmitters) dynamics for a systemic and further understanding of information representation in the neural system.

Other concerns

1. Reporting statistics: The authors fail to report detailed statistics. They claim significance in the text, show significance stars in the figures, but fail to report means/medians/variance/F stats/P stats/ d' / etc.

Response: We sincerely thank the reviewer for pointing out this issue. We have added the information in the figure legends. For those panels with too many P values to report in the legends, like **Response Figs. 2, 3** above, we present them in supplementary tables like **Table 1**.

Minor concerns

1. The odor delivery apparatus is not described in depth (“using a custom-made air pump”). This is important as we see responses in non-olfactory regions. So was there an air-puff associated with the stimulus? If yes, was it always the same? If the authors had used many different odorants this would be less of an issue, but for three....

Response: We sincerely thank the reviewer for the constructive suggestion. We have added the schematic of the delivery system in **Supplementary Fig. 6 (Response Fig. 6a)** of the revised manuscript. This system is similar to those commonly used in studies on olfactory coding, learning, and memory¹⁻⁴.

The system maintained accurate airflow (500mL/min) at the output ends using two fast mass flow controllers and meters (Alicat Scientific). Odor (M1) and air (M2) were delivered to the fly during odor stimuli and intervals respectively, ensuring a consistent airflow to the fly throughout the experiment. Before each experiment, the exact airflow at the output ends was also measured with a flow-meter to verify both airflow consistency and smoothness.

The system featured a preparation state before odor delivery, allowing the path to be prefilled with the specific odor by switching valves 1 and 2. At the onset of odor stimuli, only valves 3 and 4 switched, resulting in a very brief air-puff that was independent of the specific odors.

To further clarify this problem, we have also added control experiments to assess neural responses to the brief air-puffs and compare them with responses to odor stimuli. We added a mineral oil bottle to the path N before valve 2. As a control, we only switched valves 3 and 4, and delivered N to the fly. 9 trials of OCT, MCH, EA, and the control were delivered to the fly in a pseudo-random order. We tested 3 flies co-labeled with G7f and rACh and compared their responses to the four stimuli (**Response Fig. 6b-e**). The flies did not exhibit obvious responses during the control stimuli, resulting in a significantly smaller area under the curve (AUC) compared to the responses to odor stimuli (**Response Fig. 6c, e**), although transient responses to the brief air-puff were observed in certain trials (**Response Fig. 6b, d**). We have added **Response Fig. 6b-e** to **Extended Data Fig. 1a-d** in the revised manuscript.

Response Fig. 6 (Supplementary Fig. 6 and Extended Data Fig. 1a-d) | The custom-built odor delivery system and control experiments assessing neural responses to the brief air-puffs. a, The custom-built odor delivery system. Air is divided into two streams and fed into two fast mass flow controllers and meters, where the airflow is precisely controlled to 500 mL/min. Two combined solenoid valves are used to allow either one odor (M1, Odor 1/2/3) or air (N) to pass through. Two three-way solenoid valves are used to select the air stream delivered to the fly. Odors are generated by diluting reagents in mineral oil, while air passing through a bottle

containing only mineral oil (M2) serves as the control airflow delivered to the fly during intervals. During intervals, M2 is delivered to the fly, and N is directed to the exhaust to remove waste gas and clean the path. Before odor stimuli, the system enters a preparation state where M2 is delivered to the fly, and M1 is directed to the exhaust to refill the path with the selected odor. During odor stimuli, M1 is delivered to the fly and M2 is directed to the exhaust. Odor number can be extended as additional ports are available on valves 1, 2. FC: fast mass flow controllers and meter. H: high potential. L: low potential. **b**, $\Delta F/F$ of G7f in response to OCT, MCH, EA, and the brief air-puff (control) in a region of interest (ROI) within the olfactory regions from the center view obtained by 2pSAM. The dark-colored lines represent trial average, and the light-colored lines represent each trial (2 trials of OCT, 1 trial of MCH, 3 trials of EA, and 3 trials of control in a pseudo-random order for each fly; 3 flies with the genotype $w; UAS-rGRAB_ACh-0.5/+; nSyb-Gal4, UAS-jGCaMP7f$). **c**, The box plot of the area under the curve (AUC) of the responses to odors and the control during odor stimuli periods ($n = 18$ for Odor, $n = 9$ for Control). The dots represent the AUC values of each trial. Two-sided Mann-Whitney U test, $P = 0.00019$. **d**, **e**, Similar to **b**, **c**, but for rACh responses. $P = 0.0094$ in **e**. **** $P < 0.0001$, *** $P < 0.001$, ** $P < 0.01$, * $P < 0.05$, ns - not significant ($P > 0.05$).

2. How was the sample size determined? Did the authors conduct a power analysis? Is this number typical in this field?

Response: We thank the reviewer for the comment. We adopted a typical sample size ($n = 10$ for most cases, $n = 20$ for some results of G7f) in fly neural imaging studies. We have labeled all the sample size in the legends. According to previous research, the commonly used sample size in this field is around 7~15.⁴⁻⁷ We did not conduct a power analysis, as it is not a common prerequisite in these studies due to the relatively small sample sizes often used. Small-sample-size studies are generally accepted in this field. We performed rigorous statistical analyses to ensure the reliability of the conclusions drawn from our data.

3. Statistical thresholds: The authors are very forthcoming on their threshold values, which should be commended. However, the decision process is not always clear. For example, the authors threshold PCA at 0.8, because this value provided optimal performance. Does this reasoning not

make for a form of double-dipping? Also, they use PCA and SVM on frames 1-14, and LDA on frame 3 from odor delivery. Why these choices? Is there precedent? Is this just what worked? I am not very concerned about all this, but the authors should clarify.

Response: We sincerely thank the reviewer for the comment. We chose a fixed threshold, since the analysis is not sensitive to the thresholds. We have added more results to show the analysis results with different thresholds. We tested various thresholds, as shown in **Response Fig. 7** (Extended Data Fig. 3), and presented the analysis results of the selected thresholds for convenience. We acknowledge the importance of clearly explaining the decision process and reasoning behind threshold selection. This clarification is provided below and has been included in the revised manuscript.

In **Response Fig. 7**, we illustrate how accuracy changes with different thresholds, explaining our threshold selection process. **Response Fig. 7a, b** display the accuracy of the voxel-level odor identity classification in 9 brain regions (integrating all voxels within each region) varying with the variance explained by the retained principal components (PCs). The accuracy and the relative relationship between different indicators did not show large change across a range of explained variance values. A threshold of 0.8 was chosen as most brain regions reached a high and stable level of accuracy. This threshold selection approach was similarly applied to voxel-level and region-level multiple-brain-region odor identity classifications (**Response Fig. 7c-h**).

Regarding the use of PCA and SVM on frames 1-14 and LDA on frame 3 from odor delivery, this choice was made because the odor response is particularly strong during this period, with the response at frame 3 being close to the peak. These thresholds are also not sensitive within a certain range. Applying LDA on other frames with high response can yield similar results, not restricting to frame 3. However, applying PCA and SVM across too many frames, especially including those long after the end of odor delivery, would introduce excessive noise and complicate the identification of the low-dimensional space for odor identification.

We have improved the expressions and clarified the decision process in the revised manuscript.

Response Fig. 7 (Extended Data Fig. 3) | PCA threshold determination for odor identity classifications. **a**, The accuracy of the voxel-level odor identity classification in 9 brain regions (left-side olfactory brain regions and EB) changing with the explained variance of the PCs retained, for G7f, rACh, and integrating both channels. The accuracy does not show significant change with the explained variance. A threshold of 0.8 is taken as most brain regions reach a high and stable level of accuracy, marked by the dashed line. **b**, Similar to **a**, but for flies co-labeled by G7f and r5-HT. **c**, The accuracy of the voxel-level multiple-brain-region odor identity classification changing with the dimensions of PCs retained (Step 3 in Fig. 3c), for G7f, rACh, and integrating both channels. Integrating both channels yields a higher accuracy for almost all dimension values. The dashed line (dimension = 25) marks the threshold taken, as the average accuracy reaches a high and stable level for all indicators. **d**, Similar to **c**, but for flies co-labeled by G7f and r5-HT. **c**, **d**, are results using the denoising algorithm SRDTrans. **e**, **f**, Similar to **c**, **d**, but are results using the denoising algorithm DeepCAD-RT. Obvious accuracy gains for integrating rACh and G7f signals exist using

both denoising algorithms. **g**, The accuracy of the region-level multiple-brain-region odor identity classification changing with the explained variance of the PCs retained, for G7f, rACh, and integrating both channels. A threshold of 0.995 is taken, marked by the dashed line. **h**, Similar to **g**, but for flies co-labeled by G7f and r5-HT. 10 flies co-labeled by G7f and rACh and 10 flies co-labeled by G7f and r5-HT are analyzed. $n = 10$, mean \pm s.e.m (shades). Flies without a general coverage of region LH are excluded from the statistics of this region.

4. This question stems from my utter ignorance, but the authors should clarify for readers: Do we know for sure that olfaction is normal in such genetically modified flies?

Response: We sincerely thank the reviewer for pointing out this issue and acknowledge the necessity for clarifying this point. To address this, we conducted behavioral odor avoidance tests and compared the performances of genetically modified and wildtype flies. Flies were placed at the choice point of a T maze, where they were given 1 minute to choose between the odor (OCT / MCH / EA) and air. We then counted the flies in each arm of the maze and calculated a performance index (PI). An index of 0 corresponds to a 50:50 distribution between the odor and air, while an index of 100% corresponds to complete avoidance of the odor. Additionally, a negative PI indicates a preference for the odor over air. We observe that all flies could sense odors and exhibited similar preferences (**Response Fig. 8a**). Although the flies used in this paper (w; UAS-rGRAB_ACh-0.5/+; nSyb-Gal4, UAS-jGCaMP7f and w; UAS-rGRAB_HTR2C-0.5/+; nSyb-Gal4, UAS-jGCaMP7f) showed a bit lower avoidance to MCH (**Response Fig. 8b**), it does not affect the conclusion in this manuscript. We added these results to **Extended Data Fig. 6d, e**.

Response Fig. 8 (Extended Data Fig. 6d, e) | Olfactory avoidance of genetically modified and wildtype flies. **a, b**, Performance indices of flies with four genotypes in response to given odors (OCT / MCH / EA). 100 flies used in an experiment, $n = 8$ experiments, mean \pm s.e.m. Kruskal-Wallis test in **b**, $P = 0.68$ for OCT, $P = 0.0029$ for MCH, $P = 0.13$ for EA. **** $P < 0.0001$, *** $P < 0.001$, ** $P < 0.01$, * $P < 0.05$, ns - not significant ($P > 0.05$).

5. The document is single-spaced, which rendered critical reading a challenge.

Response: We sincerely thank the reviewer for pointing out this issue. We apologize for any difficulties this may have caused during critical reading. In the revised manuscript, we have adjusted the line spacing to 1.5 lines.

Reviewer #2 (Remarks to the Author):

In this study, Fan and coauthors use a novel two-photon synthetic aperture microscopy (2pSAM), recently introduced by the same group (Zhao et al., 2023), to examine the relationship of brain-wide calcium, acetylcholine (ACh), and serotonin (5-HT) dynamics. In response to repeated odor exposures, these three types of signals exhibited distinct properties, including response profiles, information propagation, functional connectivity and odor identity representations. The main conclusion is that the ACh signal compensates for the calcium signal in decoding odor identity. Furthermore, ACh dynamics shows a more stable odor representation than the other two signals.

From a technical perspective, this is an impressive application of advanced two-photon synthetic aperture microscopy (2pSAM) combined with Deep-learning-powered denoising algorithms in imaging behaving *Drosophila*. The authors achieved significant advancements in fly whole-brain imaging by accomplishing a large field of view, covering nearly the entire fly brain. They also conducted imaging at a high speed of 30 Hz, which surpasses the slower rates (<5 Hz) in previous methods. Furthermore, they extended the imaging period to over 2 hours, compared to the previous limit of less than 30 minutes. I especially appreciate the team's commitment to making all the data and analysis code openly accessible. This suite of methods and the data analysis pipeline will benefit the neuroscience community, using fruit flies or other animals as model systems.

However, my main critique of the current manuscript is the limited biological insights gained from the sophisticated data analysis employed. While the study successfully observes distinct calcium, ACh, and 5-HT dynamics, it does not directly address the fundamental question the authors aimed to answer at the beginning of the paper, namely, how neuronal and neuromodulatory activities interact to synergistically represent and process information.

To improve the biological aspect of this study, I recommend that the authors add additional experiments and validations, as suggested in the following.

Response: We sincerely thank the reviewer for the recognition and positive comments. Our goal is to demonstrate novel applications of our microscopy techniques, including the microscopes and algorithms, to advance related neuroscience studies by establishing useful pipelines and providing new datasets. In this work, we conducted multiple-brain-region dual-indicator imaging of the *Drosophila* brain for approximately 2 hours at high speed, studying information representation through the dynamics of calcium, ACh, and serotonin (5-HT), and discussing their relationships and differences. We aim to offer a novel perspective for the systemic study of information representation and neural networks, as well as for understanding the dynamics and functions of neuromodulators and neurotransmitters. In the revision, we have incorporated supplemental experiments and validations based on the professional suggestions provided to further enhance the biological aspect of this study.

Major issue

The authors aim to study the interaction between neuromodulation and neural activity by simultaneously imaging brain-wide calcium and neurotransmitter/neuromodulator (ACh or 5-HT) dynamics, but the logic here is a bit circular. Global calcium activity itself serves as a proxy for various neuronal signals, including the release of neurotransmitters or neuromodulators. Hence, it is expected to observe differences among different signals. To address this concern and improve the study's mechanistic interpretation, it is crucial to incorporate neuron-type specificity and have testable hypotheses.

Specifically, the authors' claim that ACh dynamics compensate for neuronal activity in odor identity representation and exhibit better temporal stability, but it may merely reflect the fact that most primary olfactory inputs from olfactory sensory neurons, projection neurons, and Mushroom Body Kenyon cells are cholinergic. On the other hand, the calcium signals originate from various modulatory neuron types, such as GABAergic and dopaminergic neurons that innervate the same brain region. Therefore, the simplest interpretation of the results is that the different

characteristics are inherited from different neuron types.

It is crucial to do additional experiments utilizing cell-type-specific labeling, such as MB-Gal4 or Cha-Gal4. If the ACh dynamics demonstrate superior identity coding compared to calcium dynamics when both signals are derived from MB Kenyon Cells, it may suggest an intrinsic difference between them. On the other hand, if both ACh and calcium dynamics exhibit similar identity coding patterns, it would imply that the differences observed using pan-neuronal labeling techniques are predominantly driven by variations between neuron types.

Response: We sincerely thank the reviewer for the insightful comments and valuable suggestions. To address this, we conducted additional experiments using OK107-Gal4 (MB-Gal4) co-labeled with G7f and rACh. The flies underwent the same experimental paradigm, consisting of 180 trials of odor stimuli. We analyzed odor representation of the mushroom body cholinergic cells.

First, we calculated accuracy maps for G7f, rACh, and their integration, and assessed the accuracy gain from the integration compared to G7f alone (**Response Fig. 9a, b**). MBPED and MBML exhibited significant accuracy gains. We also generated manifolds of olfactory representation within the mushroom body (**Response Fig. 9c, d**). As expected, the separation of odors was less pronounced compared to the multiple-brain-region results. Nonetheless, integrating the dynamics of G7f and rACh yielded significantly higher odor classification accuracy (**Response Fig. 9e**) and better separation of odors in the manifolds (**Response Fig. 9f, g**). In addition, the results accord with the expectation of the reviewer that ACh dynamics demonstrate superior identity coding compared to calcium dynamics when both signals are derived from MB Kenyon Cells (**Response Fig. 9e**). We also analyzed the temporal changes of olfactory representation across the 2-hour period but observed no obvious alteration (**Response Fig. 9h, i**).

In conclusion, the compensatory role of ACh dynamics in olfactory representation to calcium dynamics remains evident in the mushroom body. However, the representation of both channels appears stable over the 2-hour period, suggesting that the temporal changes observed in G7f

during multiple-brain-region olfactory representation may originate from other brain regions.

We believe this cell-type-specific labeling experiment substantially strengthens our conclusions, thus improving the biological aspect of this study. These analysis results are added in **Extended Data Fig. 10**.

Response Fig. 9 (Extended Data Fig. 10) | Olfactory representation of the mushroom body cholinergic cells. a, Map of decoding-accuracy gain by integrating rACh dynamics, averaged across flies ($n = 6$). Scale bar: 50 μm . **b**, Average accuracy gain of MBPED, MBVL, and MBML ($n = 6$, mean \pm s.e.m). The dots and lines represent the values of each fly. One-sided Wilcoxon signed-rank test for each region against 0. $P = 0.047$ for MBPED, 0.22 for MBVL, 0.016 for MBML. **c, d**, Low-dimensional manifolds of two example flies. Top and middle: Each line is the odor response of a trial. Bottom: Each line is the average odor response of an odor identity.

Manifolds from 2s before odor delivery to 33s after are plotted. Top: Colors denote odor identities (blue: OCT; red: MCH; orange: EA). Middle: Colors denote the trial numbers (blue: early; red: late). Bottom: Colors denote the time relative to odor delivery (blue: early; red: late). The arrow lines are arbitrary units but indicate an equivalent length in each dimension. **e**, Left: The voxel-level odor identity classification accuracy changing with the dimensions of PCs retained, for G7f, rACh, and integrating both channels. Integrating both channels yields a higher accuracy for almost all dimension values. The dashed line (dimension = 25) marks the threshold taken, as the average accuracy reaches a high and stable level for all indicators. $n = 6$, mean \pm s.e.m (shades). Right: Comparison of the voxel-level odor identity classification accuracy between using only the G7f channel and integrating both channels ($n = 6$, mean \pm s.e.m). Each light-colored line represents the result of a fly. One-sided Wilcoxon signed-rank test, $P = 0.016$. **f**, **g**, Metrics of the manifolds for integrating both channels, G7f only and rACh only. **f**, Distance to the origin. Left: Distance to the origin changing with time relative to odor delivery. $n = 6$, mean \pm s.e.m (shades). Right: Comparison of the maximum distance to the origin between using only the G7f channel and integrating both channels ($n = 6$, mean \pm s.e.m). Each light-colored line represents the result of a fly. One-sided Wilcoxon signed-rank test, $P = 0.016$. **g**, Similar to **f**, but for Inter-class distance. $P = 0.016$ in the right panel. **h**, **i**, Metrics of the manifold change across four stages. **h**, Maximum distance to the origin. **i**, Maximum inter-class distance. Left: Changes of the maximum distance across sessions. Right: Statistics of the maximum distance across four stages for each indicator. $n = 6$, mean \pm s.e.m (shades and error bars). Each light-colored line represents the result of a fly. Two-sided Wilcoxon signed-rank test and Benjamini/Hochberg multi-comparison correction are applied. No significance is observed. $P = 0.63$ for S1 vs S2 and S1 vs S4 for rACh in **i**, $P = 1.0$ for others in **h**, **i**. **** $P < 0.0001$, *** $P < 0.001$, ** $P < 0.01$, * $P < 0.05$, ns - not significant ($P > 0.05$).

Another technical concern that arises when comparing calcium, ACh, and 5-HT dynamics is the differences in kinetic and photostability among the indicators used. It is essential to characterize or discuss the extent to which the observed differences in phase delay and pulse width, as in Fig 2, are influenced by the indicators' characteristics. It would be beneficial for the authors to conduct basic validations by co-expressing calcium indicators and ACh or 5-HT indicators, specifically in a small cluster of cholinergic or serotonin neurons (such as DPM neurons). This approach would enable a direct comparison within the same neuronal population and help the readers to evaluate the other brain-wide observations presented in the study.

Response: We sincerely thank the reviewer for the insightful comments. We initially compared the dynamics (phase delay and pulse width) of the three indicators as part of our basic description of olfactory responses. We do agree with the reviewer that these dynamics are related to the intrinsic characteristics of the indicators themselves.

Co-expressing calcium indicators and ACh or 5-HT indicators in a small cluster of neurons is a great suggestion to provide a direct comparison of the indicators' responses within the same neuronal population *in vivo*. Therefore, during the revision, we attempted to co-express G7f and r5-HT in DPM neurons (using C316-Gal4 and VT064246-Gal4⁴) and recorded the olfactory responses, but this effort was unsuccessful due to insufficient signal intensity under suitable laser power. Because these two DPM-Gal4 flies are typically used to modulate neurons rather than directly record calcium or 5-HT signals.

A rigorous comparison of calcium and neurotransmitter indicator kinetics is challenging because they represent different processes. Even if we had successfully compared the dynamics of calcium and ACh/5-HT within a small cluster of neurons, the results might not be generalizable to other neurons, as dynamic characteristics vary among different neuron types. After careful consideration, we have decided to remove the comparisons of calcium, ACh, and 5-HT dynamics from the manuscript, given that this analysis did not yield accurate or significant conclusions. We only retained the comparisons across brain regions for each indicator, as these analyses are not influenced by the kinetic differences of the indicators.

Nonetheless, a basic description of the kinetics of rACh and r5-HT indicators we used is still necessary to assist in evaluating the observations with temporal analysis. Therefore, we have added new experiments on HEK293T cells *in vitro* (**Response Fig. 10**). The measured on-kinetics of rACh and r5-HT are about 120 ms and 80 ms, and the off-kinetics are 1.82 s and 1.1 s, respectively. These results are added in **Extended Data Fig. 1k, l**.

Response Fig. 10 (Extended Data Fig. 1k, l) | The kinetics of rACh and r5-HT. a, Measurement of the kinetics of rACh expressed in HEK293T cells. Schematic illustration showing the local puffing system, Scale bar, 20 μ m. **b,** Representative traces of rACh fluorescence increase in response to ACh (top) and decrease in response to Scop (bottom). **c,** Group summary of on and off kinetics for rACh. n=11 cells for on and off kinetics, mean \pm s.e.m. **d,e,f,** Kinetic analysis of the r5-HT sensor, similar to **a,b,c**.

The denoising algorithms were previously published by the same group. However, this particular application lacks sufficient validation. In Extended Data Fig.1a, the traces have several obvious abnormalities. For example, the first and last cycles of odor responses in r5-HT MBVL traces look very similar in raw signals but differ significantly after denoising. This raises concern that the denoising procedures cause distortions in both amplitudes and dynamics. It would be helpful to perform some data analysis directly on raw data and then compare the results with and without denoising. For instance, conducting functional connectivity analysis and odor identity representation analysis on raw and denoised data could offer valuable insights into the potential

impacts of the denoising procedures.

Response: We sincerely thank the reviewer for the constructive comment. We recognize the necessity of clarifying the fidelity and effectiveness of the denoising process. Firstly, it is important to clarify that the “abnormalities” observed in the traces were likely caused by the calculation method of $\Delta F/F$. We have addressed this issue in **Response Fig. 11a** and updated the results in the revised manuscript. Additionally, we have included functional connectivity analysis (**Response Fig. 11b-e**) and odor identity representation analysis (**Response Fig. 11f-k**) on raw and denoised data to provide further clarification, based on the reviewer’s suggestion. Detailed responses for each concern are listed below:

(1) Illustration of the potential abnormalities in denoised traces in Extended Data Fig.1a:

These are not the abnormalities induced by the denoising algorithm. In the original manuscript, the traces represented the average $\Delta F/F$ of two brain regions, MBVL and MBML, for three indicators of an example fly, before and after denoising with DeepCAD-RT. The $\Delta F/F$ was calculated for each voxel and then averaged within a region. However, this approach can lead to inaccuracies in the calculation of baseline fluorescence (F_0) of individual voxels before denoising due to the low signal-to-noise ratio (SNR), which subsequently affects the average $\Delta F/F$ of the region. For example, many trials of the raw r5-HT traces did not exhibit pronounced responses. Thanks to the reviewer’s reminding, we recognize this problem.

In the current version, we have calculated and presented the $\Delta F/F$ of the averaged trace of the two regions rather than at voxel level, which better demonstrates the efficacy of the denoising methods (**Response Fig. 11a**). The previously noted "abnormalities" may appear less significant. However, it is crucial to recognize that while denoising algorithms reduce noise-related uncertainty, they cannot completely eliminate it. Distortion-like signals may still persist as residual uncertainty post-denoising. Nonetheless, denoising substantially reduces overall uncertainty, thereby enabling more accurate conclusions compared to the low-SNR raw data.

Furthermore, we conducted additional analyses directly on raw data and compared the results with and without denoising, using two flies co-labeled with G7f and rACh as examples.

(2) Functional connectivity analysis on raw and denoised data:

Firstly, we compared region-level and voxel-level functional connectivity before and after denoising. At the region level, the functional connectivity matrices for both G7f and rACh exhibited similarities and significant correlations between the raw and denoised data (**Response Fig. 11b, c**). However, denoising yielded higher average connectivity (**Response Fig. 11c**). The functional connectivity networks after denoising showed improved community division with respect to olfactory responses, with a notable increase in connectivity within the community containing most olfactory regions, particularly for rACh (**Response Fig. 11d**).

In the case of voxel-level functional connectivity in a region, taking MBML as an example, the denoised data revealed a bimodal node centrality distribution, indicating greater node heterogeneity and stronger clustering (**Response Fig. 11e**). In contrast, raw data exhibited lower clustering and a more concentrated node centrality distribution.

(3) Odor identity representation analysis on raw and denoised data:

Secondly, we performed comparisons on odor identity representation analysis. We utilized DeepCAD-RT in most analyses due to its superior capability in detecting subtle signal changes. To improve the recovery of temporal characteristics in manifold analyses, we implemented SRDTrans. We conducted analyses directly on raw and denoised data, and applied a hybrid approach where we trained on denoised data and applied the transformation to raw data (SRD-raw). All three methods successfully segregated and classified odors (**Response Fig. 11f**). The manifolds produced by the three methods exhibited similar characteristics; for instance, the angle between the manifolds of OCT and MCH was significantly smaller compared to other odor pairs (**Response Fig. 11g**). However, the denoised manifold showed the largest distance from the origin, indicating the best odor identity representation performance (**Response Fig. 11h, i**). The raw-data manifold and SRD-raw manifold did not show significant differences in this metric

(Response Fig. 11h, i).

When comparing odor identity classification accuracy, denoised data presented the highest accuracy, with SRD-raw showing higher accuracy than raw data (**Response Fig. 11j**). This result suggests that denoising more effectively identifies the low-dimensional space for odor identity representation than raw data, and that this space is also applicable to raw data. Nonetheless, all three methods support the conclusion that integrating G7f and rACh dynamics enhances accuracy in odor identity classification (**Response Fig. 11k**).

In summary, both raw and denoised data yield similar conclusions in most cases. Denoising enhances performance and produces more accurate results by reducing noise-related uncertainty. Indeed, the denoising methods play an important role in the single-trial-based analyses. Notably, the low-dimensional space identified by denoised data is also applicable to raw data, offering better performance compared to using raw data alone (**Response Fig. 11f, j**). This underscores the significance of denoising and validates the rationality of the results. However, the analysis maintain similar performance for both raw and denoised data, supporting the same main conclusion of the manuscript. These results are added in **Supplementary Fig. 5** in the revised manuscript.

Response Fig. 11 (Supplementary Fig. 5) | Effects of the denoising algorithms. a, $\Delta F/F$ of two brain regions for three indicators, before and after denoising with DeepCAD-RT and SRDTrans. **b**, Functional connectivity matrices of brain regions during odor stimulation for G7f and rACh averaged across two flies, before and after

denoising. The dashed boxes sign the communities containing most olfactory regions identified in Fig. 2. The color blocks above and to the left of each matrix mark 8 olfactory (coral) and 15 non-olfactory regions (black). **c**, Left: Scatter plots of the pairwise correlations between functional connectivity matrices for raw and denoised data. Pearson correlation is calculated, and R and P statistics are labeled. Right: Statistics of the correlation values. Kolmogorov-Smirnov test applied. **d**, Functional connectivity networks based on the matrices. The colors and widths of the edges indicate normalized connection strengths. The dashed circles sign the communities as in **b**. **e**, Node degree centrality distribution. Light-colored lines for raw data and dark-colored lines for Denoised data. Each light gray lines represents the result of the denoised or raw data from a single fly. Kolmogorov-Smirnov test applied. **f**, Low-dimensional manifolds for integrating both channels of the two example flies. Top and: Each line is the odor response of a trial. Bottom: Each line is the average odor response of an odor identity. Manifolds from 2s before odor delivery to 33s after are plotted. Top: Colors denote odor identities (blue: OCT; red: MCH; orange: EA). Bottom: Colors denote the time relative to odor delivery (blue: early; red: late). The arrow lines are arbitrary units but indicate an equivalent length in each dimension. Three columns present manifolds for denoised data with SRDTrans (SRD), raw data (Raw), and training on denoised data and applying the transformation to raw data (SRD-Raw). **g**, Statistics of the angles between the trial-averaged low-dimensional manifolds of two odors ($n = 6$, mean \pm s.e.m). O-M: OCT and MCH, O-E: OCT and EA, M-E: MCH and EA. Each light-colored line represents the manifold of integrating both channels, G7f or rACh of a fly. Two-sided Wilcoxon signed-rank test and Benjamini/Hochberg multi-comparison correction are applied. SRD: $P = 0.047$ for O-M vs O-E and O-M vs M-E, $P = 0.16$ for O-E vs M-E. Raw: $P = 0.047$ for O-M vs O-E and O-M vs M-E, $P = 0.84$ for O-E vs M-E. SRD-Raw: $P = 0.047$ for O-M vs O-E and O-M vs M-E, $P = 0.094$ for O-E vs M-E. **h**, Distance to the origin of the manifolds changing with time relative to odor delivery. $n = 6$. Dark-colored lines represent the average of each situation, and each light-colored line represent the result of integrating both channels, G7f or rACh of a fly. **i**, Comparison of the maximum distance to the origin for SRD, Raw, and SRD-Raw ($n = 6$, mean \pm s.e.m). Each light-colored line represents the result of integrating both channels, G7f or rACh of a fly. Two-sided Wilcoxon signed-rank test and Benjamini/Hochberg multi-comparison correction are applied. $P = 0.047$ for SRD vs Raw and SRD vs SRD-Raw, $P = 0.84$ for Raw vs SRD-Raw. **j**, Similar to **i**, but for comparison of the odor identity classification accuracy. $P = 0.043$ for SRD vs Raw, SRD vs SRD-Raw, and Raw vs SRD-Raw. **k**, Statistics of the odor identity classification accuracy for G7f and integrating both channels. Dark-colored lines represent the results of SRD, Raw, and SRD-Raw ($n = 2$, mean \pm s.e.m). Each light-colored line represents a fly. One-sided Wilcoxon signed-rank test for G7f vs Both ($n = 6$), $P = 0.016$. **** $P < 0.0001$, *** $P < 0.001$, ** $P < 0.01$, * $P < 0.05$, ns - not significant ($P > 0.05$).

Minor points:

The authors claim the recording is 'long-term' in the abstract and several places in the text. Although the 2-hour duration is significantly improved than other whole-brain methods, the 'long-term' usually means over 24 hours to several days in the field. It would be more precise to state it as a 2-hour recording.

Response: We sincerely thank the reviewer for pointing out this issue. We recognize that the claim of "long-term" is inappropriate and may cause some misunderstanding. We have corrected this statement in the revised manuscript.

In the introduction, the authors state their aim is to explore the relationship between neuromodulation and neuron activity. However, it should be clarified that in the context of olfaction, ACh primarily functions as a neurotransmitter for the rapid relay of odor information rather than acting as a neuromodulator. The authors should consider rephrasing the question more precisely.

Response: We sincerely thank the reviewer for pointing out this issue. Our study aimed to provide a novel perspective on the systemic study of information representation and neural networks, encompassing both neuronal activity and the dynamics of neuromodulators or neurotransmitters. Our main conclusion emphasized the significant role of ACh dynamics in olfactory representation across the brain, underscoring that considering only neuronal activities is insufficient for examining information representation. We did not delve deeply into the neuromodulation properties of ACh and 5-HT. Discussions regarding neuromodulation and neuromodulators were intended to emphasize the importance of understanding the dynamics and information representation of these molecules. We apologize for misunderstanding the actual identity of ACh in olfaction and the inappropriate expressions. To illustrate the question more

precisely and avoid potential confusion, we have revised and rephrased the manuscript. We have deleted the expressions “neuromodulation” and changed “neuromodulator” to “neuromodulator and neurotransmitter” or “neurochemical”.

In Extended Data Figure 6, it would be beneficial to include the brain-wide correlation between movement and the three dynamics (calcium, ACh, and 5-HT), specifically during no-odor periods. Given that a previous whole-brain imaging study has examined the relationship between locomotion and neural activity (Brezovec et al., 2024), it would be valuable for the readers to compare the current technique with other relevant studies.

Response: We sincerely thank the reviewer for the good suggestions. We have added the corresponding reference and the whole-brain correlations between movement and three dynamics in the revised version. We show the activity traces of movement, calcium, and neurotransmitters (rACh or r5-HT) in the same fly during the same time period (**Response Fig. 12a**). It can be roughly seen that there is some correlation between locomotion and neural activity. To quantify the correlation between movement and neural activity, we computed Pearson correlations between the two at the voxel level for different time-delay conditions. The correlation matrix of voxel levels throughout multiple brain regions demonstrates that there are different patterns of correlation between different neural signals and movement (**Response Fig. 12b**). In addition, differences in voxel activity within the same brain region can be seen, e.g., different degrees of time lag between voxel activity and movement in the brain region AUTO of r5-HT dynamics.

Firstly, we analyzed the temporal characterization of the correlation between the three dynamics with motion. Correlations at the voxel level were averaged across each brain region to obtain the movement-neurological activity correlation at the brain region level and sorted by the temporal order of calcium signals (**Response Fig. 12c**). It can be seen that the temporal error between G7f and movement is small, with the presence of more brain regions with high correlation moments occurring within ± 0.5 seconds of movement. In contrast, there is a longer

time delay between r5-HT dynamics and movement. We calculated the average cross-correlation of motor-neural activity for each time-delay condition (**Response Fig. 12d**). It can be found that the moments of high correlation between G7f (and rACh) and movement appeared within a time delay of 1 second, whereas the moments of high correlation between r5-HT and movement were expected to occur after 1 second. In addition, we calculated the proportion of motor-neural activity correlations above the mean for each delay condition (**Response Fig. 12e**), again showing that G7f was able to correlate with movement with small time delays (within 0.5 seconds), whereas r5-HT had larger time delays (above 1 second).

Secondly, we compared the correlation between movement and neural activity across the three dynamics. We show the probability distributions of correlations between movement and neural activity for G7f, rACh and r5-HT (**Response Fig. 12f**), and it can be seen that the peaks of correlations for G7f and rACh are different from those for r5-HT. Further statistical tests show that the average correlation of G7f signals is significantly higher than that of rACh, whereas r5-HT has the lowest average correlation with movement (**Response Fig. 12g**). Finally, we analyzed the correlation between movement and neural activity in the 23 brain regions. The results show the differentiation of the three dynamics in each brain region (**Response Fig. 12h**). These results are included in **Extended Data Fig. 5**.

Response Fig. 12 (Extended Data Fig. 5h-o) | Brain-wide correlation between movement and the three dynamics. **a**, The activity of movement, calcium, and neurotransmitters (rACh or r5-HT) during the same time period. **b**, Correlation matrix between movement and neural activity of voxel levels throughout multiple brain regions. The horizontal coordinate indicates the time lag of the neural activity in terms of the zero point of the movement, where the positive values represent neural activity delayed from movement and negative values represent neural activity advanced from movement. The horizontal coordinates of **c**, **d** and **e** have the same meaning. **c**, Correlation matrix between movement and neural activity at the brain region level. **d**, The average cross-correlation of motor-neural activity for each time-delay condition. **e**, The proportion of motor-neural activity correlations above the mean for each delay condition. **f**, The probability distributions of motor-neural activity correlations for each signal. **g**, The average cross-correlation of motor-neural activity for each signal. **h**, The average cross-correlation of motor-neural activity in the 23 brain regions. Normalized correlations are used in **b-h** for presentation and calculation. Two-sided Kruskal-Wallis test in **d**, **e** and **h**. Two-sided Wilcoxon signed-rank test and Benjamini/Hochberg multi-comparison correction applied in **g**. **** $P < 0.0001$, *** $P < 0.001$, ** $P < 0.01$, * $P < 0.05$, ns - not significant ($P > 0.05$, not shown in **e**).

Most of the statistical analyses in this study are rigorous and appropriately used. However, when multiple comparisons are involved (e.g., Fig. 2e,f,k,l; Fig. 3h-j; Fig. 4f-h; Fig. 5b-j), the authors

should use multi-comparison corrections, which helps mitigate the risk of false positives.

Response: We sincerely thank the reviewer for the insightful comments and valuable suggestions. We have now included multi-comparison corrections in the relevant statistical analyses using the Python package `statsmodels.stats.multitest.multipletests`, selecting the Benjamini/Hochberg method (`fdr_bh`). This method is also the default in the Python package `scipy.stats.false_discovery_control`. The correction results are consistent using the two packages.

Reviewer #3 (Remarks to the Author):

To understand the spatiotemporal dynamics of neuronal and neuromodulatory activities (e.g. serotonin (5-HT) and acetylcholine (ACh)), Fan et al employed 2pSAM to image neuronal intracellular calcium, ACh and 5-HT using pan-neuronally expressed genetically encoded indicators in the *Drosophila* brain. The authors performed volumetrically recording at 30 Hz for about 2 hours during presentation of three odors. To analyze odor representation, the authors used denoising algorithms to extract signals at voxel level across various brain regions. They had a few key findings: 1) odor-triggered responses of calcium, ACh and 5-HT are wide-spread throughout various brain regions, 2) the intensity of responses displays brain region specificity, 3) dynamic profile and global connectivity of each indicator are distinct. They further analyzed odor representation encoded by each signal and claimed that odor identity encoding is more accurately represented by calcium and ACh signals, but not by 5-HT. Specifically, ACh outperforms calcium activity in odor coding in certain brain regions, such as MBML and EB. Integrating ACh signals enhances the decoding of odor representation. Finally, the manifold analysis showed that calcium signals can distinguish between aversive and attractive/appetitive odors, while ACh and serotonin do not.

This study successfully demonstrated the utility of brain-wide, simultaneous two-color imaging of intracellular calcium and neuromodulator signals. The authors successfully performed characterization of responses detected by indicators to conclude that each signal can independently encode odor identity, albeit different levels of accuracy. They also successfully analyzed that integrating ACh signals to calcium signals enhances encoding of odor identity. To these two ends, analyses at the systems level were solid and well-presented.

However, any other conclusions beyond these two points require further evidence. The main weakness appears to be that the analyses do not elucidate/support whether calcium as the signal for neuronal activity, was modulated by either ACh or 5-HT signals, which was set and stated as

the goals of this study (lines 49-51, 363, 369). Without cell-type specificity and anatomic level analysis, imaging information on its own is not sufficient to provide further evidence to enable this type of study. Although the manifold analysis showed that the three odors can be segregated, the authors did not demonstrate odor identification or segregation of odor identity coding in physical space at the voxel level (ensembles of odor representations). Conducting this analysis would further strengthen the data by showing odor encoding and potentially its segregation.

In conclusion, while the study is interesting, its significance, novelty and knowledge gained are very limited. Additionally, the methods used need further clarification. Therefore at current format, this study is not suitable for publication in this journal.

Response: We sincerely thank the reviewer for the recognition, insightful comments and constructive suggestions. We acknowledge the weaknesses in question illustration, analysis and method clarification mentioned in the comments. We have substantially revised the manuscript to illustrate the question and methods more precisely and added a series of experiments with cell type specificity and ensemble analyses to strengthen our conclusion. We believe the revised version has significant improvements in the aspect of biological insights. Detailed modifications are listed below:

(1) Cell-type-specific labeling and analysis:

In accordance with Reviewer 2's comment, we recognize the importance of adding cell-type-specific labeling and analyzing odor identity representation to further strengthen our conclusions. We conducted additional experiments using OK107-Gal4 (MB-Gal4) co-labeled with G7f and rACh. The flies underwent the same experimental paradigm, consisting of 180 trials of odor stimuli. We analyzed odor representation of the mushroom body cholinergic cells.

First, we calculated accuracy maps for G7f, rACh, and their integration, and assessed the accuracy gain from the integration compared to G7f alone (**Response Fig. 13a, b**). MBPED and MBML exhibited significant accuracy gains. We also generated manifolds of olfactory representation within the mushroom body (**Response Fig. 13c, d**). As expected, the separation of

odors was less pronounced compared to the multiple-brain-region results. Nonetheless, integrating the dynamics of G7f and rACh yielded significantly higher odor classification accuracy (**Response Fig. 13e**) and better separation of odors in the manifolds (**Response Fig. 13f, g**). In addition, the results showed that ACh dynamics demonstrate superior identity coding compared to calcium dynamics when both signals are derived from MB Kenyon Cells (**Response Fig. 13e**). We also analyzed the temporal changes of olfactory representation across the 2-hour period but observed no obvious alteration (**Response Fig. 13h, i**).

In conclusion, the compensatory role of ACh dynamics in olfactory representation to calcium dynamics remains evident in the mushroom body. However, the representation of both channels appears stable over the 2-hour period, suggesting that the temporal changes observed in G7f during multiple-brain-region olfactory representation may originate from other brain regions. These analysis results are added in **Extended Data Fig. 10**.

Response Fig. 13 (Extended Data Fig. 10) | Olfactory representation of the mushroom body cholinergic cells. **a**, Map of decoding-accuracy gain by integrating rACh dynamics, averaged across flies ($n = 6$). Scale bar: $50 \mu\text{m}$. **b**, Average accuracy gain of MBPED, MBVL, and MBML ($n = 6$, mean \pm s.e.m). The dots and lines represent the values of each fly. One-sided Wilcoxon signed-rank test for each region against 0. $P = 0.047$ for MBPED, 0.22 for MBVL, 0.016 for MBML. **c, d**, Low-dimensional manifolds of two example flies. Top and middle: Each line is the odor response of a trial. Bottom: Each line is the average odor response of an odor identity. Manifolds from 2s before odor delivery to 33s after are plotted. Top: Colors denote odor identities (blue: OCT; red: MCH; orange: EA). Middle: Colors denote the trial numbers (blue: early; red: late). Bottom: Colors denote the time relative to odor delivery (blue: early; red: late). The arrow lines are arbitrary units but indicate an equivalent length in each dimension. **e**, Left: The voxel-level odor identity classification accuracy changing with the dimensions of PCs retained, for G7f, rACh, and integrating both channels. Integrating both channels yields a higher accuracy for almost all dimension values. The

dashed line (dimension = 25) marks the threshold taken, as the average accuracy reaches a high and stable level for all indicators. $n = 6$, mean \pm s.e.m (shades). Right: Comparison of the voxel-level odor identity classification accuracy between using only the G7f channel and integrating both channels ($n = 6$, mean \pm s.e.m). Each light-colored line represents the result of a fly. One-sided Wilcoxon signed-rank test, $P = 0.016$. **f, g**, Metrics of the manifolds for integrating both channels, G7f only and rACh only. **f**, Distance to the origin. Left: Distance to the origin changing with time relative to odor delivery. $n = 6$, mean \pm s.e.m (shades). Right: Comparison of the maximum distance to the origin between using only the G7f channel and integrating both channels ($n = 6$, mean \pm s.e.m). Each light-colored line represents the result of a fly. One-sided Wilcoxon signed-rank test, $P = 0.016$. **g**, Similar to **f**, but for Inter-class distance. $P = 0.016$ in the right panel. **h, i**, Metrics of the manifold change across four stages. **h**, Maximum distance to the origin. **i**, Maximum inter-class distance. Left: Changes of the maximum distance across sessions. Right: Statistics of the maximum distance across four stages for each indicator. $n = 6$, mean \pm s.e.m (shades and error bars). Each light-colored line represents the result of a fly. Two-sided Wilcoxon signed-rank test and Benjamini/Hochberg multi-comparison correction are applied. No significance is observed. $P = 0.63$ for S1 vs S2 and S1 vs S4 for rACh in **i**, $P = 1.0$ for others in **h, i**. **** $P < 0.0001$, *** $P < 0.001$, ** $P < 0.01$, * $P < 0.05$, ns - not significant ($P > 0.05$).

(2) Neuromodulation:

For the modulation capability of ACh or 5-HT signals, we are sorry about the misleading arose from inappropriate phrasing at the beginning of the main text. We mentioned “the relationship between neuronal activity and neuromodulator release” (lines 49-51, 363, 369), which means the quantitative similarities and difference of their dynamics across the brain *in vivo*. Our study aimed to provide a novel perspective on the systemic study of information representation and neural networks, encompassing both neuronal activity and the dynamics of neuromodulators or neurotransmitters. Our main conclusion emphasized the significant role of ACh dynamics in olfactory representation across the brain, underscoring that considering only neuronal activities is insufficient for examining information representation. We did not focus on the neuromodulation properties of ACh and 5-HT.

Discussions regarding neuromodulation and neuromodulators were intended to emphasize the importance of understanding the dynamics and information representation of these molecules. However, these expressions may inadvertently suggest a focus on "modulation," which extends

beyond the scope of this paper. Additionally, based on Reviewer 2's suggestions, it is crucial to clarify that, in the context of olfaction, acetylcholine (ACh) primarily functions as a neurotransmitter for the rapid relay of odor information, rather than as a neuromodulator. We apologize for the inappropriate expressions and the misleading. To illustrate the question more precisely and avoid potential confusion, we have revised and rephrased the manuscript. We have removed "shaping" from the title, and changed the title to "Prominent involvement of acetylcholine dynamics in stable olfactory representation across the *Drosophila* brain" to avoid latent confusion aroused by "shaping" and better emphasize the question. In the abstract and introduction parts, we removed the content about neuromodulation and emphasized the importance and novelty of simultaneously recording and analyzing calcium and neurochemical (including neuromodulators and neurotransmitters) dynamics for a systemic and further understanding of information representation in the neural system.

(3) Ensembles of odor representations:

To further strengthen our analyses of olfactory information representation, we have added ensemble analysis in our revised manuscript.

In the previous version of the manuscript, we analyzed the responsiveness and odor identity classification accuracy across the brain (**Response Fig. 14a, b**), which revealed the distributions of reliable olfactory responses. Voxels with reliable olfactory responses could be recognized as olfactory ensembles.

We further analyzed odor tuning across the brain, which we defined as the difference in the area under the curve (AUC) of the responses to a specific odor compared to the other two odors. Voxels with significant tuning values were retained and displayed in **Response Fig. 14c**. Subsequently, we calculated the standardized ratio of voxels with specific tuning in different brain regions (**Response Fig. 14d, e**), focusing exclusively on brain regions within the community that encompasses the majority of olfactory regions for each indicator (Fig. 2f). The

tuning exhibited a distributed pattern for all indicators. The distributions of tuning for each indicator (**Response Fig. 14d**) and each odor (**Response Fig. 14e**) showed evident distinctions.

Additionally, based on our odor identity representation analysis, we calculated the identification weight for each odor across voxels and generated corresponding weight maps (**Response Fig. 14f-h**). The distributions of tuning and weight exhibited clear similarities. For instance, both analyses highlighted the strong role of MBML for rACh and LH for G7f in the identification of OCT (**Response Fig. 14c-h**). Furthermore, all three indicators revealed a strong effect of MBVL in the identification of EA (**Response Fig. 14c-h**).

Functional connectivity analysis could also provide further insight into olfactory representation ensembles. We measured the functional connectivity of voxels with high tuning to specific odors and generated functional connectivity matrices (**Response Fig. 14i**). Voxels responding to different odor stimuli displayed significantly distinct activity traces (**Response Fig. 14i, right part**). Correspondingly, voxels responding to the same stimulus exhibited strong functional connectivity, while the connectivity between voxel ensembles responding to different stimuli was weak, resulting in three clear functional modules (clusters obtained by clustering) in the functional connectivity matrices (**Response Fig. 14i, left part**). This phenomenon was consistent across all three signals (**Response Fig. 14i**), confirming that voxel ensembles for each signal are capable of odor identification to some extent.

We also discerned differences in the functional connectivity networks of voxel ensembles in physical space. We mapped the physical spatial distribution of the strongest functionally connected edges (those with top 10% weights) of a fly under three odor stimuli (**Response Fig. 14j**). The functional connectivity networks for the three odors exhibited significant differences in physical space, particularly in the location of core voxel ensembles and the spatial extent of the voxel network. This phenomenon was observed consistently across all three signals (**Response Fig. 14j**). Statistical tests across multiple flies and signals revealed that for G7f and rACh, the odor MCH had a significantly larger weighted coverage area in the functional network compared to OCT (**Response Fig. 14k**), suggesting more long-range, large-weighted functional connectivity

edges in MCH network than in the OCT network. These results are added in **Fig. 3k, l** and **Extended Data Fig. 7**.

Response Fig. 14 (Fig. 3k, l and Extended Data Fig. 7) | Ensembles of odor representations. **a**, Top left: Maps of correlation between G7f, rACh, and r5-HT dynamics and odor stimuli (averaged across flies, $n = 20$ for G7f, $n = 10$ for rACh, $n = 10$ for r5-HT). Statistical charts: The average of the top 20% correlation values for each

region (mean \pm s.e.m, $n = 20$ for G7f, $n = 10$ for rACh, $n = 10$ for r5-HT). **b**, Top left: Maps of odor identity classification accuracy for G7f, rACh, and r5-HT (averaged across flies, $n = 20$ for G7f, $n = 10$ for rACh, $n = 10$ for r5-HT). Statistical charts: The average odor identity classification accuracy of each region (mean \pm s.e.m, $n = 20$ for G7f, $n = 10$ for rACh, $n = 10$ for r5-HT). The gray dashed lines indicate the average accuracy of the non-olfactory regions. One-sided Wilcoxon signed-rank test for each olfactory brain region against the average value. G7f: $P = 0.00031$ for SLP, $P < 0.0001$ for others. rACh: $P = 0.00081$ for MBPED, 0.00081 for MBVL, 0.00037 for MBML, 0.038 for LH, 0.996 for SLP, 0.00081 for SIP, 0.0093 for SMP, 0.00037 for CRE. r5-HT: $P = 0.18$ for MBPED, 0.0034 for MBVL, 0.0034 for MBML, 0.78 for LH, 0.16 for SLP, 0.0073 for SIP, 0.0036 for SMP, 0.0036 for CRE. **c**, Maps of odor tuning. Colors denote the average difference of area under the curve (AUC) of the responses to the specific odor compared to the other two odors (averaged across flies, $n = 20$ for G7f, $n = 10$ for rACh, $n = 10$ for r5-HT). **d**, **e**, Statistics of the standardized ratio of voxels with specific tuning in each region (mean \pm s.e.m, $n = 20$ for G7f, $n = 10$ for rACh, $n = 10$ for r5-HT). Regions within the community containing most olfactory regions for each indicator are displayed. Each chart shows the result for a specific tuning in **d**, results for a specific indicator in **e**. **f-h**, Similar to **c-e**, but for odor identification weight. **i**, Functional connectivity matrix and traces of voxel ensembles. The connectivity matrix is clustered into 3 modules by hierarchical clustering methods. **j**, Functional connectivity network of three odors. The nodes are the ensemble of voxels that responded strongly to odor stimuli, and the connectivity edges shown are the top 10% functional edges with the highest correlations. **k**, Comparison of weighted distances (mean \pm s.e.m, $n = 20$ for G7f, $n = 10$ for rACh, $n = 10$ for r5-HT). The weighted distance was calculated as the physical spatial distance between a pair of voxels multiplied by the functional correlation between them. Two-sided Wilcoxon signed-rank test and Benjamini/Hochberg multi-comparison correction applied. $P = 0.0024$ for OCT vs MCH, $P < 0.0001$ for OCT vs EA, $P = 0.029$ for MCH vs EA in G7f; $P = 0.032$ for OCT vs MCH, $P = 0.09$ for OCT vs EA, $P = 0.032$ for MCH vs EA in rACh; $P = 0.28$ for OCT vs MCH, $P = 0.0019$ for OCT vs EA, $P = 0.0019$ for MCH vs EA in r5-HT. 10 flies co-labeled by G7f and rACh and 10 flies co-labeled by G7f and r5-HT are analyzed. Scale bars: $50 \mu\text{m}$ in **a**, **b**. **** $P < 0.0001$, *** $P < 0.001$, ** $P < 0.01$, * $P < 0.05$, ns - not significant ($P > 0.05$).

Minor concerns:

Fig. 1 - Did you try air puff stimulation to ensure these signals are odor-specific and not encoding mechanical stimulation to the antennae or a general aversive response (control experiment)?

Response: We sincerely thank the reviewer for this great suggestion. We have conducted control experiments and validated that the signals are odor-specific rather than encoding mechanical stimulation.

We employed a custom-built odor delivery system, similar to those commonly used in studies on olfactory coding, learning, and memory¹⁻⁴. The structure of the system is illustrated in **Response Fig. 15a**. The system maintained accurate airflow (500mL/min) at the output ends using two fast mass flow controllers and meters (Alicat Scientific). M1 (Odor) and M2 (Air) were delivered to the fly during odor stimuli and intervals respectively, ensuring a consistent airflow to the fly throughout the experiment. Before each experiment, the exact airflow at the output ends was also measured with a flow-meter to verify both airflow consistency and smoothness. The system featured a preparation state before odor delivery, allowing the path to be prefilled with the specific odor by switching valves 1 and 2. At the onset of odor stimuli, only valves 3 and 4 switched, resulting in a very brief air-puff that was independent of the specific odors.

We conducted control experiments to assess neural responses to brief air-puffs and compare them with responses to odor stimuli. We added a mineral oil bottle to the path N before valve 2. As a control, we only switched valves 3 and 4, and delivered N to the fly. 9 trials of OCT, MCH, EA, and the control were delivered to the fly in a pseudo-random order. We tested 3 flies co-labeled with G7f and rACh and compared their responses to the four stimuli (**Response Fig. 15b-e**). The flies did not exhibit obvious responses during the control stimuli, resulting in a significantly smaller area under the curve (AUC) compared to the responses to odor stimuli (**Response Fig. 15c, e**), although transient responses to the brief air-puff were observed in certain trials (**Response Fig. 15b, d**). We have added **Response Fig. 15a** to **Supplementary Fig. 6** and **Response Fig. 15b-e** to **Extended Data Fig. 1a-d**.

Response Fig. 15 (Supplementary Fig. 6 and Extended Data Fig. 1a-d) | The custom-built odor delivery system and control experiments assessing neural responses to the brief air-puffs. a, The custom-built odor delivery system. Air is divided into two streams and fed into two fast mass flow controllers and meters, where the airflow is precisely controlled to 500 mL/min. Two combined solenoid valves are used to allow either one odor (M1, Odor 1/2/3) or air (N) to pass through. Two three-way solenoid valves are used to select the air stream delivered to the fly. Odors are generated by diluting reagents in mineral oil, while air passing through a bottle containing only mineral oil (M2) serves as the control airflow delivered to the fly during intervals. During intervals, M2 is delivered to the fly, and N is directed to the exhaust to remove waste gas and clean the path. Before odor stimuli, the system enters a preparation state where M2 is delivered to the fly, and M1 is directed to the exhaust to prefill the path with the selected odor. During odor stimuli, M1 is delivered to the fly and M2 is directed to the exhaust. Odor number can be extended as additional ports are available on valves 1, 2. FC: fast mass flow controllers and meter. H: high potential. L: low potential. **b**, $\Delta F/F$ of G7f in response to OCT, MCH, EA, and the brief air-puff (control) in a region of interest (ROI) within the olfactory regions from the center view obtained by 2pSAM. The dark-colored lines represent trial average, and the light-colored lines represent each trial (2 trials of OCT, 1 trial of MCH, 3 trials of EA, and 3 trials of control in a pseudo-random order for each fly; 3 flies with the genotype *w; UAS-rGRAB_ACh-0.5/+; nSyb-Gal4, UAS-jGCaMP7f*). **c**, The box plot of the area under the curve (AUC) of the responses to odors and the control during odor stimuli periods ($n = 18$ for Odor, $n = 9$ for Control). The dots represent the AUC values of each trial. Two-sided Mann-Whitney U test, $P = 0.00019$. **d**, **e**, Similar to **b**, **c**, but for rACh responses. $P = 0.0094$ in **e**. **** $P < 0.0001$, *** $P < 0.001$, ** $P < 0.01$, * $P < 0.05$, ns - not significant ($P > 0.05$).

Fig. 1b - What is the identity of the region showing the strongest G7f signal (Fig. 1b, region next to MB)? It would be helpful to indicate this. The 5-HT signal seems to correlate negatively with G7f activity in this region (compare G7f and r5-HT signals in Fig. 1b, 2a). Please comment.

Response: We sincerely thank the reviewer for the comment. From the maps of response intensity, we could observe that G7f showed a strong response in the region mentioned by the reviewer, while r5-HT exhibited a lower response (**Response Fig. 16a**). According to the brain atlas (**Response Fig. 16c**), this region primarily corresponds to the Superior intermediate protocerebrum (SIP) and the Superior medial protocerebrum (SMP), with a small portion extending into the Superior lateral protocerebrum (SLP). We have indicated them in the revised main texts. We also assessed responsiveness across the brain by calculating the correlation between stimuli and indicator dynamics, as shown in **Response Fig. 16b**. In response to a subsequent comment, we adjusted the calculation method for responsiveness to better demonstrate regional specificity. Although r5-HT shows low response intensity in this region, its responsiveness exhibits a high level (**Response Fig. 16a, b**). We also calculated the correlation between the two signals, G7f and r5-HT, across the brain and created the map (**Response Fig. 16d**). The average of the top 20% correlation values were calculated for each region to make comparisons (**Response Fig. 16e**). SIP and SMP show high correlation between G7f and r5-HT. Additionally, comparing the trial-averaged olfactory responses of the voxels in SIP, SMP, and SLP, the two indicators also represent a positive correlation in total (**Response Fig. 16f**).

Response Fig. 16 | Clarifying identity of the region showing the strongest G7f signal. **a**, Maps of the average standard deviation of $\Delta F/F$ during odor stimulation across trials for G7f, rACh, and r5-HT (averaged across flies, $n = 15$ for G7f, $n = 5$ for rACh, $n = 10$ for r5-HT). **b**, Maps of correlation between G7f, rACh, and r5-HT dynamics and odor stimuli (averaged across flies, $n = 20$ for G7f, $n = 10$ for rACh, $n = 10$ for r5-HT). **c**, Highlighting the related brain regions in the z-projections of the brain atlas. **d**, Maps of the correlation between G7f and r5-HT during odor stimulation (averaged across flies, $n = 10$). **e**, The average of the top 20% correlation values for each region (mean \pm s.e.m, $n = 10$). **f**, Trial-averaged G7f and r5-HT responses to 3 odor stimuli of the sample voxels in 8 olfactory regions (coral) and 2 non-olfactory regions (black). Dashed lines sign the start of odor delivery. Short lines above sign odor stimulus periods (5 s) (blue: OCT; red: MCH; orange: EA). 10 flies co-labeled by G7f and rACh and 10 flies co-labeled by G7f and r5-HT are analyzed in **a**, **b**, **d**, **e**. Flies co-labeled by G7f and rACh with high-intensity non-specific fluorescence on the upper edge of the brain are excluded for clarity in **a**. Scale bars: 50 μm in **a-d**.

Fig. 2a, Ext Fig. 1b - It would be beneficial to see the significance values of these apparently high correlations. Ext. Data Fig. 1b shows that most regions reported have generally high correlation with odor stimulus. Are there regions showing low correlation out of the 43 regions imaged to demonstrate the regional specificity of the response? It would be helpful to show these in the same plot to identify regions specifically involved in responding to odors and odor identification.

Response: We sincerely thank the reviewer for this insightful comment. We have considered the significance values during the calculation of correlation in the reviser manuscript.

In the original manuscript, we calculated the Pearson correlation between trial-averaged $\Delta F/F$ and the stimulus (binary sequence) for each voxel to generate responsiveness maps for each fly. The correlation for each region was determined by averaging the correlations of the voxels within that region. As we used trial-averaged $\Delta F/F$, the correlation values were very high across all recorded regions (we displayed 23 of the 43 regions, focusing on the left-side regions for those with bilateral symmetry). To better capture regional variance and specificity, we reanalyzed the data by concatenating the 180-trial odor responses instead of relying on trial averages (**Response Fig. 17**), which revealed more consistent and reliable odor responses. For the calculation of the

maps, we considered only significant correlations ($P < 0.05$, using the Python package `scipy.stats.pearsonr`) and recorded these values. We then calculated the average of the top 20% correlation values for each region to make comparisons. This approach provides a clearer and more accurate view of the regions specifically involved in responding to odors.

For calcium, all olfactory regions and some non-olfactory regions (SCL, EB, FB, AOTU) displayed high responsiveness (**Response Fig. 17a, b**). For ACh, two olfactory regions, LH and SLP, showed lower responsiveness (**Response Fig. 17a, c**). Regarding 5-HT, its slower response and the pronounced decrease in response with increasing trial numbers resulted in low overall correlations across all regions. However, among the olfactory regions, LH, MBPED, and SLP exhibited even lower responsiveness (**Response Fig. 17a, d**). The results are updated in **Fig. 2** and **Extended Data Fig. 1**.

Response Fig. 17 (Fig. 2a and Extended Data Fig. 1e, f) | Responsiveness to odor stimulation across the brain. a, Maps of correlation between G7f, rACh, and r5-HT dynamics and odor stimuli (averaged across flies, $n = 20$ for G7f, $n = 10$ for rACh, $n = 10$ for r5-HT). **b-d**, The average of the top 20% correlation values for each region (mean \pm s.e.m, $n = 20$ for G7f in **b**, $n = 10$ for rACh in **c**, $n = 10$ for r5-HT in **d**). 10 flies co-labeled by G7f and rACh and 10 flies co-labeled by G7f and r5-HT are analyzed. Scale bar: 50 μ m in **a**.

Fig. 2j - It would be better to have the nodes labeled as in Ext Data Fig. 3, with the names of the

brain regions.

Response: We thank the reviewer for the suggestion. We have revised accordingly.

Fig. 1b and Fig. 2a - Why is the standard deviation of dF/F_0 , and not dF/F_0 , used to represent the response intensity (Methods, p.11, lines 498-499, and Fig. 2a, right panel)? Do the authors mean a z-score, which is the dF/F_0 (trace) divided by the standard deviation?

Response: We sincerely thank the reviewer for this insightful comment. We used standard deviation as the metric for response intensity because it captures the dynamic changes in $\Delta F/F$ during the specific period. A high response is typically associated with significant dynamic changes in $\Delta F/F$, resulting in a larger standard deviation. Other commonly used metrics for response intensity include the mean and the area under the curve (AUC). To ensure robustness, we also calculated the AUC as a supplementary metric, which yielded similar results (**Response Fig. 18**). We added the results in **Supplementary Fig. 1**.

Response Fig. 18 (Supplementary Fig. 1) | Response intensity across the brain. a, Maps of the average standard deviation of $\Delta F/F$ during odor stimulation across trials for G7f, rACh, and r5-HT (averaged across flies, $n = 15$ for G7f, $n = 5$ for rACh, $n = 10$ for r5-HT). **b,** The average standard deviation of each region (mean \pm s.e.m, $n = 15$ for G7f, $n = 5$ for rACh, $n = 10$ for r5-HT). **c, d,** Similar to **a, b,** but for AUC. 10 flies co-labeled by G7f and rACh and 10 flies co-labeled by G7f and r5-HT are analyzed. Flies co-labeled by G7f and rACh with high-intensity non-specific fluorescence on the upper edge of the brain are excluded for clarity. Scale bars: 50 μm in **a, c.**

Fig. 2 - Is the onset of G7f before or after that of rACh or r5-HT signals? It would be helpful if a quantification is provided. Since the response of r5-HT is significantly slower and the authors show it 16 seconds after odor delivery compared to 8 seconds for G7f, it seems that r5-HT does not have a role in triggering the G7f response. This analysis will help elucidate whether the neuromodulators play a role in modulating the G7f signal.

Response: The suggestion to compare the onset of the signals is excellent. We tried this analysis but were unable to detect a clear gap between the start of odor delivery and the onset of responses, as odor response onset appears to be a much faster process than we could capture now at 30 Hz. Given that we did not delve deeply into the neuromodulation properties of ACh and 5-HT, and since this analysis did not yield accurate or significant conclusions, we have decided to remove the comparisons of calcium, ACh, and 5-HT dynamics and only remain the comparisons across brain regions for each indicator.

Global information propagation - It is unclear why the phase delay (defined as the time between the onset and the peak of the response in Methods), rather than the time between the onset of odor delivery and the onset of the signals of individual sensors, is used to construct the schematic of global olfactory information propagation. Phase delay indicates the on kinetics of the sensor, so it is not a good measure for when the information is propagated to a specific region or voxel.

Response: We sincerely thank the reviewer for this insightful comment. As mentioned above, we were unable to detect a clear gap between the start of odor delivery and the onset of responses, as odor response onset appears to be a much faster process than we could capture. Electrophysiology can record such transient dynamics, but is hard to be applied in *Drosophila* across a wide region with high resolution. Instead, the phase delay difference serves as an approximation of information propagation.

The detected responses reflect neuronal activities and the release dynamics of ACh/5-HT, which are convolved with the kinetics of the corresponding indicators. Therefore, the phase delay we measured is not merely a direct reflection of the on-kinetics of the indicators but is also influenced by these kinetics. To better evaluate these observations, a basic description of the kinetics of rACh and r5-HT is necessary. We conducted new experiments on HEK293T cells, finding that the on-kinetics of rACh and r5-HT are 120 ms and 80 ms, and the off-kinetics are 1.82 s and 1.1 s, respectively (**Response Fig. 19a-f**). Fortunately, these on-kinetics are much

shorter than the phase delays we measured, indicating the fidelity of the phase delay.

To be concise and accurate, we have also refined the analyses of global information transmission by including only the olfactory brain regions with significant olfactory responses, namely those regions demonstrating high responsiveness and significant odor classification accuracy (**Response Fig. 19g-j**). We calculated the average response of the brain regions for each indicator and compared the phase delay and pulse width (**Response Fig. 19k-m**). For G7f, MB, especially MBVL exhibited faster response than other regions. For rACh, MBVL and SIP displayed faster response and MB showed significantly longer pulse width. The schematic of the global olfactory information propagation is inferred from the phase delay (**Response Fig. 19n**). The new results of indicator kinetics are added in **Extended Data Fig. 1k, l**, and the analysis of global information propagation is updated in **Fig. 2d, e**.

Response Fig. 19 (Extended Data Fig. 1e, f, i-l, and Fig. 2d, e) | Global information propagation. a-f, The kinetics of rACh and r5-HT. **a**, Measurement of the kinetics of

rACh expressed in HEK293T cells. Schematic illustration showing the local puffing system, Scale bar, 20 μm . **b**, Representative traces of rACh fluorescence increase in response to ACh (top) and decrease in response to Scop (bottom). **c**, Group summary of on and off kinetics for rACh. $n=11$ cells for on and off kinetics, mean \pm s.e.m. **d,e,f**, Kinetic analysis of the r5-HT sensor, similar to **a,b,c**. **g**, Maps of correlation between G7f, rACh, and r5-HT dynamics and odor stimuli (averaged across flies, $n = 20$ for G7f, $n = 10$ for rACh, $n = 10$ for r5-HT). **h**, The average of the top 20% correlation values for each region (mean \pm s.e.m, $n = 20$ for G7f, $n = 10$ for rACh, $n = 10$ for r5-HT). **i**, Maps of odor identity classification accuracy for G7f, rACh, and r5-HT (averaged across flies, $n = 20$ for G7f, $n = 10$ for rACh, $n = 10$ for r5-HT). **j**, The average odor identity classification accuracy of each region (mean \pm s.e.m, $n = 20$ for G7f, $n = 10$ for rACh, $n = 10$ for r5-HT). The gray dashed lines indicate the average accuracy of the non-olfactory regions. One-sided Wilcoxon signed-rank test for each olfactory brain region against the average value. G7f: $P = 0.00031$ for SLP, $P < 0.0001$ for others. rACh: $P = 0.00081$ for MBPED, 0.00081 for MBVL, 0.00037 for MBML, 0.038 for LH, 0.996 for SLP, 0.00081 for SIP, 0.0093 for SMP, 0.00037 for CRE. r5-HT: $P = 0.18$ for MBPED, 0.0034 for MBVL, 0.0034 for MBML, 0.78 for LH, 0.16 for SLP, 0.0073 for SIP, 0.0036 for SMP, 0.0036 for CRE. **** $P < 0.0001$, *** $P < 0.001$, ** $P < 0.01$, * $P < 0.05$, ns - not significant ($P > 0.05$). 10 flies co-labeled by G7f and rACh and 10 flies co-labeled by G7f and r5-HT are analyzed. Scale bars: 50 μm in **g**, **i**. **k-m**, Box plots of phase delay and pulse width of each indicator (G7f in **k**, rACh in **l**, r5-HT in **m**) in olfactory brain regions in the left semi-brain with significant olfactory responses. The dashed lines sign the mean values. $n = 20$ for G7f, $n = 10$ for rACh, $n = 10$ for r5-HT. **n**, Schematic of the global olfactory information propagation across olfactory regions for G7f (green), rACh (purple), and r5-HT (blue) inferred from the phase delay.

Fig. 3 - Are there specific voxels, especially in high-accuracy regions (e.g., LH for G7f and MBML for rACh), that are tuned to a specific odor, thus constituting tuning towards an odor? Would these “maps” be distinct for each of the three odors for any of the three sensors? Identification of voxels encoding each specific odor will enable their functional perturbation (e.g., by optogenetics) to test their functional involvement in odor coding in future studies, in addition to a theoretical description of their existence.

Response: We sincerely thank the reviewer for this constructive suggestion. We have added odor tuning analysis as part of the ensemble analysis. We analyzed odor tuning across the brain, which we defined as the difference in the area under the curve (AUC) of the responses to a

specific odor compared to the other two odors. Voxels with significant tuning values were retained and displayed in **Response Fig. 20a**. We then calculated the standardized ratio of voxels with specific tuning in different brain region, focusing exclusively on brain regions within the community that encompasses the majority of olfactory regions for each indicator. The tuning exhibited a distributed pattern for all indicators. As the reviewer expected, the distributions of tuning for each indicator (**Response Fig. 20b**) and each odor (**Response Fig. 20c**) showed evident distinctions. These results are added in **Fig. 3** and **Extended Data Fig. 7**.

Response Fig. 20 (Fig. 3k and Extended Data Fig. 7a-c) | Odor tuning of voxels across the brain. a, Maps of odor tuning. Colors denote the average difference of area under the curve (AUC) of the responses to the specific odor compared to the other two odors (averaged across flies, $n = 20$ for G7f, $n = 10$ for rACh, $n = 10$ for r5-HT). **b**, **c**, Statistics of the standardized ratio of voxels with specific tuning in each region (mean \pm s.e.m, $n = 20$ for G7f, $n = 10$ for rACh, $n = 10$ for r5-HT). Regions within the community containing most olfactory regions for each indicator are displayed. Each chart shows the result for a specific tuning in **b**, results for a specific indicator in **c**. 10 flies co-labeled by G7f and rACh and 10 flies co-labeled by G7f and r5-HT are analyzed.

Fig. 3d, e, f, middle panels - It is unclear why trials cluster by number according to LDA. The authors mentioned in lines 234-235 that the trial order did not exhibit discernible separation. Please clarify.

Response: We sincerely thank the reviewer for this insightful comment. In these panels, the

colors represent the trial numbers (1-180). This labeling was included to illustrate how the representations change with trial number (which indicates the time) in the low-dimensional manifolds. Interestingly, for G7f and r5-HT, the latter trials exhibit shorter extensions, particularly for r5-HT. However, the trial order did not show discernible separation for rACh, indicating better representation stability. The detailed analyses related to temporal stability in odor representation are presented in Fig. 5. We apologize for not explaining this clearly in the main text and have revised that.

Fig. 4 legends - Is the “deflation ratio” the same as the difference matrix? Its calculation is not explicitly mentioned in Methods. Please clarify.

Response: We sincerely thank the reviewer for the comment. The “deflation ratio” is calculated as $(M - \bar{M}) / \bar{M}$, where M and \bar{M} denote the correlation matrix of G7f (or rACh, r5-HT) and its average value. The difference matrix defined in Methods is the difference between the deflation ratio of G7f and rACh (or r5-HT), shown in **Response Fig. 21**. We have added clear clarification on how to calculate the “deflation ratio” and difference matrix in Methods.

Response Fig. 21 (Fig. 4i, j, Extended Data Fig. 8i, j, and Extended Data Fig. 9a, b) | The deflation ratio and difference matrix. a, b. Functional connectivity matrices (top) and networks (bottom) of the voxels within each of the six brain regions by G7f and rACh during odor stimulation (a) and the resting state (b). Three columns are descriptions of the functional connectivity of G7f, rACh, and the *difference matrix*, from left to right. The *deflation ratio* of G7f (or rACh) is calculated as $(M - \bar{M}) / \bar{M}$, where M and \bar{M} denote the correlation matrix of G7f (or rACh) and its average value. The *difference matrix* is calculated as the difference between the *deflation ratio* of G7f and rACh. **c.** Functional connectivity matrices (top) and networks (bottom) of the voxels within MBML and LH by G7f and r5-HT during odor stimulation. The matrices are ordered and computed in the same way as in (a) and (b).

Fig. 4i - Please state in Methods how the clustering coefficient C is calculated.

Response: We sincerely thank the reviewer for this comment. We have added how to calculate the clustering coefficient C in Methods. In a network with node v_i , the local clustering coefficient C_{v_i} is calculated as $C_{v_i} = 2 * E(v_i) / k(v_i) * [k(v_i) - 1]$, where $E(v_i)$ is the number of edges between the neighbors of node v_i , $k(v_i)$ is the degree of node v_i , i.e., the number of edges connected to it. The global clustering coefficient $\langle C \rangle$ represents the average of

the local clustering coefficient of all the nodes. We calculate the global clustering coefficient as

$$\langle C \rangle = \frac{1}{N} \sum_{i=1}^N C_{v_i}.$$

References:

1. Yang, Q. *et al.* Spontaneous recovery of reward memory through active forgetting of extinction memory. *Current Biology* **33**, 838-848.e3 (2023).
2. Campbell, R. A. A. *et al.* Imaging a Population Code for Odor Identity in the *Drosophila* Mushroom Body. *J Neurosci* **33**, 10568–10581 (2013).
3. Hige, T., Aso, Y., Rubin, G. M. & Turner, G. C. Plasticity-driven individualization of olfactory coding in mushroom body output neurons. *Nature* **526**, 258–262 (2015).
4. Zeng, J. *et al.* Local 5-HT signaling bi-directionally regulates the coincidence time window for associative learning. *Neuron* (2023) doi:10.1016/j.neuron.2022.12.034.
5. Mann, K., Deny, S., Ganguli, S. & Clandinin, T. R. Coupling of activity, metabolism and behaviour across the *Drosophila* brain. *Nature* **593**, 244–248 (2021).
6. Brezovec, B. E. *et al.* Mapping the neural dynamics of locomotion across the *Drosophila* brain. *Current Biology* **34**, 710-726.e4 (2024).
7. Schaffer, E. S. *et al.* The spatial and temporal structure of neural activity across the fly brain. *Nat Commun* **14**, 5572 (2023).

We sincerely thank the reviewers for their thoughtful comments and constructive suggestions that we believe have helped us improve the manuscript substantially. We have provided a point-by-point response letter here and have revised the manuscript accordingly. The original reviewers' comments are shown in black color, whereas for ease of communication, our specific answers are in blue. Before we go into the details of each reviewer's comment and our point-by-point responses, here is a brief top-level summary of our revisions:

1. Improved writing: To address the reviewer's concerns and enhance the overall clarity of our manuscript, we have revised ambiguous expressions and corrected inappropriate word usage as noted in the comments. Additionally, we have carefully reviewed the manuscript and addressed other potential writing issues.

2. More rigorous statistics: In response to the reviewer's comments, we have improved the rigor of the statements and statistics. We have guaranteed statistical verification to statements, presented more comprehensive statistical test results, and explained the logic of test method choice in Methods.

3. Additional information: In response to the reviewer's comments, we have added example traces of GCaMP and rACh from the Mushroom Body regions in Extended Data Fig. 10 to provide a clearer demonstration of the cell-type specific comparison.

REVIEWER COMMENTS

Reviewer #1 (Remarks to the Author):

I appreciate the efforts the authors made in their revision, and I remain impressed by the effort underlying this entire manuscript, but the product itself leaves me uneasy. A big part of this may reflect writing style, which impacts the paper overall. For example, this sentence in the abstract:

“We discovered the compensation between neuronal activity and cholinergic dynamics, both in the odor identity representation across the brain and the functional connectivity network structures of specific brain regions.”

I read this again and again, but am not sure what is said here. I am not trying to be hard, or insistent, I just genuinely don't understand what is stated here. This is kind of word salad... This in part may merely be an English language usage issue, and that, with the help of Nature Editorial team, can be corrected with ease. For example, the above word-salad sentence may make (albeit only partial) sense if the third word, namely “the”, is deleted. Moreover, I think the authors should run a find-and-replace on “we discovered” and replace it with “we found” (many instances), as well as delete the many uses of “interestingly” throughout the manuscript (for example: ”Interestingly, the accuracy became stable after surpassing a certain threshold of dimensions.” - why is this interesting...?). English alone is correctable, and doing so may take this manuscript way further.

Response: We sincerely appreciate the reviewer's valuable comments, which definitely help to enhance the quality of our manuscript. We have carefully revised the manuscript, addressing the writing issues and improving clarity.

Regarding the first issue raised by the reviewer, we acknowledge that certain expressions and the incorrect use of “the” may have led to confusion. To provide a clearer summary of our findings, we have revised the abstract accordingly, with the

specific changes of the original sentence highlighted below:

Presenting three distinct odors across multiple trials, we investigated the spatiotemporal olfactory information representation of these dynamics. Our findings revealed notable differences in response properties, functional connectivity networks, and odor decoding accuracy distribution among these dynamic signals. Furthermore, we constructed low-dimensional manifolds capturing the global odor representation encoded by these distinct signals. **Integrating calcium and ACh dynamics enhanced classification accuracy both locally and in the global low-dimensional manifold. Consistently, we observed complementation in the functional connectivity networks of these two signals in brain regions with accuracy gain.** Moreover, at the temporal scale, low-dimensional manifold and functional connectivity analyses demonstrated the superior stable representation of ACh dynamics.

The inappropriate usage of “we discovered”, “interestingly”, and similar expressions, is avoided in the revised manuscript. Additionally, we have gone through the manuscript and amended other writing issues.

But my issue goes beyond English alone. I remain particularly bothered by the countless statements of fact without statistical verification. For example: Lines 156-157 “For G7f, MB, especially MBVL exhibited faster response than other regions”. In my discipline (indeed not the type of imaging used by the authors), this statement should appear as follows:

“For G7f, MB, especially MBVL exhibited faster response than other regions (MBVL time = $X \pm X$, Other regions mean = $X \pm X$, Statistic (df) = X , Effect size = X , p value = X).

However, the authors just state fact, and I am expected to accept it, or “see” it in the figure (here the authors direct to Figure 2e, but no statistics in 2e legend).

Now if this happened once or twice then fine, but its constant... e.g.:

Line 148: “Interestingly, the response intensities of G7f and rACh exhibited some

degree of complementarity, with specific voxels displaying low intensity for G7f but high intensity for rACh (Fig. 2b, c).” Here I would expect a statistical test of “G7f vs. rACh”.

Line 132: “We discovered that, three indicators displayed evident distinction in responsiveness.” Here I would expect an ANOVA.

I could go on and on, results stated as facts without statistical tests. Notably, some stats were added after my previous comments on this front, but what was added was only p values, without the associated statistics, let alone measures of effect size. One should not report p values alone, and here the authors added entire tables of p values alone. Similarly, the authors report using Kolmogorov-Smirnov tests and Wilcoxon signed-rank tests, but don’t justify why they use which test and when. For example, a test of distribution normality should be used to justify either parametric or non-parametric tests. And why once Kolmogorov-Smirnov and other times Wilcoxon signed-rank? Unclear... Moreover, the tests are occasionally one-sided, again without justification that I could find.

Response: We sincerely thank the reviewer for pointing out the statistical issues. In the revised manuscript, we have added statistical verification to statements, presented more comprehensive statistical test results, and explained the logic of test method choice. Detailed adjustments are listed below.

(1) Statistical verification to statements:

We have carefully checked our statements in this manuscript, adding necessary statistical verification and modifying imprecise statements. Specific responses to the problems pointed out in the reviewer’s comments are as follows.

“For G7f, MB, especially MBVL exhibited faster response than other regions.” We have added the entire set of region-to-region statistical results to the phase delay and pulse width analysis. For phase delay, brain regions display significant distinctions in calcium responses. The MBPED exhibits lower phase delay

than any other brain region. Besides, the MBVL shows significant difference with the LH, SLP, and SMP; the MBML shows significant difference with the LH, SLP, SIP, and SMP; and the SLP shows significant difference with the CRE. For pulse width, the SIP displays significantly shorter pulse width than the MB and CRE in ACh responses. These results are described in the main text, Fig. 2d (**Response Fig. 1**), and the legend. Limited by space, detailed statistical results of each region pair are listed in the Source data (an Excel file containing data and detailed statistical results of all figures and statements in the manuscript, like in **Response Fig. 2**), listing the mean and standard error (SEM) of each brain region compared, the statistic, degrees of freedom (dof), effect size, and p value. For data passing the distribution normality test, results of both parametric and non-parametric tests are shown.

This statement in the main text is adjusted to “For calcium signals, the MB, especially the MBPED, exhibits faster response than other regions (Fig 2d, Source data). For ACh, the SIP displays significantly shorter pulse width than the MB and CRE (Fig 2d, Source data)”.

Response Fig. 1 (Fig. 2d) | Phase delay and pulse width of olfactory responses in specific brain regions. Box plots of phase delay and pulse width of each signal (G7f, rACh, r5-HT from left to right) in olfactory brain regions in the left semi-brain with high olfactory responsiveness. Box plots: center line, median; box limits, upper and lower quartiles; whiskers, 1.5x interquartile range. Each point represents the result of a fly. The dashed lines sign the mean values. $n = 20$ for G7f, $n = 10$ for rACh, $n = 10$ for r5-HT. **Limited by space, only the significance between the MBPED and other brain regions is shown in the phase delay of G7f.** Besides, the MBVL shows significant difference with the LH, SLP, and SMP; the MBML shows significant difference with the LH, SLP, SIP, and SMP; and the SLP shows significant difference with the CRE. In other panels, the

absence of a significance marker indicates no significance. The entire statistical test result is listed in the Source data. 10 flies co-labeled by G7f and rACh and 10 flies co-labeled by G7f and r5-HT are analyzed. Flies co-labeled by G7f and rACh with high-intensity non-specific fluorescence on the upper edge of the brain are excluded for clarity in results of specific regions. Two-sided Mann-Whitney U test and Benjamini/Hochberg multi-comparison correction applied. **** $P < 0.0001$, *** $P < 0.001$, ** $P < 0.01$, * $P < 0.05$, ns - not significant ($P > 0.05$).

MBPED vs. MBVL		MBPED vs. MBML		MBPED vs. LH	
Normality	5, True; $P = 0.2253$, True	Normality	5, True; $P = 0.0548$, True	Normality	5, True; $P = 0.025$, False
Variance equality	$P = 0.7216$, True	Variance equality	$P = 0.3279$, True	Variance equality	$P = 0.595$, True
mean1 ± s.e.m1	0.7852±0.0461	mean1 ± s.e.m1	0.7852 ± 0.0461	mean1 ± s.e.m1	0.7852 ± 0.0461
mean2 ± s.e.m2	1.0583±0.0517	mean2 ± s.e.m2	1.0133 ± 0.0666	mean2 ± s.e.m2	1.2412 ± 0.0598
t test		t test		U test	
Statistic (T-value)	-3.9071	Statistic (T-value)	-2.7588	Statistic (U-value)	11.5
Alternative	Two-sided	Alternative	two-sided	Alternative	two-sided
Degrees of freedom	36	Degrees of freedom	36	Effect size (rank-biserial)	-0.9248
Effect size (Cohen's d)	1.2694	Effect size (Cohen's d)	0.8963	Effect size (common lang)	0.0376
Effect size (Hedges' g)	1.2428	Effect size (Hedges' g)	0.8775	P value	<0.0001
P value	0.0004	P value	0.0091	P value (Benjamini/Hoch)	<0.0001
P value summary	***	P value summary	**	P value summary	****
Significantly different (P <	Yes	Significantly different (P <	Yes	Significantly different (P <	Yes
U test		U test			
Statistic (U-value)	72.5	Statistic (U-value)	96.5		
Alternative	Two-sided	Alternative	two-sided		
Effect size (rank-biserial)	-0.5972	Effect size (rank-biserial)	-0.4639		
Effect size (common lang)	0.2014	Effect size (common lang)	0.2681		
P value	0.0017	P value	0.0151		
P value (Benjamini/Hoch)	0.006	P value (Benjamini/Hoch)	0.0325		
P value summary	**	P value summary	*		
Significantly different (P <	Yes	Significantly different (P <	Yes		

Response Fig. 2 (Source data) | Detailed statistical results listed in the Source data. This is a part of the sheet of Fig. 2d in the Source data showing the statistical results of specific brain region pairs for phase delay comparisons in the calcium responses. For data passing the distribution normality test, results of both parametric and non-parametric tests are shown.

“Interestingly, the response intensities of G7f and rACh exhibited some degree of complementarity, with specific voxels displaying low intensity for G7f but high intensity for rACh (Fig. 2b, c).” This is a qualitative description. In Fig. 2b, c, we randomly sampled voxels from brain regions with G7f and rACh signals from the same voxels in a fly. Some voxels show low intensity for G7f but high intensity for rACh. To be clear, we have labeled these voxels in Fig. 2b, c (**Response Fig. 3**). The statement in the main text is adjusted to “Additionally, the response intensity of calcium and ACh exhibits some degree of complementation, with specific voxels displaying low calcium intensity but high ACh intensity (indicated by black arrows in Fig. 2b, c)”.

Response Fig. 3 (Fig. 2b, c) | Heterogeneous and distinct olfactory responses for G7f, rACh, and r5-HT. a, Trial-averaged G7f, rACh, and r5-HT responses to 3 odor stimuli of the sample voxels in 8 olfactory regions (coral) and 2 non-olfactory regions (black) of a fly. Dashed lines sign the start of odor delivery. Short lines above sign odor stimulus periods (5 s) (blue: OCT; red: MCH; orange: EA). G7f and rACh responses are from a co-labeled fly, and the r5-HT response is from another fly co-labeled by G7f and r5-HT. **b**, $\Delta F/F$ traces of some voxels in the MBVL, MBML, and LH across several trials. The gray lines sign the start of odor delivery. **Black arrows indicate voxels with low response intensity for G7f but high response intensity for rACh in a, b.**

“We discovered that, three indicators displayed evident distinction in responsiveness.” We have added an ANOVA to this statement, adapting this sentence as “We found that three dynamics displayed evident distinction in responsiveness distribution across the brain (two-way ANOVA for factors “dynamics” and “brain regions”: $P < 0.0001$ for both factors, $F = 43.2392$, degrees of freedom = 2, and effect size (partial eta-squared) = 0.7004 for “dynamics”. The non-parametric Scheirer-Ray-Hare test yielded similar significance; see the Source data for details)”. Detailed statistical results are listed in the Source data (**Response Fig. 4**). This statement is also supported by the maps of responsiveness and relative statistics of the responsiveness of brain regions (**Response Fig. 5**).

ANOVA test in results	
Distinction in responsiveness across three dynamics	
Normality (G7f)	P = 0.2325, True
Normality (rACh)	P = 0.4996, True
Normality (r5-HT)	P = 0.4996, True
Variance equality	P < 0.0001, False
Two-way mixed-design ANOVA (pingouin.mixed_anova)	
Greenhouse-Geisser correction for variance inequality	
Factor name	dynamics
Sums of squares	4.9408
Degrees of freedom	2
Mean squares	2.4704
F-values	43.2392
Effect size (np2)	0.7004
P value	<0.0001
P value summary	****
Significantly different (P < 0.05)?	Yes
Scheirer-Ray-Hare test	
Statistic (H-value)	319.5555
Degrees of freedom	2
P value	<0.0001
P value summary	****
Significantly different (P < 0.05)?	Yes

Response Fig. 4 (Source data) | Statistical test results of the responsiveness distinction among different dynamic signals. Distribution normality test, two-way ANOVA, and the non-parametric Scheirer-Ray-Hare test applied.

Response Fig. 5 (Extended Data Fig. 1e, f) | Responsiveness across brain

regions. a, Maps of correlation between G7f, rACh, and r5-HT dynamics and odor stimuli (averaged across flies, $n = 20$ for G7f, $n = 10$ for rACh, $n = 10$ for r5-HT). **b**, The average of the top 20% correlation values for each region (mean \pm s.e.m). The gray dashed lines indicate the average level of the non-olfactory regions. One-sided Wilcoxon signed-rank test for each olfactory brain region against the average of the non-olfactory regions performed. Scale bar: 50 μm in **a**. **** $P < 0.0001$, *** $P < 0.001$, ** $P < 0.01$, * $P < 0.05$, ns - not significant ($P > 0.05$).

We have carefully reviewed the manuscript and addressed other related issues, adding statistical verification and modifying inaccurate statements.

(2) More comprehensive statistical test results:

We have recognized the necessity of presenting more comprehensive statistical test results. Accordingly, some results have been incorporated into the main text and figure legends. Due to space constraints, the full statistical test results are provided in the Source data, including the mean, standard error of the mean (SEM), test statistic, degrees of freedom (dof), effect size, and p value (**Response Fig. 2**). For data that passed the normality test, we present results from both parametric and non-parametric tests, whereas for data that did not pass, only non-parametric test results are reported. This information is explicitly stated in the main text and figure legends, guiding readers to the Source data for detailed statistical results.

MBPED vs. MBVL		MBPED vs. MBML		MBPED vs. LH	
Normality	5, True; $P = 0.2253$, True	Normality	5, True; $P = 0.0548$, True	Normality	5, True; $P = 0.025$, False
Variance equality	$P = 0.7216$, True	Variance equality	$P = 0.3279$, True	Variance equality	$P = 0.595$, True
mean1 \pm s.e.m1	0.7852 \pm 0.0461	mean1 \pm s.e.m1	0.7852 \pm 0.0461	mean1 \pm s.e.m1	0.7852 \pm 0.0461
mean2 \pm s.e.m2	1.0583 \pm 0.0517	mean2 \pm s.e.m2	1.0133 \pm 0.0666	mean2 \pm s.e.m2	1.2412 \pm 0.0598
t test		t test		U test	
Statistic (T-value)	-3.9071	Statistic (T-value)	-2.7588	Statistic (U-value)	11.5
Alternative	Two-sided	Alternative	two-sided	Alternative	two-sided
Degrees of freedom	36	Degrees of freedom	36	Effect size (rank-biserial)	-0.9248
Effect size (Cohen's d)	1.2694	Effect size (Cohen's d)	0.8963	Effect size (common lang)	0.0376
Effect size (Hedges' g)	1.2428	Effect size (Hedges' g)	0.8775	P value	<0.0001
P value	0.0004	P value	0.0091	P value (Benjamini/Hochl)	<0.0001
P value summary	***	P value summary	**	P value summary	****
Significantly different (P <	Yes	Significantly different (P <	Yes	Significantly different (P <	Yes
U test		U test			
Statistic (U-value)	72.5	Statistic (U-value)	96.5		
Alternative	Two-sided	Alternative	two-sided		
Effect size (rank-biserial)	-0.5972	Effect size (rank-biserial)	-0.4639		
Effect size (common lang)	0.2014	Effect size (common lang)	0.2681		
P value	0.0017	P value	0.0151		
P value (Benjamini/Hochl)	0.006	P value (Benjamini/Hochl)	0.0325		
P value summary	**	P value summary	*		
Significantly different (P <	Yes	Significantly different (P <	Yes		

Response Fig. 2 (Source data) | Detailed statistical results listed in the Source data. This is a part of the sheet of Fig. 2d in the Source data showing the statistical results of specific brain region pairs for phase delay comparisons in the calcium responses. For data passing the distribution normality test, results of both parametric and non-parametric tests are shown.

(3) **The logic of test method choice:**

We used non-parametric tests in the last version considering the small sample size we used. Distribution normality test may not be effective in this condition, and non-parametric tests are robust with fewer distribution assumptions. **To solidify our results, we have added parametric test results for data passing the distribution normality test, which provide similar results to the non-parametric tests.** The entire statistical test results are listed in the Source data, and results of the non-parametric tests are reported in the main text.

We mainly used five different statistical test methods, Mann-whitney U test for unpaired comparisons of two groups, Wilcoxon signed-rank test for paired comparisons, Kolmogorov-Smirnov test for comparisons of the distribution of two groups, Kruskal-Wallis test for comparisons of multiple groups (the non-parametric version of ANOVA), and Scheirer–Ray–Hare test for two factors (non-parametric two-way ANOVA test). Mann-whitney U test in this manuscript is used for comparisons between different signals or brain regions which do not consider the pairing of individuals. Whereas Wilcoxon signed-rank test is used for tests where pairing matters, like the statistics of accuracy gain. In the revised manuscript, the corresponding parametric tests are added in the Source data when data passes the distribution normality test.

We used two-sided tests to show the difference of groups and used one-sided tests to examine whether the specific group of data shows a significantly higher / lower level compared to the other group or a certain threshold. For example, in Fig. 4b (**Response Fig. 6a**), we used one-sided tests to show whether each brain has significant accuracy gain (higher than 0). In Fig. 4c (**Response Fig. 6b**), we used one-sided test to show whether integrating two channels yielded significant accuracy gain at the multiple-brain-region level (accuracy significantly higher than G7f).

We have added a section in Methods explaining how statistical tests are performed.

Response Fig. 6 | Accuracy gain in local brain regions and at the multiple-brain-region level. a, Statistics of the average accuracy gain in each brain region. **b**, Comparison of the voxel-level multiple-brain-region odor identity classification accuracies between using only the G7f channel and integrating both channels. $n = 10$ flies co-labeled by G7f and rACh, mean \pm s.e.m. One-sided Wilcoxon signed-rank test applied. **** $P < 0.0001$, *** $P < 0.001$, ** $P < 0.01$, * $P < 0.05$, ns - not significant ($P > 0.05$, not shown in a).

All and all, this remains a very impressive, and I think important data set, presented within a manuscript that I cannot say is good. I don't really know what to recommend here, as I don't think this effort should be rejected, nor would I publish it as is. Perhaps my review should receive less weight than the other Referees, and if they are positive, then go ahead and publish.

Response: We genuinely thank the reviewer for the suggestions and acknowledgement. Your insightful suggestions will help our work go further and arise broader interest. We have carefully adjusted our manuscript in aspects of writing and statistics, hoping to improve its clarity and rigor. We are open to additional suggestions and improve our manuscript further.

Reviewer #2 (Remarks to the Author):

I thank the authors for providing a substantial revision which I believe have greatly improved the clarity of their text. These revisions along with additional experiments have improved the manuscript and addressed most of our previous concerns. After

minor changes, this manuscript would be ready for publication.

Response: We sincerely thank the reviewer for the acknowledgement of our revision. Your suggestions are valuable for improving our manuscript. We have made some additional changes following your comments, hoping to address your concerns.

Remaining issue:

In the Extended Data Fig. 10, it would be better to include example traces of GCaMP and rACh from the Mushroom Body regions (MBPED, MBVL, MBHL). Some plots similar to Fig. 2b,c will provide a clearer demonstration of this cell-type specific comparison. I'm particularly curious to see if the MB-specific GCaMP and rACh signals attenuate less across the 180 odor trials compared to the traces in Fig. 2c. This could be the reason why the manifolds are stable across all four stages now.

Response: We sincerely thank the reviewer for the insightful comments. We have added example traces for the MB-specific signals in Extended Data Fig. 10a, b (**Response Fig. 7**). The ACh response in the MBML tends to show higher intensity than G7f, consistent to the result of the pan-neuronal labeling in Fig. 2a, b (**Response Fig. 3**). The attenuation across trials is not obvious.

Response Fig. 7 | Odor response of the mushroom body cholinergic cells.

a, Trial-averaged G7f and rACh responses to 3 odor stimuli of the sample voxels in the MBPED, MBVL, MBML of a fly. Dashed lines sign the start of odor delivery. Short lines above sign odor stimulus periods (5 s) (blue: OCT; red: MCH; orange: EA). **b**, $\Delta F/F$ traces of some voxels in the MBPED, MBVL, MBML across several trials. The gray lines sign the start of odor delivery.

Response Fig. 3 (Fig. 2b, c) | Heterogeneous and distinct olfactory responses for G7f, rACh, and r5-HT. a, Trial-averaged G7f, rACh, and r5-HT responses to 3 odor stimuli of the sample voxels in 8 olfactory regions (coral) and 2 non-olfactory regions (black) of a fly. Dashed lines sign the start of odor delivery. Short lines above sign odor stimulus periods (5 s) (blue: OCT; red: MCH; orange: EA). G7f and rACh responses are from a co-labeled fly, and the r5-HT response is from another fly co-labeled by G7f and r5-HT. **b**, $\Delta F/F$ traces of some voxels in the MBVL, MBML, and LH across several trials. The gray lines sign the start of odor delivery. **Black arrows indicate voxels with low response intensity for G7f but high response intensity for rACh in a, b.**

Reviewers' comments:

Reviewer #1 (Remarks to the Author):

I am sorry and feel badly about saying this, as I appreciate the authors have invested a lot of work, and yet, I cannot say that I think this is a good manuscript. For one, I remain irked by the style of reporting. This starts already at the abstract:

“Collectively, our unbiased and comprehensive investigation highlights the prominent involvement of ACh dynamics in stable olfactory representation across the brain, underscoring the inadequacy of solely considering neuronal activities when examining information representation of the brain.”

What do you mean by “unbiased”? It is in fact super-biased... you looked at ACh/5-HT dynamics alone. Now, that is absolutely fine that you did that, it's your measure, and it's informative, but it is not unbiased... all measures, by definition, are biased. Moreover, I can't understand why you insist on all this “underscoring the inadequacy of solely considering neuronal activities”, rather than concentrate on what it is you add....

Response:

We apologize if our writing style caused discomfort for the reviewer. We believe that the specific word choices and expressions do not affect the key findings of the manuscript and can be removed or rephrased to avoid confusion. While we have removed these descriptive terms and accordingly revised other expressions in this version, we would like to briefly explain the rationale behind their initial inclusion:

The term “unbiased” refers to a comprehensive, multiple-region screening and mapping of neural activities and dynamic signals, without preconceived emphasis on specific brain regions or cell types. We aimed to emphasize this point as this term is widely used in current large field-of-view imaging studies and reviews (e.g., “This study highlights the power of using an unbiased, brain-wide approach for mapping the functional organization of sensory activity.” (Pacheco *et al.*, *Nature Neuroscience* 2021), “... in part via the unprecedented ability to perform unbiased neural activity screens for principles of brain function, spanning dozens

of brain areas and from local to global scales” (Machado, Kauvar and Deisseroth, *Nature Reviews Neuroscience* 2022)).

Regarding the phrase “underscoring the inadequacy of solely considering neuronal activities”, our intent is to convey that simultaneously recording and analyzing calcium and ACh dynamics yields more information, and that only considering neuronal activities may not provide a comprehensive understanding of the brain. While we fully acknowledge that neuronal activity analysis remains central in this field, our results suggest that neurochemical signals may play a previously underappreciated role in information representation. That said, we have rephrased this statement in the revised manuscript to avoid potential misunderstanding.

The abstract is revised to:

“Despite the vital role of neuromodulators and neurotransmitters in the neural system, their spatiotemporal correlation with neuronal activities across multiple brain regions remain unclear. Here, we employed two-photon synthetic aperture microscopy (2pSAM) and neurochemical indicators to simultaneously record calcium and acetylcholine (ACh) / 5-HT dynamics across multiple regions of the *Drosophila* brain over 2 hours. Presenting 3 different odors across multiple trials, our analyses revealed signal-specific differences in responsiveness, functional connectivity, and odor classification accuracy across the brain. We further constructed low-dimensional manifolds to characterize the global odor-related dynamics. Incorporating both calcium and ACh signals enhanced odor classification accuracy in the global low-dimensional manifold and in specific brain regions where their functional connectivity network features exhibited complementary patterns. Moreover, ACh dynamics demonstrated relatively stable temporal characteristics compared to calcium and 5-HT. These results suggest the potential contribution of ACh to consistent odor representations and illustrate the utility of multi-signal imaging in studying neural computation.”

Additionally, we would like to note that, in terms of the key findings of this manuscript, other reviewers have previously described our manuscript as “solid and well-presented” and that “Most of the statistical analyses in this study are rigorous and appropriately used”.

Therefore, we have thoroughly revised the manuscript and eliminated unnecessary and inappropriate descriptions unrelated to the key findings.

But more substantively regarding reporting style, for example, page 12:

"Remarkably, we observed substantial accuracy gain upon integrating ACh signals at the multiple-brain-region level (Fig. 4c)."

Here (and CONSTANTLY throughout the manuscript), no values or statistics reported in the text. What is the "remarkable gain"? from what value to what value? If i try to pull it out of the figure (that doesn't start at 0, i.e., its stretched), it looks like a shift from about 94% to about 97%. Is that a remarkable gain? It does look consistent in the figure, but remarkable...? The manuscript is just full of this....

Response:

We are sorry for confusing the reviewer with our reporting style. In response to the reviewer's comments, we have added some values and statistics in the main text and revised the inappropriate adjective usages.

In response to the reviewer's comments in the last round of revision, we added a comprehensive set of statistical results—including all requested metrics—for every analysis. Given the large number of tests conducted, these statistics were presented in the Source Data file, which contained 85 pages of detailed information. Accordingly, they were not included in the main text, a decision that was clearly explained in our prior response letter. The level of statistical detail we provided exceeds the standard in many comparable publications, where typically only the sample size (n) and p-values or significance indicators are reported in the figure legends (Mann *et al.*, *Nature* 2021; Münch, Goldschmidt and Ribeiro, *Nature* 2022; Zeng *et al.*, *Neuron* 2023). Our initial version followed this practice. Still, we invested great effort in re-running the statistical analyses to fully address the reviewer's concerns and improve our manuscript. Therefore, we respectfully disagree with the assertion that we "CONSTANTLY" provide "no values or statistics", as it does not reflect the content of our

revised submission.

To facilitate understanding, we have incorporated some key values and statistics in the main text in this revised version, while still directing readers to the Source data for more details. For example, we have added in lines 234-236 that “The overall performances for calcium and ACh are comparable (calcium (mean \pm s.e.m): 92.14% \pm 1.33%, ACh (mean \pm s.e.m): 94.83% \pm 1.88%; two-sided Mann-Whitney U test: U-value = 61, Effect size (rank-biserial correlation) = -0.39, P = 0.09; Fig. 3g)”. And for the parts where the long list of statistics is hard to include, we direct readers by “see the Source data for detailed statistical results” or “Detailed statistical test results in the Source data”.

The use of the term “remarkable” was intended to reflect the statistical significance and consistency of the observed accuracy gains. We have found the unfitness of using this word here, and have carefully revised this and other adjectives. We have deleted unnecessary and inappropriate adjectives and adverbs, such as “remarkable”, “substantial”, “significant”, “particularly”, “notably”, etc. This specific sentence is revised to “We found accuracy gain upon integrating ACh signals at the multiple-brain-region level (difference (mean \pm s.e.m): 4.39% \pm 0.94%; one-sided Wilcoxon signed-rank test: W-value = 55, Effect size (rank-biserial correlation) = 1, P = 0.00098; Fig. 4c)”. While we are willing to adjust these terms to avoid misunderstanding, we believe this wording issue does not affect the validity or rigor of our conclusions.

Regarding the accuracy gain, although the accuracy of calcium is already high (an expected outcome), the integration still yields accuracy gain in each fly. Our primary focus has been on establishing the robustness of decoding performance enhancement phenomena, rather than investigating methodological approaches to maximize accuracy improvements. Further, comparisons in single regions, ensembles, and network analysis are provided to support this finding. Importantly, our conclusions are not solely dependent on this one result; they are grounded in multiple lines of evidence and statistically significant results.

Beyond all this, and MOST importantly, what is it that here shows us that acetylcholine

dynamics have a role in “stable olfactory representation”? Is there anything causal here? The fact that the authors improve THEIR decoding accuracy by integrating ACh signals doesn't imply that the fly brain is doing the same! The way things stand, the manuscript is a very impressive data collection on the dynamics of ACh/5-HT in the drosophila brain during odor exposure. It could be that this data will be useful. I don't know.

Response:

We would like to clarify that our study is observational and aims to characterize the informational content of ACh dynamics.

The central conclusion of our study—“Prominent involvement of acetylcholine dynamics in stable olfactory information representation across the *Drosophila* brain”—is supported by a comprehensive set of analyses, including: 1) ACh dynamics enables high-accuracy odor classification across multiple brain regions. 2) Integrating ACh signals with calcium dynamics enhances odor classification performance, and this is validated through functional network analysis. 3) Compared to calcium and 5-HT dynamics, ACh exhibits relatively stable temporal characteristics in odor representation. These conclusions were derived through rigorous analytical methods applied to the large-scale dataset.

The integration of calcium and ACh signals was employed purely as an analytical strategy to examine whether simultaneous imaging reveals additional information. This approach does not imply that such integration necessarily occurs within the fly brain. As previously clarified in our first-round response, causal interpretation lies beyond the scope of the present study. To prevent misleading, we have carefully revised the expressions which may cause misunderstanding and concerns of the causality between signals. We have also added discussions of causality as a limitation of our study and future directions (lines 458-460, “Further investigations of genetically targeted circuits and neuron types combined with targeted manipulations are necessary to further elucidate the mechanisms underlying our findings and the causality between different dynamics.”).

To address the reviewer's concerns regarding this study's advancement, we clarify the main contributions of this work. We have performed the first simultaneous imaging of

calcium and neurochemical dynamics across multiple brain regions of *Drosophila*, and conducted a comprehensive set of analyses to draw the key conclusion. This observational study proposes a novel perspective, informing and delivering data support for further investigations. The properties of brain-wide ACh dynamics may serve as a foundation and inspire future investigations into its specific roles and mechanisms.

Lastly, we appreciate the reviewer's recognition of our open-source dataset. We believe it represents a valuable resource that can support future research efforts in the field.

Reviewer #2 (Remarks to the Author):

I thank the authors for addressing the final points raised in my previous review. I have reviewed the latest revisions and the accompanying response. The changes made have successfully resolved my remaining concerns. I am satisfied with the current version and have no further suggestions for improvement. I now recommend the manuscript for publication.

Response:

We sincerely thank the reviewer for the time and efforts invested in evaluating our work.

References:

Machado, T.A., Kauvar, I.V. and Deisseroth, K. (2022) 'Multiregion neuronal activity: the forest and the trees', *Nature Reviews Neuroscience*, 23(11), pp. 683–704. Available at: <https://doi.org/10.1038/s41583-022-00634-0>.

Mann, K. *et al.* (2021) 'Coupling of activity, metabolism and behaviour across the *Drosophila* brain', *Nature*, 593(7858), pp. 244–248. Available at: <https://doi.org/10.1038/s41586-021-03497-0>.

Münch, D., Goldschmidt, D. and Ribeiro, C. (2022) 'The neuronal logic of how internal states control food choice', *Nature*, 607(7920), pp. 747–755. Available at: <https://doi.org/10.1038/s41586-022-04909-5>.

Pacheco, D.A. *et al.* (2021) 'Auditory activity is diverse and widespread throughout the central brain of *Drosophila*', *Nature Neuroscience*, 24(1), pp. 93–104. Available at: <https://doi.org/10.1038/s41593-020-00743-y>.

Zeng, J. *et al.* (2023) 'Local 5-HT signaling bi-directionally regulates the coincidence time window for associative learning', *Neuron* [Preprint]. Available at: <https://doi.org/10.1016/j.neuron.2022.12.034>.

REVIEWERS' COMMENTS

Reviewer #1 (Remarks to the Author):

The authors addressed the comments, and the manuscript provides valuable data

Response:

We sincerely thank the reviewer for the time and efforts invested in evaluating our work.